# Ideal fracton superfluids

Jay Armas,[1] Emil Have[2]

[1] *Institute for Theoretical Physics and Dutch Institute for Emergent Phenomena, University of Amsterdam, 1090 GL Amsterdam, The Netherlands*

[2] *School of Mathematics and Maxwell Institute for Mathematical Sciences, University of Edinburgh, Peter Guthrie Tait Road, Edinburgh EH9 3FD, UK*

*E-mail:* j.armas@uva.nl, emil.have@ed.ac.uk

ABSTRACT: We investigate the thermodynamics of equilibrium thermal states and their near-equilibrium dynamics in systems with fractonic symmetries in arbitrary curved space. By explicitly gauging the fracton algebra we obtain the geometry and gauge fields that field theories with conserved dipole moment couple to. We use the resultant fracton geometry to show that it is not possible to construct an equilibrium partition function for global thermal states unless part of the fractonic symmetries is spontaneously broken. This leads us to introduce two classes of fracton superfluids with conserved energy and momentum, namely $p$-wave and $s$-wave fracton superfluids. The latter phase is an Aristotelian superfluid at ideal order but with a velocity constraint and can be split into two separate regimes: the U(1) fracton superfluid and the pinned $s$-wave superfluid regimes. For each of these classes and regimes we formulate a hydrodynamic expansion and study the resultant modes. We find distinctive features of each of these phases and regimes at ideal order in gradients, without introducing dissipative effects. In particular we note the appearance of a sound mode for $s$-wave fracton superfluids. We show that previous work on fracton hydrodynamics falls into these classes. Finally, we study ultra-dense $p$-wave fracton superfluids with a large kinetic mass in addition to studying the thermodynamics of ideal Aristotelian superfluids.

# 1   Introduction

Fracton phases of matter have received considerable attention in recent years [1–5] because of their potential applications in quantum information [6–9], ultracold atoms [10, 11] and holography [12, 13], and because they offer new perspectives on systems with constrained excitations including supercooled liquids (see, e.g., [14, 15] for a review). Indeed, it has been found that certain conventional phases, at least in appropriate linear regimes, can be recast as fracton phases, such as ordinary crystals with topological defects and fluids as well as superfluids with vortices [14–16].

Most phases of matter can be characterised by a particular pattern of symmetry breaking, and fracton phases of matter appear to be no exception. These exotic phases are characterised by *fractonic symmetries* which impose various mobility constraints on inherent quasi-particles (i.e., fractons, lineons or planons). The low-energy effective field theories that govern the dynamics of such phases naturally couple to Aristotelian background geometries in combination with U(1) and higher-rank symmetric gauge fields that implement multipolar symmetries [17, 18], although the dynamics governing certain symmetric tensor gauge theories [4, 19–21] are only consistent on certain special backgrounds [17, 18, 22].

Given the novelty of these phases of matter, an important problem is to identify unique experimental signatures of many-body quantum systems with fractonic symmetries. To this end, one seeks to understand how to characterise the equilibrium thermodynamics of such systems and their low-energy, long-wavelength near-equilibrium excitation spectra. In other words, one would like to develop a consistent framework for *fracton hydrodynamics* based on symmetry principles, a well-defined gradient expansion and the second law of thermodynamics. The aim of this paper is precisely to formulate different classes of fracton hydrodynamics in generic curved spacetimes, and to identify the low energy spectra on flat backgrounds.

Many approaches to fracton hydrodynamics have been pursued recently. In particular, [23–25] have used the fracton algebra and associated Poisson brackets in flat space to define classes of ideal and dissipative fracton hydrodynamics by postulating that only momentum gradients appear in the constitutive relations; while others [26–29] have used Schwinger–Keldysh effective field theory to formulate classes of zero and finite temperature dissipative fracton hydrodynamics in flat space and mostly without conserved energy currents. Here we complement and provide a different perspective to these approaches by formulating fracton hydrodynamics based on the

geometry to which the field theories are coupled and their associated conservation laws (Ward identities); in particular we do not postulate any such conservation laws but derive them from symmetry principles. Our analysis will be restricted to ideal order and thus we will not consider dissipative effects, except in specific cases. Below we describe in further detail the results of this work.

One of the central results of our work is the construction of an equilibrium partition function for many classes of fracton hydrodynamics with conserved dipole moment and non-zero charge density, following the methods of [30–32]. This requires a consistent coupling of the fluid to the appropriate background geometry and gauge fields. To achieve this, we explicitly gauge the fracton algebra using the methodology of [33]. The gauging of the spacetime symmetries leads to Aristotelian geometry, while the gauging of the dipole symmetry leads to a U(1) gauge field $B_\mu$ and a gauge field $\tilde{A}_\mu^a$. In particular, we show that by setting the U(1) curvature to zero, the gauge field $\tilde{A}_\mu^a$ reduces to the spatially symmetric tensor gauge field $A_{\mu\nu}$ considered in [17, 18, 20, 21].[1]

Armed with the appropriate geometry, we show that, as in the case of higher-form symmetries at finite temperature [34–38], it is not possible to find global thermal states with non-zero charge density and non-zero fluid flow unless the fractonic symmetry is spontaneously broken.[2] This implies that there are no classes of fracton fluids, and instead we uncover different classes of fracton superfluids (see Fig. 1). We show that these classes are sufficiently broad to encompass much previous work on fracton hydrodynamics.

Given the existence of U(1) and dipole global symmetries, there are two possible patterns of spontaneous symmetry breaking, one of which can be split into two different regimes. When the dipole symmetry is spontaneously broken, there is a massless vector[3] Goldstone field $\Psi_\mu$ in the hydrodynamic theory. Adopting the terminology of [41] we refer to this phase as $p$-wave fracton superfluids, which is also the phase discussed in [29]. If in addition we spontaneously break the U(1) symmetry another Goldstone field $\phi$ is introduced and we land in a $s$-wave fracton superfluid phase, again adopting the terminology of [41]. The combination of the two Goldstone fields allows for a "mass term" for $\Psi_\mu$ in the effective action akin to the mass terms appearing in [38, 42]. Depending on the strength $\ell'$ of that mass term it is possible to identify two different regimes of $s$-wave fracton superfluids. For large $\ell'$ the vector Goldstone can be integrated out, leading to a regime in which only the

---

[1]The vanishing of the U(1) curvature corresponds physically to the absence of elementary dipoles, as pointed out in [18].

[2]This is in line with the work of [29] where it was argued, using a version of the Mermin–Wagner theorem, that the dipole symmetry must be spontaneously broken.

[3]Formally, $\Psi_\mu$ are the components of a one-form. To avoid confusion with the terminology used in the context of higher-form symmetries, see, e.g., [39, 40], where $p$-form Goldstones transform as gauge fields, we refer to the Goldstone $\Psi_\mu$ as a *vector* Goldstone field.

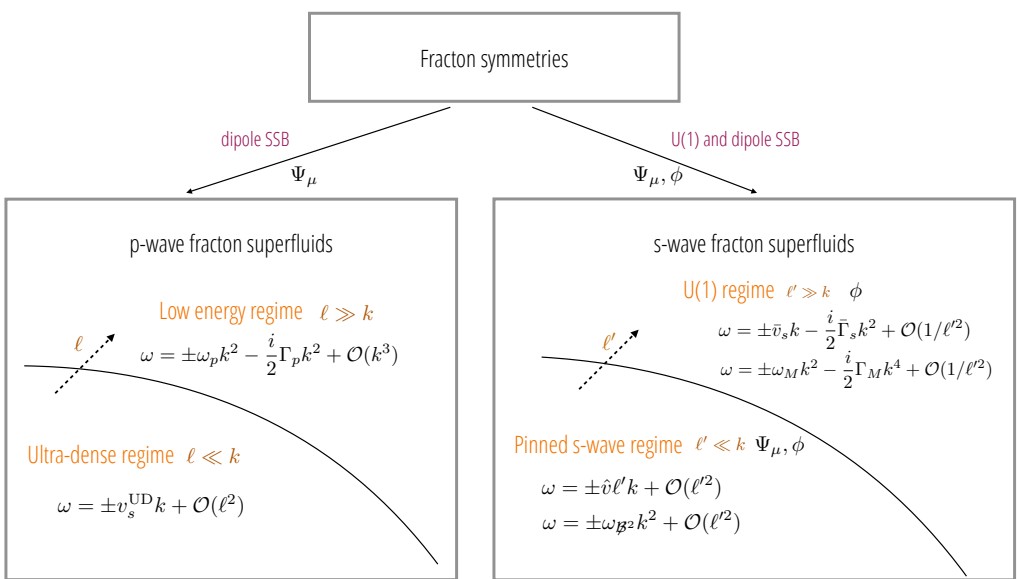

**Figure 1**: Diagram of ideal fracton superfluid phases and regimes studied in this paper. The phase of $p$-wave superfluids occurs due to spontaneous symmetry breaking (SSB) of dipole symmetry leading to a vector Goldstone $\Psi_\mu$. It is possible to introduce a parameter $\ell$ controlling the relative strength of the kinetic mass. For large $\ell \gg k$ with $k$ the wavenumber probing the system, the kinetic mass is small and we are in true low energy regime of $p$-wave superfluids that exhibits magnon-like (quadratic) dispersion relations with velocity $\omega_p$ and attenuation $\Gamma_p$. On the other hand, for small $\ell \ll k$ we are in a regime of ultra-dense fracton superfluid and the spectrum has a sound mode with velocity $v_s^{\mathrm{UD}}$ of order $\mathcal{O}(\ell)$. In turn the $s$-wave fracton superfluid is the result of SSB of U(1) and dipole symmetries leading to a scalar $\phi$ and vector $\Psi_\mu$ Goldstone fields. Two regimes can be distinguished depending on the strength $\ell'$ of a mass term for $\Psi_\mu$ that features in the effective action. For large $\ell' \gg k$ we are in a U(1) fracton superfluid regime in which $\Psi_\mu$ can be integrated out leading to sounds modes with velocity $\bar{v}_s$ and attenuation $\bar{\Gamma}_s$ and to magnon modes with velocity $\omega_M$ and subdiffusive attenuation $\Gamma_M$. Otherwise, for $\ell' \ll k$ we are in a pinned $s$-wave fracton superfluid regime in which both Goldstone fields feature in the theory and the mode structure also contains sound modes with velocity $\hat{v}$ and magnon modes with velocity $\omega_{\mathcal{B}^2}$. The direction of the dashed lines denotes the direction of increasing $\ell$ and $\ell'$. The solid curved line represents the separation between the two regimes.

Goldstone $\phi$ features in the hydrodynamic theory with both U(1) and dipole symmetries remaining spontaneously broken. We refer to this regime as a U(1) fracton superfluid. On the other hand, in the small $\ell'$ regime of $s$-wave fracton superfluids,

which we refer to as a pinned $s$-wave fracton superfluids, both Goldstone fields feature the hydrodynamic theory leading to novel effects (see Fig. 1). This latter regime is a conventional Aristotelian superfluid at ideal order but with an additional velocity constraint. Given its relevance to fracton hydrodynamics, we study Aristotelian superfluids using the formalism of equilibrium partition functions (see Appendix B).

The field content of the different phases of fracton superfluids consists of the temperature $T$, a fluid velocity $u^\mu$, a set of chemical potentials $\mu^I$, where $I$ is a label, and the Goldstone fields $\phi$ and/or $\Psi_\mu$ coupled to the Aristotelian spacetime geometry, parameterised by a clock form $\tau_\mu$ and a spatial metric $h_{\mu\nu}$, and the gauge fields $B_\mu$ and $A_{\mu\nu}$. In order to formulate a hydrodynamic theory, a gradient ordering must be specified. In conventional theories of superfluidity, space and time derivatives have the same gradient ordering but, contrary to conventional superfluids, in $p$-wave fracton superfluids time derivatives scale as $\mathcal{O}(\partial^2)$ and spatial derivatives as $\mathcal{O}(\partial)$ in agreement with the observations made in [25]. Furthermore, in typical hydrodynamic theories of Aristotelian fluids [43–47] the fluid velocity is of ideal order, but a central result of this work is the observation that for $p$-wave fracton superfluids the spatial velocity is derivative suppressed $u^i \sim \mathcal{O}(\partial)$; a result which follows from the consistent definition of the chemical potentials $\mu^I$ and from the dipole Ward identity. Furthermore, in the case of $s$-wave fracton superfluids there are two possible gradient expansion schemes, both with peculiar features. In Appendix C we study in detail the scheme we used for $p$-wave superfluids applied to $s$-wave superfluids while in the main text we adopt the alternative gradient scheme proposed in [48] in which time and space derivatives scale as usual with $\mathcal{O}(\partial)$. Within the context of this latter scheme we show that $s$-wave superfluids are Aristotelian superfluids at ideal order but with a constraint on the spatial fluid velocity, though the spatial fluid velocity is not gradient suppressed in this case. In Section 6 we discuss some of the issues with both schemes.

The above features make $p$-wave fracton hydrodynamics different from earlier work on Aristotelian hydrodynamics [43–47] while $s$-wave fracton hydrodynamics shares several similarities. A striking difference of $p$-wave superfluids compared with Aristotelian fluids is that $p$-wave fracton superfluids do not admit equilibrium states with constant background spatial velocity, though $s$-wave fracton superfluids can. The structure of linear perturbations is also rather different as there are no sound modes at ideal order in $p$-wave fracton superfluids. In particular $p$-wave superfluids exhibit attenuated magnon-like dispersion relations with "magnon velocity" $\omega_p$ and attenuation $\Gamma_p$ (see Fig. 1)[4], similarly to what was reported in [25, 29, 49]. The U(1) fracton superfluid regime of $s$-wave fracton superfluids has dispersion relations that are also magnon-like with "magnon velocity" $\omega_M$ but with sub-diffussive attenuation

---

[4]By "magnon-like" dispersion relations we mean that the dispersion relation is quadratic in $k$ and we refer to "magnon velocity" as the coefficient appearing with $k^2$ in the dispersion relation though it does not have dimensions of velocity.

$\Gamma_M$ as well as a linear sound mode with velocity $\bar{v}_s$ and attenuation $\bar{\Gamma}_s$. Pinned $s$-wave fracton superfluids also have a similar mode structure with sound modes with velocity $\hat{v}$ and magnon modes with "velocity" $\omega_{\mathcal{B}^2}$, as depicted in Fig. 1. Furthermore, equilibrium states for U(1) fracton superfluids can have non-zero spatial velocities leading to dispersion relations in which each of two sound modes has different velocity. We also consider a new regime of *ultra-dense* fracton superfluids with large kinetic mass which can lead to sound modes with velocity $v_s^{\mathrm{UD}}$ (see Fig. 1). Introducing a parameter $\ell$ controlling the strength of the kinetic mass reveals a transition between an ultra-dense $p$-wave fracton superfluid regime and the true low energy regime of $p$-wave superfluids in which the kinetic mass is a second order transport coefficient.

This paper is structured as follows. In Section 2 we introduce the fracton algebra, gauge it to find the fracton geometry, and discuss the conservation laws. In Section 3 we show that fracton fluids cannot flow unless the fractonic symmetry is spontaneously broken. In Section 3.3 we discuss the different spontaneous symmetry breaking patterns of fractonic symmetries. In Section 4 we study $p$-wave superfluids and construct the corresponding equilibrium partition functions. We also discuss out of equilibrium dynamics by finding solutions to the adiabaticity equation and deriving dynamical equations for the Goldstone fields. In this section we also give a linearised analysis of the equations and obtain the different mode spectra for each phase of fracton superfluids. In Section 5 we perform a similar analysis for $s$-wave fracton superfluids. In Section 6 we conclude with some open questions and suggestions for future work. We also provide some appendices. In particular, in Appendix A we provide additional details on geometric variations, while in Appendix B we review ideal relativistic superfluids and study ideal Aristotelian superfluids. In Appendix C we study the consequences of applying another gradient expansion scheme to $s$-wave fracton superfluids.

**Note added**

While completing this work several papers appeared on ArXiv which overlap with parts of this work [25, 29] and, more recently, [50] in which different two versions of fracton superfluids are discussed. This paper also has overlap with the paper by Akash Jain, Kristan Jensen, Ruochuan Liu and Eric Mefford [48] that appeared on the same day and which we describe in more detail. Ref. [48] discusses in detail the structure of $p$-wave fracton superfluids beyond the ideal order analysis. As such our Section 4 overlaps with, and agrees with the results of [48], at ideal order. Section 4.3 in which we consider a special case where the mass density scales in a specific way with thermodynamic parameters is not considered in [48]. Ref. [48] briefly considers $s$-wave superfluids, in particular what we termed the U(1) regime and suggests an alternative derivative counting with respect to the one employed for $p$-wave superfluids. However Ref. [48] does not explore any of the two derivative counting schemes for $s$-wave superfluids in great detail. To wit, Ref. [48] does not

consider the regime which we termed pinned $s$-wave superfluids. As such only parts of Section 5.5 and parts of Appendix C.2 have some overlap with [48] as far as $s$-wave fracton superfluids are concerned.

## 2 Fracton symmetries and geometry

In this section, we discuss general aspects of fracton field theories and their algebraic properties. We begin by introducing dipole symmetries and the fracton algebra. We then gauge the fracton algebra to derive the geometry to which these field theories couple to. We discuss the spacetime Aristotelian geometry that results from that gauging as well as the fracton gauge fields. Using this fracton geometry we derive all conservation laws/Ward identities. Finally, we spell out the procedure for gauge fixing the dipole shift symmetry which is useful for certain applications.

### 2.1 Dipole symmetries and fracton algebra

We first consider field theories with a conserved dipole moment in flat space before generalising to curved spacetime. We assume the existence of a conserved current $J^\mu = (\rho, J^i)$, where $i = 1, \ldots, d$ is a spatial index, satisfying $\partial_\mu J^\mu = \dot\rho + \partial_i J^i = 0$, with $\rho$ the charge density and $J^i$ the charge flux. Given this conservation law we may construct a Noether charge $Q_{(0)} = \int d^d x \, \rho$ satisfying

$$\dot{Q}_{(0)} = -\int d^d x \, \partial_i J^i = 0 \,, \tag{2.1}$$

since $J^i$ is assumed to vanish at the boundary. The dipole moment of $\rho$ is given by

$$Q^i_{(2)} = \int d^d x \, x^i \rho \,, \tag{2.2}$$

which is conserved if
$$J^i = \partial_j J^{ij} \,, \tag{2.3}$$
for some symmetric current $J^{ij}$, since $\dot{Q}^i_{(2)} = -\int d^d x \, x^i \partial_j \partial_k J^{jk} = 0$. This implies that the charge conservation equation takes the form

$$\partial_\mu J^\mu = \dot\rho + \partial_i \partial_j J^{ij} = 0 \,. \tag{2.4}$$

The currents $J^\mu$ and $J^{ij}$ couple to the gauge fields $B_\mu$ and $A_{ij}$, respectively, where, like $J^{ij}$, the gauge field $A_{ij}$ is symmetric in its two spatial indices.

Suppose, then, that we have an action $S$ describing "fracton matter", abstractly denoted by $\varphi$, coupled to such gauge fields. Then the currents $J^\mu$ and $J^{ij}$ arise by variation with respect to the gauge fields $B_\mu$ and $A_{ij}$

$$\delta S[\varphi, B_\mu, A_{ij}] = \int d^{d+1}x \left[ E_\varphi \delta\varphi - J^\mu \delta B_\mu + J^{ij} \delta A_{ij} \right] \,, \tag{2.5}$$

where $E_\varphi$ is the equation of motion for $\varphi$. The conservation equations (2.3) and (2.4) should arise as Ward identities for the gauge transformations of the field $B_\mu$ and $A_{ij}$. The conservation of the charge current $\partial_\mu J^\mu = 0$ comes from endowing the gauge field $B_\mu$ with a standard U(1) gauge transformation

$$\delta_\sigma B_\mu = \partial_\mu \sigma \,. \tag{2.6}$$

In addition, to ensure that (2.3) holds, we must introduce a "dipole shift symmetry" $\Sigma_i$ such that

$$\delta_\Sigma B_i = -\Sigma_i \,, \qquad \delta_\Sigma A_{ij} = \partial_{(i} \Sigma_{j)} \,. \tag{2.7}$$

This is in fact a Stueckelberg symmetry and we can set $B_i = 0$ by appropriately choosing $\Sigma_i$, leaving behind the residual transformations

$$\Sigma_i = \partial_i \sigma \,. \tag{2.8}$$

As such, once fixing the Stueckelberg symmetry only the time component of the gauge field

$$\Phi := B_t \,, \tag{2.9}$$

remains as well as the symmetric gauge field $A_{ij}$ itself transforming as

$$\delta \Phi = \partial_t \sigma \,, \qquad \delta A_{ij} = \partial_{(i} \partial_{j)} \sigma \,. \tag{2.10}$$

In this context, we will refer to the transformation with parameter $\sigma$ as a dipole gauge transformation.

In addition to the U(1) and dipole symmetries, we demand invariance under spacetime translations and rotations. Together, these symmetries generate the *fracton algebra* $\mathfrak{f}$ with generators $\{H, P_i, L_{ij}, Q^{(0)}, Q_i^{(2)}\}$, where $H$ is the generator of time translations, $P_i$ of space translations, $L_{ij}$ of spatial rotations, $Q^{(0)}$ of U(1) gauge transformations, while $Q_i^{(2)}$ generates dipole transformations. The associated Poisson brackets are

$$\begin{aligned}
\{L_{ij}, L_{kl}\} &= -\delta_{ik} L_{jl} + \delta_{il} L_{jk} + \delta_{jk} L_{il} - \delta_{jl} L_{ik} \,, \\
\{L_{jk}, P_i\} &= -\delta_{ij} P_k + \delta_{ik} P_j \,, \\
\{L_{jk}, Q_i^{(2)}\} &= -\delta_{ij} Q_k^{(2)} + \delta_{ik} Q_j^{(2)} \,, \\
\{P_i, Q_j^{(2)}\} &= \delta_{ij} Q^{(0)} \,,
\end{aligned} \tag{2.11}$$

where we note, in particular, that the Hamiltonian $H$ is central. Ignoring the generators $Q^{(0)}$ and $Q_i^{(2)}$ leaves us with the generators of spacetime symmetries, in which case the algebra in Eq. (2.11) reduces to the Aristotelian algebra. We will be using the full algebra above as a starting point for writing fracton field theories in curved space.

## 2.2 Gauging the fracton algebra

To determine the geometry that these field theories couple to, we can "gauge" the fracton algebra $\mathfrak{f}$ in (2.11). This will simultaneously tell us about the background spacetime geometry that fracton field theories couple to and the gauge fields present in the low-energy description, as well as the transformation properties of these fields. "Gauging algebras" [51, 52] has a long and illustrious history in the context of non-Lorentzian geometry [53–56], while complementary approaches involving Lie algebra-valued connections have previously been considered for fractonic theories in [57–60].

As was demonstrated in [33], the gauging procedure is formally equivalent to building a Cartan geometry on a manifold $M$, namely, starting with a Lie algebra $\mathfrak{g}$, one must first specify a subalgebra $\mathfrak{h}$, known as the stabiliser, such that $\dim \mathfrak{g} - \dim \mathfrak{h} = \dim M$. In general, many different such subalgebras exist for the same $\mathfrak{g}$, which then lead to different geometric structures on $M$ [61]. For us, $\mathfrak{g} = \mathfrak{f}$ will be the fracton algebra with brackets given in (2.11). We then choose the isotropy subalgebra to be

$$\mathfrak{h} = \langle L_{ab}, Q_a^{(2)}, Q^{(0)} \rangle \,, \tag{2.12}$$

where $a, b = 1, \ldots, d$ are spatial "tangent space indices" that replace $i, j, \ldots$ in (2.11) in the general case. As it stands, the Klein pair $(\mathfrak{f}, \mathfrak{h})$ is non-effective, that is, $\mathfrak{h}$ contains the Abelian ideal $\mathfrak{i}$ consisting of the "internal symmetries"

$$\mathfrak{i} = \langle Q^{(0)}, Q_a^{(2)} \rangle \,. \tag{2.13}$$

Quotienting by this ideal allows us to construct the locally effective and geometrically realisable Klein pair $(\mathfrak{f}/\mathfrak{i}, \mathfrak{h}/\mathfrak{i})$, that is, the Klein pair of the $(d+1)$-dimensional static Aristotelian spacetime, which will be the homogeneous space on which the Cartan geometry is modelled. In this way, the dipole and U(1) symmetries that form $\mathfrak{i}$ will not be part of the spacetime symmetries, but rather play the rôle of internal symmetries as we alluded to above. The next step involves writing the Cartan connection $\mathscr{A} \in \Omega^1(M, \mathfrak{f})$, which is an $\mathfrak{f}$-valued 1-form defined on $M$. The Cartan connection has components

$$\mathscr{A}_\mu = H\tau_\mu + P_a e_\mu^a + \frac{1}{2}\omega_\mu^{ab}L_{ab} + Q^{(0)}B_\mu - Q_a^{(2)}\tilde{A}_\mu^a. \tag{2.14}$$

The gauge fields associated to the internal U(1) and dipole transformations, respectively, are related to the fracton gauge fields that we introduced in Section 2.1, and will be discussed in further detail below. Under $\mathfrak{h}$-gauge transformations, the Cartan connection transforms as

$$\delta\mathscr{A} = d\Lambda + [\mathscr{A}, \Lambda] \,, \tag{2.15}$$

where $\Lambda \in \Omega^0(M, \mathfrak{h})$ specifies the gauge transformation and which we can parameterise according to

$$\Lambda = \frac{1}{2}\lambda^{ab}L_{ab} - \Sigma^a Q_a^{(2)} + \sigma Q^{(0)} \,, \tag{2.16}$$

where $\lambda^{ab}$ is a local rotation, $\Sigma^a$ is a local "dipole shift", and $\sigma$ is a standard U(1) gauge transformation. Under these, the components of the gauge fields in the Cartan connection (2.14) transform according to

$$\delta\tau_\mu = 0\,, \qquad \delta e^a_\mu = e^b_\mu \lambda_b{}^a\,, \qquad \delta\omega^{ab}_\mu = \partial_\mu \lambda^{ab} + \omega^a_{\mu c}\lambda^{cb} + \omega^b_{\mu c}\lambda^{ac}\,,$$
$$\delta B_\mu = \partial_\mu \sigma - e^a_\mu \Sigma_a\,, \qquad \delta \tilde{A}^a_\mu = \partial_\mu \Sigma^a + \omega^a_{\mu b}\Sigma^b + \tilde{A}^b_\mu \lambda_b{}^a\,, \tag{2.17}$$

under $\mathfrak{h}$. Only the fracton gauge fields $B_\mu$ and $\tilde{A}^a_\mu$ transform under the internal symmetries $\mathfrak{i}$. In the next subsections we describe in detail the meaning of the various fields appearing in Eq. (2.17).

## 2.3 Aristotelian geometry

The $\mathfrak{g}/\mathfrak{h}$-valued gauge fields $\tau$ and $e^a$ appearing in Eq. (2.17) define an *Aristotelian structure* on $M$. They consist of a nowhere-vanishing 1-form $\tau$, the *clock form*, and the spatial vielbein (or coframe) $e^a$. This notion of Aristotelian geometry was first considered in the context of boost-agnostic fluid dynamics in [46] (see also [47] for a formulation that includes a charge current using Schwinger–Keldysh methods), and was shown to provide the correct background spacetime geometry to which fracton field theories couple [17, 18]). It is useful to introduce a dual basis (or frame) $(v^\mu, e^\mu_a)$ to $(\tau_\mu, e^a_\mu)$ such that

$$v^\mu \tau_\mu = -1\,, \qquad v^\mu e^a_\mu = e^\mu_a \tau_\mu = 0 \qquad e^a_\mu e^\mu_b = \delta^a_b\,. \tag{2.18}$$

The frame and coframe satisfy the completeness relation

$$h^\mu_\nu := e^\mu_a e^a_\nu = \delta^\mu_\nu + v^\mu \tau_\nu\,, \tag{2.19}$$

which also defines the spatial projector $h^\mu_\nu$. In what follows, we will primarily work with "rulers" $h_{\mu\nu}$ and $h^{\mu\nu}$, defined as

$$h_{\mu\nu} = \delta_{ab} e^a_\mu e^b_\nu\,, \qquad h^{\mu\nu} = \delta^{ab} e^\mu_a e^\nu_b\,. \tag{2.20}$$

Notably, the geometric data that make up the Aristotelian structure, viz., $\tau_\mu, v^\mu$ and $e^a_\mu, e^\mu_a$, have no boost tangent space transformations and transform only under diffeomorphisms and local spatial rotations. This stands in contrast to other (non)-Lorentzian geometries, where the vielbeins transform under an appropriate boost symmetry such as Lorentz boosts for Lorentzian geometry, Galilei (or Milne) boosts for Newton–Cartan geometry, and Carroll boosts for Carrollian geometry. Note furthermore that an Aristotelian geometry admits *many* invariants [62, 63] and it can be viewed as being simultaneously a Carrollian and Galilean geometry, admitting both Carrollian and Galilean invariants, but also both Lorentzian and Riemannian invariants. For example, the combination $g_{\mu\nu} = -\tau_\mu \tau_\nu + h_{\mu\nu}$ is a bona fide Lorentzian

metric, although being made up from more "fundamental" Aristotelian invariants, it is not a particularly useful object.

In [17], it was shown that there exists a special affine connection $\nabla$ compatible with the Aristotelian structure with the property that its torsion is given only in terms of the so-called intrinsic torsion of the Aristotelian geometry [63]. This connection is given by

$$\Gamma^\rho_{\mu\nu} = -v^\rho \partial_\mu \tau_\nu + \frac{1}{2} h^{\rho\sigma} \left( \partial_\mu h_{\nu\sigma} + \partial_\nu h_{\mu\sigma} - \partial_\sigma h_{\mu\nu} \right) - h^{\rho\sigma} \tau_\nu K_{\mu\sigma} , \tag{2.21}$$

where

$$K_{\mu\nu} = -\frac{1}{2} \pounds_v h_{\mu\nu} , \tag{2.22}$$

is symmetric and spatial, i.e., $v^\mu K_{\mu\nu} = 0$. Compatibility with the Aristotelian structure means that

$$\nabla_\mu \tau_\nu = \nabla_\mu v^\nu = \nabla_\mu h_{\nu\rho} = \nabla_\mu h^{\nu\rho} = \nabla_\mu h^\nu_\rho = 0 , \tag{2.23}$$

while the torsion of the affine connection is given by

$$T^\rho_{\mu\nu} := 2\Gamma^\rho_{[\mu\nu]} = -v^\rho \tau_{\mu\nu} + 2 h^{\rho\sigma} \tau_{[\mu} K_{\nu]\sigma} , \tag{2.24}$$

where $\tau_{\mu\nu} = 2\partial_{[\mu}\tau_{\nu]}$. As demonstrated in [63], $K_{\mu\nu}$ and $\tau_{\mu\nu}$ capture the intrinsic torsion of an Aristotelian geometry. We remark that $\tau_{\mu\nu}$ is the torsion of a Galilean geometry, while $K_{\mu\nu}$ is the intrinsic torsion of a Carrollian geometry. This connection is related to the rotation connection $\omega^{ab}_\mu$ via the *vielbein postulate*

$$\partial_\mu e^a_\nu - \Gamma^\rho_{\mu\nu} e^a_\rho + \omega^a_{\mu b} e^b_\nu = 0 . \tag{2.25}$$

To integrate functions on Aristotelian geometries we need to introduce the volume form defined as

$$e d^{d+1} x , \qquad \text{where} \qquad e = \det(\tau_\mu, e^a_\mu) . \tag{2.26}$$

We note that throughout this work we will, for simplicity, assume that the intrinsic torsion vanishes, which implies that (2.24) is zero. This concludes the discussion of the spacetime geometry obtained by gauging the fracton algebra but we give additional details in appendix A. We now turn our attention to the remaining gauge fields associated with the generators of $\mathfrak{i}$, which become the fracton gauge fields.

## 2.4   The fracton gauge fields

The curved space generalisations of the fracton gauge fields that feature in (2.5) consist of a gauge field $B_\mu$ and a symmetric spatial gauge field $A_{\mu\nu}$ satisfying $v^\mu A_{\mu\nu} = 0$. Their gauge transformations generalised to curved backgrounds are

$$\delta B_\mu = \partial_\mu \sigma - \Sigma_\mu , \qquad \delta A_{\mu\nu} = h^\rho_\mu h^\sigma_\nu \nabla_{(\rho} \Sigma_{\sigma)} , \tag{2.27}$$

where $\Sigma_\mu = \Sigma_a e_\mu^a$. While the gauge field $B_\mu$ is identical to that which appears in the Cartan connection (2.14), $A_{\mu\nu}$ is not, at first glance, equivalent to $\tilde{A}_\mu^a$. To make them equivalent, we must impose a *curvature constraint*, as we will now discuss.

Associated to the Cartan connection (2.14) is an $\mathfrak{f}$-valued 2-form curvature $\mathscr{F} \in \Omega^2(M, \mathfrak{f})$ defined by

$$\mathscr{F} = d\mathscr{A} + \frac{1}{2}[\mathscr{A}, \mathscr{A}] \,, \tag{2.28}$$

where the bracket hides a wedge. This curvature is not dipole invariant in the general case. However, the U(1) component *is* dipole invariant when the intrinsic torsion vanishes. The dipole curvature is given by

$$\left( \mathscr{F}\big|_{Q^{(0)}} \right)^a = -d^\nabla \tilde{A}^a \,, \tag{2.29}$$

where $d^\nabla \tilde{A}^a = d\tilde{A}^a + \omega^a{}_b \wedge \tilde{A}^b$, which transforms under dipole transformations as

$$\delta_\Sigma(-d^\nabla \tilde{A}^a) = -\Omega^{ab}\Sigma_b \,, \tag{2.30}$$

where $\Omega^{ab} = d\omega^{ab} + \omega^a{}_c \wedge \omega^{cb}$ is the $L$-component of $\mathscr{F}$. This is nothing but the usual statement that the field strength of the dipole gauge field transforms into a Riemann tensor [17, 18, 22]. The transformation of the U(1) curvature under dipole transformations is

$$\delta_\Sigma \left( \mathscr{F}\big|_{Q^{(0)}} \right) = -\Sigma_a \Theta^a \,, \tag{2.31}$$

where $\Theta^a = d^\nabla e^a$ is the torsion, i.e., the $P$-component of $\mathscr{F}$. We could therefore add a torsion term with the $B_\mu$-field to make the gauge-fixing condition dipole invariant, but then it would not be U(1) invariant.

Ignoring the $\mathfrak{i}$-valued part of $\mathscr{F}$, this curvature describes the deviation of the Cartan geometry from the model static Aristotelian spacetime with Klein pair $(\mathfrak{f}/\mathfrak{i}, \mathfrak{h}/\mathfrak{i})$. In what follows, we only need the $Q^{(0)}$-component of $\mathscr{F}$, so we will refrain from writing down the other components. This $Q^{(0)}$-component is given by

$$\mathscr{F}\big|_{Q^{(0)}} = dB + \tilde{A}^a \wedge e_a \,. \tag{2.32}$$

To make contact with the fracton gauge fields introduced in Section 2.1, we demand that the U(1) curvature $\mathscr{F}\big|_{Q^{(0)}}$ vanishes[5], leading to

$$\tilde{A}_\mu^a e_{\nu a} - \tilde{A}_\nu^b e_{\mu a} = -\partial_\mu B_\nu + \partial_\nu B_\mu =: F_{\nu\mu} \,, \tag{2.33}$$

where we defined the field strength $F = dB$ with components

$$F_{\mu\nu} = \partial_\mu B_\nu - \partial_\nu B_\mu \,. \tag{2.34}$$

---

[5]Note that this is a gauge-invariant statement since we asumme that $K_{\mu\nu} = \tau_{\mu\nu} = 0$.

While this field strength is invariant under U(1) gauge transformations, it transforms under dipole shifts as

$$\delta_\Sigma F_{\mu\nu} = -2\partial_{[\mu}\Sigma_{\nu]} \,. \tag{2.35}$$

The relation (2.33) reduces the number of independent components in $\tilde{A}_\mu^a$ from $d(d+1)$ to $d(d+1)/2$ which is the same number of components as in $A_{\mu\nu}$ and related to $\tilde{A}_\mu^a$ via

$$A_{\mu\nu} = \tilde{A}_\rho^a e_{a(\mu} h_{\nu)}^\rho \,. \tag{2.36}$$

This symmetric and spatial gauge field transforms as (using the vielbein postulate (2.25))

$$\delta A_{\mu\nu} = h_\mu^\rho h_\nu^\sigma \nabla_{(\rho}\Sigma_{\sigma)} \,, \tag{2.37}$$

under dipole transformations in agreement with (2.27). We can interpret the imposition of the curvature constraint as a particular choice of improvement terms for the dipole and U(1) currents. To see this, note that the relevant part of a generic variation involving $\tilde{A}_\mu^\nu = \tilde{A}_\mu^a e_a^\nu$

$$\delta S \supset \int d^{d+1}x\, e \left[ -J^\mu \delta B_\mu + D^\mu{}_\nu \delta \tilde{A}_\mu^\nu \right] \,. \tag{2.38}$$

The U(1) Ward identity is, as before,

$$\nabla_\mu J^\mu = 0 \,, \tag{2.39}$$

while the dipole shift Ward identity now reads

$$J^\nu h_\nu^\mu - \nabla_\nu D^{\nu\mu} = 0 \,, \tag{2.40}$$

where $D^{\nu\mu} = D^\nu{}_\rho h^{\rho\mu}$. In the absence of torsion, these Ward identities are invariant under the improvements[6]

$$\tilde{J}^\mu = J^\mu + \nabla_\nu \chi^{\nu\mu} \,, \qquad \tilde{D}^{\mu\nu} = D^{\mu\nu} + \chi^{\mu\rho} h_\rho^\nu \,, \tag{2.41}$$

where $\chi^{\mu\nu} = -\chi^{\nu\mu}$. In flat spacetime, the total dipole charge is now

$$d^i = \int d^d x \left( x^i J^t - D^{ti} \right) \,, \tag{2.42}$$

where $D^{ti}$ is the flat space version of $\tau_\mu D^\mu{}_\nu$ and captures the the"internal dipole density". We may, however, choose $\chi^{\mu\nu}$ above such that

$$\tilde{D}^{ti} = 0 \,, \qquad \tilde{D}^{[ij]} = 0 \,, \tag{2.43}$$

which *removes* the internal dipole density by removing $d(d+1)/2$ components of the dipole current. Imposing the curvature constraint, which removes the same number of components from $\tilde{A}_\mu^a$, corresponds to choosing the improvement above. In the rest of this work, we will choose this improvement and work with the gauge fields $B_\mu$ and $A_{\mu\nu}$.

---

[6]We thank an anonymous referee for pointing this out to us.

## 2.5 Currents and conservation laws

Given the Aristotelian geometry and the gauge fields discussed above, we can now couple fracton field theories to curved backgrounds. As noted in (2.27), the fracton gauge fields $B_\mu$ and $A_{\mu\nu}$ transform according to

$$\delta B_\mu = \pounds_\xi B_\mu + \partial_\mu \sigma - \Sigma_\mu \,, \qquad \delta A_{\mu\nu} = \pounds_\xi A_{\mu\nu} + h_\mu^\rho h_\nu^\sigma \nabla_{(\rho} \Sigma_{\sigma)} \,, \tag{2.44}$$

under fracton gauge transformations parameterised by $(\Sigma, \sigma)$ and infinitesimal diffeomorphisms $\xi^\mu$. Consider now the variation of an action functional $S[\tau_\mu, h_{\mu\nu}, B_\mu, A_{\mu\nu}]$ that depends on the Aristotelian structure and the fracton gauge fields

$$\delta S[\tau_\mu, h_{\mu\nu}, B_\mu, A_{\mu\nu}] = \int d^{d+1}x\, e \left[ -T^\mu \delta\tau_\mu + \frac{1}{2}\mathcal{T}^{\mu\nu} \delta h_{\mu\nu} - J^\mu \delta B_\mu + J^{\mu\nu} \delta A_{\mu\nu} \right] \,, \tag{2.45}$$

where $T^\mu$ is the energy current, $\mathcal{T}^{\mu\nu} = \mathcal{T}^{(\mu\nu)}$ the stress-momentum tensor, $J^\mu$ the U(1) current, and $J^{\mu\nu} = J^{(\mu\nu)}$ the dipole current. The stress-momentum tensor $\mathcal{T}^{\mu\nu}$ is only defined up to terms of the form $Xv^\mu v^\nu$, where $X$ is an arbitrary function since $v^\mu h_{\mu\nu} = 0$. The dipole current $J^{\mu\nu}$ is purely spatial, i.e.,[7]

$$J^{\mu\nu}\tau_\nu = 0\,. \tag{2.46}$$

The conservation laws (or Ward identities) corresponding to U(1) gauge transformations and dipole shifts, cf., (2.44), are

$$\nabla_\mu J^\mu = 0\,, \tag{2.47a}$$

$$J^\nu h_\nu^\mu - \nabla_\nu J^{\nu\mu} = 0\,, \tag{2.47b}$$

where we remind the reader that we assumed, as we do throughout, that the torsion (2.24) vanishes.

The energy current and the momentum-stress tensor are not invariant under dipole transformations $\Sigma_\mu$, as was observed in [18]. To determine their transformation properties under dipole transformations, we use the fact that $S$ is dipole invariant, which means that the second variation vanishes

$$
\begin{aligned}
0 &= \delta(\delta_\Sigma S) \\
&= \int d^{d+1}x\, e \bigg[ - \delta_\Sigma T^\mu \delta\tau_\mu + \frac{1}{2}\delta_\Sigma \mathcal{T}^{\mu\nu}\delta h_{\mu\nu} - \delta_\Sigma J^\mu \delta B_\mu + \delta_\Sigma J^{\mu\nu}\delta A_{\mu\nu} + \delta\tau_\mu J^{\mu\nu} v^\rho \nabla_\rho \Sigma_\nu \\
&\quad + \delta h_{\mu\nu}\left( -\tau_\sigma J^\sigma \Sigma_\rho h^{\rho\nu} v^\mu + h^{\rho\nu}\nabla_\sigma(J^{\mu\sigma}\Sigma_\rho) - \frac{1}{2}h^{\rho\sigma}\nabla_\sigma(J^{\mu\nu}\Sigma_\rho) \right) \bigg]\,,
\end{aligned}
\tag{2.48}
$$

---

[7]This can be implemented explicitly as a Ward identity by replacing $A_{\mu\nu}$ with an arbitrary symmetric tensor $\hat{A}_{\mu\nu}$ with a Stueckelberg symmetry of the form $\delta_\chi \hat{A}_{\mu\nu} = \chi_{(\mu}\tau_{\nu)}$. The associated Ward identity is precisely (2.46), and after imposing this off-shell, we can fix the Stueckelberg symmetry by setting $\hat{A}_{\mu\nu} = A_{\mu\nu}$. We will see an explicit example of this procedure in Section 4.

where we used the (torsion-free) variation of the Aristotelian connection (A.8) as well as the variations of the Aristotelian geometric objects in (A.3). Additionally, we used the Ward identities (2.47). Hence, we conclude that the U(1) and dipole currents are invariant

$$\delta_\Sigma J^\mu = \delta_\Sigma J^{\mu\nu} = 0 \,, \tag{2.49}$$

while the stresses transform as[8]

$$\delta_\Sigma T^\mu = J^{\mu\nu} v^\rho \nabla_\rho \Sigma_\nu \,,$$
$$\delta_\Sigma \mathcal{T}^{\mu\nu} = h^{\rho\sigma} \nabla_\sigma (J^{\mu\nu} \Sigma_\rho) + 2\tau_\sigma J^\sigma \Sigma_\rho h^{\rho(\mu} v^{\nu)} - 2h^{\rho(\mu} \nabla_\sigma (J^{\nu)\sigma} \Sigma_\rho) \,. \tag{2.50}$$

We remark that for simplicity we have not included additional matter fields and their equations of motion in Eq. (2.45), but in later sections we will add additional matter in the form of Goldstone fields. The variation in (2.45) is the curved space generalisation of (2.5).

The diffeomorphism Ward identity is

$$\nabla_\nu T^\nu{}_\mu + F_{\mu\nu} J^\nu - J^{\nu\rho} \nabla_\mu A_{\nu\rho} + 2\nabla_\nu (J^{\nu\rho} A_{\rho\mu}) = 0 \,, \tag{2.51}$$

where we defined the energy-momentum tensor

$$T^\mu{}_\nu = -T^\mu \tau_\nu + \mathcal{T}^{\mu\rho} h_{\rho\nu} \,, \tag{2.52}$$

and used the U(1) Ward identity (2.47a). Eq. (2.51) expresses that the energy-momentum tensor is not conserved due to the presence of Lorentz-type forces induced by non-vanishing gauge fields. Furthermore, although (2.51) does not *look* gauge invariant, it is in fact gauge invariant when taking into account the dipole transformation of the energy-momentum tensor that follows from (2.50), in particular

$$\delta_\Sigma T^\nu{}_\mu = -\tau_\mu J^{\nu\rho} v^\sigma \nabla_\sigma \Sigma_\rho + \nabla_\sigma (J^\nu{}_\mu \Sigma^\sigma) + \tau_\sigma J^\sigma v^\nu \Sigma_\mu - \nabla_\sigma (J^\sigma{}_\mu \Sigma^\nu) - \nabla_\sigma (J^{\nu\sigma} \Sigma_\mu) \,. \tag{2.53}$$

Using this, one may show that the diffeomorphism Ward identity (2.51) is invariant under dipole shifts upon repeatedly using the U(1) and dipole Ward identities in (2.47) as well as the Ricci identity (A.6).

When $S$ is a fluid functional, as we shall consider in later sections, these conservation laws become the hydrodynamic equations of motion together with additional Josephson relations for Goldstone fields which we will introduce. Below we discuss a particular gauge-fixing of the fracton gauge fields introduced above.

---

[8]These expressions differ from those in [18] due to our choice of Aristotelian connection.

## 2.6 Gauge fixing

For some of the symmetry-breaking patterns and regimes that we are interested in, in particular those in which the U(1) symmetry is spontaneously broken, it will be useful and most natural to gauge fix the dipole symmetry. As such we begin by decomposing $B_\mu$ into its temporal and spatial components according to

$$B_\mu = \Phi \tau_\mu + h_\mu^\nu \bar{B}_\nu \,. \tag{2.54}$$

This definition implies that $\bar{B}_\mu$ enjoys a shift Stueckelberg symmetry

$$\delta \bar{B}_\mu = f \tau_\mu \,, \tag{2.55}$$

where $f$ is an arbitrary function. This shift symmetry makes sure that the pair $(\Phi, \bar{B}_\mu)$ has the same number of components as $B_\mu$. As we will see below, the associated Ward identity ensures that the temporal component of the current that couples to $\bar{B}_\mu$ is zero, hence leading to the same number of components in the current as prior to the decomposition (2.54). The fields $\Phi$ and $\bar{B}_\mu$ in this decomposition inherit the following transformations from $B_\mu$

$$\delta \Phi = \pounds_\xi \Phi - v^\mu \partial_\mu \sigma \,, \qquad \delta \bar{B}_\mu = \pounds_\xi \bar{B}_\mu + h_\mu^\nu \partial_\nu \sigma - \Sigma_\mu \,. \tag{2.56}$$

Using these new variables we can express variations of the action (2.45) as

$$\delta S[\tau_\mu, h_{\mu\nu}, \Phi, \bar{B}^\mu, A_{\mu\nu}] = $$
$$\int d^{d+1}x \, e \left[ -\bar{T}^\mu \delta \tau_\mu + \frac{1}{2} \bar{\mathcal{T}}^{\mu\nu} \delta h_{\mu\nu} - \bar{J}^0 \delta \Phi - \bar{J}^\mu \delta \bar{B}_\mu + J^{\mu\nu} \delta A_{\mu\nu} \right] \,, \tag{2.57}$$

where the currents that appear in (2.57) are related to those that appear in (2.45) via

$$\bar{T}^\mu = T^\mu + \Phi J^\mu + h_\nu^\mu J^\nu v^\rho \bar{B}_\rho \,, \quad \bar{\mathcal{T}}^{\mu\nu} = \mathcal{T}^{\mu\nu} + 2\tau_\rho J^\rho \bar{B}_\sigma h^{\sigma(\mu} v^{\nu)} \,,$$
$$\bar{J}^0 = \tau_\mu J^\mu \,, \quad \bar{J}^\mu = h_\nu^\mu J^\nu \,, \tag{2.58}$$

while $J^{\mu\nu}$ remains unchanged. The Ward identity for the shift symmetry (2.55) derivable from (2.57) implies that

$$\bar{J}^\mu \tau_\mu = 0 \,, \tag{2.59}$$

ensuring that $(\bar{J}^0, \bar{J}^\mu)$ have the same number of components as $J^\mu$. The Ward identity (2.47b) associated to dipole shifts now takes the form

$$\bar{J}^\mu - h_\nu^\mu h_\sigma^\rho \nabla_\rho J^{\nu\sigma} = 0 \,, \tag{2.60}$$

while the U(1) Ward identity becomes

$$-v^\mu \partial_\mu \bar{J}^0 + \nabla_\mu \bar{J}^\mu = 0 \,, \tag{2.61}$$

where, in both cases, we used the shift Ward identity (2.59) to simplify the expressions. Both of these Ward identities reduce to (2.47a) and (2.47b), respectively, when using the relations in (2.58). The diffeomorphism Ward identity obtained from (2.57) reads

$$-\nabla_\nu \bar{T}^\nu{}_\mu - \bar{J}^0 \partial_\mu \Phi - \bar{J}^\nu \nabla_\mu \bar{B}_\nu + \nabla_\nu(\bar{J}^\nu \bar{B}_\mu) + J^{\nu\rho} \nabla_\mu A_{\nu\rho} - 2\nabla_\nu(A_{\mu\rho} J^{\nu\rho}) = 0 \,, \tag{2.62}$$

where the full energy-momentum tensor is given by $\bar{T}^\mu{}_\nu = -\bar{T}^\mu \tau_\nu + \bar{\mathcal{T}}^{\mu\rho} h_{\rho\nu}$. Using the inverse relations $\Phi = -v^\mu B_\mu$ and $\bar{B}_\mu = h^\nu_\mu B_\nu$, this Ward identity reduces to (2.51) when using the relations between the currents (2.58). Since the Ward identities (2.47) are true off-shell, we may impose (2.60) in (2.57) leading to the following action variation

$$\delta S[\tau_\mu, h_{\mu\nu}, \Phi, \bar{B}^\mu, A_{\mu\nu}] = \\ \int d^{d+1}x\, e \left[ -\bar{T}^\mu \delta\tau_\mu + \frac{1}{2}\bar{\mathcal{T}}^{\mu\nu} \delta h_{\mu\nu} - \bar{J}^0 \delta\Phi - h^\mu_\nu h^\rho_\sigma \nabla_\rho J^{\nu\sigma} \delta\bar{B}_\mu + J^{\mu\nu} \delta A_{\mu\nu} \right] . \tag{2.63}$$

As we already remarked in Section 2.1, the dipole shift $\Sigma_\mu$ is a Stueckelberg symmetry, which together with the shift symmetry (2.55), can be used to set $\bar{B}_\mu = 0$. Before we fix this particular gauge, however, it is useful to make yet another change of variables. We can define a dipole shift invariant symmetric two-tensor

$$\tilde{A}_{\mu\nu} = A_{\mu\nu} + h^\rho_{(\mu} h^\sigma_{\nu)} \nabla_\rho \bar{B}_\sigma \,, \tag{2.64}$$

which only transforms under diffeomorphisms and U(1) gauge transformations

$$\delta\tilde{A}_{\mu\nu} = \pounds_\xi \tilde{A}_{\mu\nu} + h^\rho_\mu h^\sigma_\nu \nabla_\rho \partial_\sigma \sigma \,. \tag{2.65}$$

The variation of the functional $S$ (2.63) now takes the form

$$\delta S[\tau_\mu, h_{\mu\nu}, \Phi, \bar{B}^\mu, \tilde{A}_{\mu\nu}] = \int d^{d+1}x\, e \left[ -\tilde{T}^\mu \delta\tau_\mu + \frac{1}{2}\tilde{\mathcal{T}}^{\mu\nu} \delta h_{\mu\nu} - \bar{J}^0 \delta\Phi + J^{\mu\nu} \delta\tilde{A}_{\mu\nu} \right] , \tag{2.66}$$

with $\tilde{T}^\mu = \bar{T}^\mu + \cdots$ and $\tilde{\mathcal{T}}^{\mu\nu} = \bar{\mathcal{T}}^{\mu\nu} + \cdots$, where the "$\cdots$" represents terms involving $\bar{B}_\mu$ that we refrain from writing since we are interested in gauge fixing $\bar{B}_\mu = 0$. The particular choice of gauge $\bar{B}_\mu = 0$ is achieved by choosing $f$ and $\Sigma_\mu$ such that both $\bar{B}_\mu = 0$ and $\delta\bar{B}_\mu = 0$. The latter condition states that the condition $\bar{B}_\mu = 0$ is stable under variations and imposes relations between the gauge parameters $(f, \Sigma_\mu)$ and $(\xi, \sigma)$. Since only $\bar{B}_\mu$ transforms under $f$ and $\Sigma_\mu$, the precise form of these relations

is not important. Fixing the Stueckelberg symmetry therefore leaves us with the gauge fields $\phi$ and $\tilde{A}_{\mu\nu}$ equipped with the gauge transformations

$$\delta\Phi = -v^\mu \partial_\mu \sigma\,, \qquad \delta\tilde{A}_{\mu\nu} = h_\mu^\rho h_\nu^\sigma \nabla_\rho \partial_\sigma \sigma\,. \qquad (2.67)$$

These are preciely the curved spacetime analogues of the flat space fields that feature in (2.10). As in Section 2.5, the stresses in (2.66) are not invariant under dipole gauge transformations. In particular, the second variation is

$$
\begin{aligned}
0 &= \delta(\delta_\sigma S) \\
&= \int d^{d+1}x\, e \bigg[ -\delta_\sigma \tilde{T}\delta_\mu + \frac{1}{2}\delta_\sigma \tilde{\mathcal{T}}^{\mu\nu}\delta h_{\mu\nu} - \delta_\sigma \bar{J}^0 \delta\Phi + \delta_\sigma J^{\mu\nu}\delta\tilde{A}_{\mu\nu} \\
&\qquad + \delta\tau_\mu \left( \bar{J}^0 v^\mu v^\nu \partial_\nu \sigma + 2J^{\mu\nu}v^\rho \nabla_\rho \partial_\nu \sigma \right) \\
&\qquad + \delta h_{\mu\nu} \left( h^{\rho(\mu}\nabla_\lambda (J^{\nu)\lambda}\partial_\rho \sigma) - \frac{1}{2}h^{\rho\lambda}\nabla_\lambda (J^{\mu\nu}\partial_\rho \sigma) - \bar{J}^0 h^{\rho(\mu}v^{\nu)}\partial_\rho \sigma \right) \bigg]\,.
\end{aligned}
$$
$$(2.68)$$

Just like in (2.48), we conclude that $\bar{J}^0$ and $J^{\mu\nu}$ are invariant under dipole gauge transformations, while the energy current and stress-momentum tensor now transform as

$$
\begin{aligned}
\delta_\sigma \tilde{T}^\mu &= \bar{J}^0 v^\mu v^\nu \partial_\nu \sigma + 2J^{\mu\nu}v^\rho \nabla_\rho \partial_\nu \sigma\,, \\
\delta_\sigma \tilde{\mathcal{T}}^{\mu\nu} &= h^{\rho\lambda}\nabla_\lambda (J^{\mu\nu}\partial_\rho \sigma) + 2\bar{J}^0 h^{\rho(\mu}v^{\nu)}\partial_\rho \sigma - 2h^{\rho(\mu}\nabla_\lambda (J^{\nu)\lambda}\partial_\rho \sigma)\,.
\end{aligned}
$$
$$(2.69)$$

The diffeomorphism Ward identity obtained from (2.66) is

$$-\nabla_\nu \bar{T}^\nu_{\ \mu} - \bar{J}^0 \partial_\mu \Phi + J^{\nu\rho}\nabla_\mu A_{\nu\rho} - 2\nabla_\nu (A_{\mu\rho}J^{\nu\rho}) = 0\,, \qquad (2.70)$$

while the dipole Ward identity becomes

$$-v^\mu \partial_\mu \bar{J}^0 + \nabla_\mu \nabla_\nu J^{\mu\nu} = 0\,. \qquad (2.71)$$

Combining this with (2.69), and using the Ricci identity (A.6), one may verify that the diffeomorphism Ward identity in (2.70) is invariant under dipole gauge transformations. This completes the discussion of the geometric aspects of fracton field theories. In the remainder of this paper, we will make use of these notions to understand equilibrium partition functions and hydrodynamic modes for fracton (super)fluids.

# 3 Fracton fluids do not flow

In this section we show, first using the fracton algebra and later using the equilibrium partition function in curved space, that global thermal states cannot have a non-zero flow velocity, suggesting that the fracton symmetry must be spontaneously broken.

The approach pursued here and the results obtained are in fact similar to those found in the context of equilibrium partition functions of higher-form hydrodynamics [34–38]. As such, at the end of this section we discuss the possible symmetry breaking patterns. In Sections 4 and 5 we will use these results to construct classes of fracton superfluids that *can* have a non-zero fluid velocity.

## 3.1 No-flow theorem

We begin with the observation that a conserved dipole density does not give rise to an additional hydrodynamic variable [26, 27], which means that the hydrodynamic variables are the same as those of a charged Aristotelian fluid as developed in [44, 46, 47]. In particular we can introduce a temperature $T$, chemical potential $\mu$ and spatial fluid velocity $u^i$ satisfying the Euler relation

$$\mathcal{E} = Ts - P + u^i p_i + \mu \rho \,, \tag{3.1}$$

where $P$ is the pressure, $s$ the entropy density, $\mathcal{E}$ the energy density, $p^i$ the momentum density and $\rho$, as in Section 2.1, the charge density. It is important to note that all thermodynamic functions and constitutive relations are functions of $T, \mu, u^i$ and their derivatives, in particular $P = P(T, \mu, \vec{u}^2)$, where $\vec{u}^2 = u^i u_i$, at ideal order in derivatives. In the grand canonical ensemble, the associated first law of thermodynamics reads

$$d\mathcal{E} = Tds + u^i dp_i + \mu d\rho \,. \tag{3.2}$$

Now, given that $p_i$ must be given in terms of $T, \mu, u^i$, at ideal order we can only write

$$p_i = m u^i \,, \tag{3.3}$$

where $m$ is the kinetic mass density (or momentum susceptibility). The Gibbs–Duhem relation obtained from $P = P(T, \mu, \vec{u}^2)$ is

$$dP = sdT + \frac{1}{2} m d\vec{u}^2 + \rho d\mu \,, \tag{3.4}$$

where $s = \partial P / \partial T$ is the entropy, $m = 2 \partial P / \partial \vec{u}^2$ the kinetic mass density that enters in (3.3), and $\rho = \partial P / \partial \mu$ the U(1) charge. A perfect Aristotelian fluid has the following energy-momentum tensor and U(1) current

$$
\begin{aligned}
T^0{}_0 &= -\mathcal{E} \,, & T^0{}_j &= p_j \,, & T^i{}_0 &= -(\mathcal{E} + P) u^i \,, \\
T^i{}_j &= P \delta^i_j + u^i p_j \,, & J^0 &= \rho \,, & J^i &= \rho u^i = \partial_j J^{ji} \,.
\end{aligned}
\tag{3.5}
$$

Since the Noether charges of a dipole invariant charged Aristotelian fluid realise the algebra (2.11), the bracket $\{P_i, Q_j^{(2)}\}$ implies that the momentum density transforms under a dipole transformation with (constant) parameter $\Sigma_i$ as

$$\delta_\Sigma p_i = \Sigma_i \rho \,. \tag{3.6}$$

Using Eq. (3.3), this implies that $\delta_\Sigma(mu^i) = \Sigma^i \rho$ and hence that either $m$ and/or $u^i$ transform under dipole transformations. On the one hand, $u^i$ is an Aristotelian spatial vector defined (in general) in terms of a background Killing vector that does not transform under $\Sigma_i$ but only under diffeomorphisms. Another way to see this is to notice that $J^\mu$ is invariant under dipole transformations, in which case (3.5) again implies that $u^i$ is invariant. On the other hand, given that $P$ must be invariant under dipole transformations, the kinetic mass density $m = 2\partial P/\partial \vec{u}^2$ is also invariant since $u^i$ is also invariant. This makes (3.6) a contradiction for nonzero $\rho$. Hence, the formulation cannot involve $p_i$, which is operationally equivalent to setting $u^i = 0$. Therefore, given (3.6), for $P$ to be invariant, and for consistency with Section 2.3, we must have that either

$$\rho = 0 \qquad \text{or} \qquad u^i = 0 \,. \tag{3.7}$$

This is in fact what is naively expected from a thermal bath of fracton particles. In particular, if the charge density is non-zero, the fracton fluid cannot move (all fluxes will be zero) and thus does not merit the moniker "fluid" and in which case the currents (3.5) become trivial. If, on the other hand, the charge density is zero, we are left with an ordinary neutral Aristotelian fluid without any dipole moment.

The conclusion (3.7) suggests that in fact a well defined theory of fracton fluids requires additional fields beyond $T, \mu, u^i$ that can realise (3.6). The authors of [23–25] avoided the choices (3.7) by working with a dipole invariant spatially symmetric two tensor $V_{ij} = \partial_{(i}(\rho^{-1}p_{j)})$ and modifying the first law (3.2) appropriately. We will show in Section 4.6 that this is indeed possible to do but requires the introduction of new hydrodynamic fields and interpreting it as a theory of fracton superfluidity. Below, we corroborate these results from the perspective of an equilibrium partition function.

## 3.2 The view from the hydrostatic partition function

Global equilibrium thermal states are defined via a partition function $\mathcal{Z}$ obtained from an Euclidean path integral [30, 46, 47, 64]

$$\mathcal{Z}[\tau_\mu, h_{\mu\nu}, B_\mu, A_{\mu\nu}] = \int \mathcal{D}\varphi \exp\left(-S^{\text{HS}}[\tau_\mu, h_{\mu\nu}, B_\mu, A_{\mu\nu}; \varphi]\right) , \tag{3.8}$$

over all possible configurations of dynamical fields $\varphi$. Here we have introduced the hydrostatic effective action $S^{\text{HS}}[\tau_\mu, h_{\mu\nu}, B_\mu, A_{\mu\nu}; \varphi]$ which is constructed from all symmetry invariants in the theory, in this case, invariant under diffeomorphisms, U(1) and dipole shift transformations. We focus on the case in which there are no additional low energy fields $\varphi$ and hence $S^{\text{HS}} \equiv S^{\text{HS}}[\tau_\mu, h_{\mu\nu}, B_\mu, A_{\mu\nu}]$. The invariance of $S^{\text{HS}}$ requires the existence of a set of symmetry parameters $K = (k^\mu, \sigma^K, \Sigma_\mu^K)$

satisfying the equilibrium conditions

$$\delta_K \tau_\mu = \pounds_k \tau_\mu = 0\,, \quad \delta_K h_{\mu\nu} = \pounds_k h_{\mu\nu} = 0\,,$$
$$\delta_K B_\mu = \pounds_k B_\mu + \partial_\mu \sigma^K - \Sigma_\mu^K = 0\,, \tag{3.9}$$
$$\delta_K A_{\mu\nu} = \pounds_k A_{\mu\nu} + h_\mu^\rho h_\nu^\sigma \nabla_{(\rho} \Sigma_{\sigma)}^K = 0\,.$$

From these conditions we deduce that $k^\mu$ is a Killing vector field and that the symmetry parameters transform under diffeomorphisms $\xi^\mu$ and gauge transformations with parameters $\sigma$ and $\Sigma_\mu$ according to

$$\delta k^\mu = \pounds_\xi k^\mu\,, \qquad \delta \Sigma_\mu^K = \pounds_\xi \Sigma_\mu^K - \pounds_k \Sigma_\mu\,, \qquad \delta \sigma^K = \pounds_\xi \sigma^K - \pounds_k \sigma\,. \tag{3.10}$$

These conditions are obtained by requiring that the last two equations in (3.9) are stable under variations, e.g., $\delta(\delta_K B_\mu) = 0$ and using, where possible, the equilibrium conditions themselves to simplify the expressions. Now, given the symmetry parameters $K$ and the fracton geometry as well as their transformations, we wish to characterise the possible structures that can enter in $S^{\mathrm{HS}}$ and which are invariant under all symmetries. It is straightforward to note that there are two scalar and one vector invariant, in particular

$$T = \frac{T_0}{k^\mu \tau_\mu}\,, \qquad \vec{u}^2 = h_{\mu\nu} u^\mu u^\nu\,, \qquad u^\mu = \frac{1}{k^\rho \tau_\rho} k^\mu\,. \tag{3.11}$$

Here $T$ is interpreted as the local fluid temperature, $T_0$ a constant global temperature, $u^\mu$ the fluid velocity and $\vec{u}^2$ the square of the spatial fluid velocity

$$\vec{u}^\mu := h_\nu^\mu u^\nu\,. \tag{3.12}$$

These are precisely the same invariants obtained for a neutral Aristotelian fluid [46, 47]. In particular we see that given Eq. (3.10) they only transform under diffeomorphisms as expected. The spatial component of the fluid velocity $u^a = e_\mu^a u^\mu$ (or $\vec{u}^\mu$) is precisely the fluid velocity introduced in[9] Eq. (3.1), which, as earlier advertised, does not transform under dipole shifts but only as a spacetime vector field.

For arbitrary $k^\mu$ it is not difficult to see that the invariants $T, \vec{u}^2, \vec{u}^\mu$ and gradients thereof are the only invariants that are possible to construct with the geometry at hand. In particular, the typical chemical potential associated to a charged Aristotelian fluid [47]

$$\mu = T(\sigma^K + k^\mu B_\mu)\,, \tag{3.13}$$

is not invariant under dipole transformations. However, if we restrict the Killing vector field to be purely temporal such that

$$k^\mu = -\frac{1}{T} v^\mu\,, \tag{3.14}$$

---

[9]In flat space, we may identify the tangent space indices $a, b, c, \cdots$ with the spatial indices $i, j, k, \cdots$.

or equivalently that $\vec{u}^\mu = 0$ (or $u^a = 0$), then the chemical potential (3.13) *is* invariant under all transformations in (3.10). Thus, we reach the same conclusion as in Eq. (3.7), that is, either the fluid cannot carry a charge density or the fluid cannot "flow". Below discuss the different symmetry breaking patterns that introduce low energy dynamical fields in (3.8), interpreted as Goldstone fields, allowing to define invariant chemical potentials.

## 3.3 Different symmetry breaking patterns

As discussed in Section 2.4 the symmetries associated with the fracton gauge fields consist of a U(1) symmetry and a dipole shift symmetry. These symmetries can be spontaneously broken leading to two symmetry breaking patterns:

- **$p$-wave fracton superfluids:** In this case the dipole symmetry is spontaneously broken but the U(1) symmetry remains intact, leading to a spatial vector Goldstone $\Psi_\mu$. This turns out to be the conceptually simpler case and hence the first case we present below. This is also the case addressed in [29] and we show that it is equivalent to [25].

- **$s$-wave fracton superfluids:** This class of superfluids has both the U(1) and the dipole symmetry spontaneously broken. Describing this phase requires introducing both Goldstone fields $\phi$ and $\Psi_\mu$. However, this pattern of symmetry breaking can be split into two different regimes depending on the strength $\ell'$ of the mass term proportional to $(B_\mu + \partial_\mu \phi - \Psi_\mu)$ that can be added to the hydrostatic effective action. For large $\ell' \sim \mathcal{O}(\partial^{-1})$, $\Psi_\mu$ can be integrated out and only the Goldstone $\phi$ features in the theory with both U(1) and dipole symmetries spontaneously broken [41]. We refer to this regime as a U(1) fracton superfluid and we discuss it in detail in Section 5.5. In the second regime, for small $\ell' \sim \mathcal{O}(1)$, the mass term plays a crucial role, and we call the resulting phase a *pinned* $s$-wave fracton superfluid phase akin to the pinned phases that appear in [38, 42, 65].[10]

In the next sections we discuss these symmetry breaking patters and regimes of $s$-wave fracton superfluids in detail, following common approaches to conventional superfluids. As an aid to the reader unfamiliar with such treatments, we have included Appendix B where we review relativistic superfluids, and where we also provide a formulation of U(1) superfluids coupled to Aristotelian geometry.

---

[10]This *pinned* phase is different than other pinned phases appearing in the context of charge density waves, crystals and higher-form symmetries studied in [38, 42, 65]. In these situations the pinning is induced by explicit symmetry breaking while in the context of $s$-wave fracton superfluids there is no explicit symmetry breaking. Nevertheless it is possible to define a "mass term" involving a specific combination of Goldstone fields.

## 4 The $p$-wave fracton superfluid

As mentioned in Section 3.3, the $p$-wave fracton superfluid phase spontaneously breaks the dipole symmetry while leaving the U(1) symmetry unbroken. In this case we can introduce a Goldstone field $\tilde{\Psi}_\mu$ that transforms under dipole gauge transformations and temporal Stueckelberg shifts according to

$$\delta\tilde{\Psi}_\mu = \chi\tau_\mu - \Sigma_\mu\,, \tag{4.1}$$

where we have ignored diffeomorphism transformations and we remind the reader that $v^\mu\Sigma_\mu = 0$. Only the spatial components of the Goldstone $\tilde{\Psi}_\mu$ are physical since $\chi$ is an arbitrary function parameterising the Stueckelberg shift symmetry and can be used to set the temporal component to zero.[11] For this reason, we define the spatial Goldstone field $\Psi_\mu$ such that

$$\Psi_\mu = h_\mu^\nu\tilde{\Psi}_\nu\,, \qquad \delta\Psi_\mu = -\Sigma_\mu\,. \tag{4.2}$$

The usage of $\tilde{\Psi}_\mu$ as a dynamical field in the action of fracton superfluids allows us to set appropriate constraints on currents but once action variations are performed, we will gauge fix the $\chi$-transformation to set $\tilde{\Psi}_\mu = \Psi_\mu$.

As in conventional U(1) superfluids [66] and in higher-form superfluids [35, 36], Goldstone fields only feature the hydrodynamic theory via gauge-invariant combinations. Hence, analogously, we can also introduce a dipole and U(1) gauge invariant superfluid "velocity" $\mathcal{A}_{\mu\nu}$ defined as

$$\mathcal{A}_{\mu\nu} = A_{\mu\nu} + h_\mu^\rho h_\nu^\sigma\nabla_{(\rho}\Psi_{\sigma)}\,, \tag{4.3}$$

which in this case is a symmetric and spatial two-tensor superfluid velocity. Unlike convential and higher-form superfluids, the theory of $p$-wave fracton superfluids can, in fact, also depend directly on $\Psi_\mu$ via the dipole invariant gauge field $\hat{B}^\mu$ that transforms as a typical U(1) gauge field, namely

$$\hat{B}_\mu = B_\mu - \Psi_\mu\,, \qquad \delta\hat{B}_\mu = \partial_\mu\sigma\,. \tag{4.4}$$

The two objects introduced here, specifically (4.3)–(4.4), form the basis of the theory of $p$-wave fracton superfluidity.

### 4.1 Conservation laws

Before explicitly constructing the equilibrium partition function it is useful to make some comments about the general form of the action functional that we are interested in and the resulting conservation laws. In the presence of Goldstone fields we consider

---

[11]Instead of introducing this Stueckelberg shift symmetry, we could have worked with a Goldstone field $\Psi_a$ where $a$ is a tangent space index. To avoid a cluttering of indices we refrain from doing so.

functionals of the form $S[\tau_\mu, h_{\mu\nu}, B_\mu, A_{\mu\nu}; \tilde{\Psi}_\mu]$ for which arbitrary variations can be parameterised according to

$$\delta S = \int d^{d+1}x\, e \left[ -T^\mu \delta\tau_\mu + \frac{1}{2}\mathcal{T}^{\mu\nu}\delta h_{\mu\nu} - J^\mu \delta B_\mu + J^{\mu\nu}\delta A_{\mu\nu} + K^\mu \delta\tilde{\Psi}_\mu \right]. \qquad (4.5)$$

We note that even in the case in which the functional $S$ only depends on $\tilde{\Psi}_\mu$ via (4.3) and (4.4), $T^\mu$ and $\mathcal{T}^{\mu\nu}$ may explicitly depend on $\tilde{\Psi}_\mu$. The response $K^\mu$ is the equation of motion for $\tilde{\Psi}_\mu$. Indeed for arbitrary variations of the Goldstone field we must have

$$K^\mu = 0. \qquad (4.6)$$

Given the form (4.5) we readily see that under the transformations (4.1) we obtain the Ward identities

$$J^\nu h^\mu_\nu - \nabla_\nu J^{\mu\nu} = K^\mu, \qquad K^\mu \tau_\mu = 0. \qquad (4.7)$$

The first of these equations is the dipole Ward identity which reduces to the correct form (2.47) once the Goldstone equation of motion (4.6) is imposed. The second Ward identity is associated with the Stueckelberg shift symmetry and states that only spatial components of $K^\mu$ are physical.

As the theories we are interested in depend explicitly on (4.3) and (4.4), it is useful and convenient to make a change of variables and parametrize variations according to

$$\delta S = \int d^{d+1}x\, e \left[ -T^\mu_\Psi \delta\tau_\mu + \frac{1}{2}\mathcal{T}^{\mu\nu}_\Psi \delta h_{\mu\nu} - J^\mu \delta\hat{B}_\mu + J^{\mu\nu}\delta\mathcal{A}_{\mu\nu} + \mathcal{K}^\mu \delta\tilde{\Psi}_\mu \right], \qquad (4.8)$$

where we have defined the modified response to $\tilde{\Psi}_\mu$ variations such that

$$\mathcal{K}^\mu = K^\mu - (J^\nu h^\mu_\nu - \nabla_\nu J^{\mu\nu}), \qquad (4.9)$$

and defined the dipole gauge invariant energy current $T^\mu_\Psi$ and stress $\mathcal{T}^{\mu\nu}_\Psi$. It is important to remark that the two variations (4.5) and (4.8) are equivalent to each other up to boundary terms, which do not play a role in this paper. In addition, it is necessary to keep in mind that in order to obtain the dynamical equation for $\tilde{\Psi}_\mu$ one must take into account that $\delta\hat{B}^\mu$ and $\delta\mathcal{A}_{\mu\nu}$ are not independent from variations of $\tilde{\Psi}_\mu$ and hence that the equation of motion is still (4.6). The variation (4.8) has various advantages. In particular, a dipole gauge transformation now leads to $\mathcal{K}^\mu = 0$ recovering the Ward identity (4.7) directly as the other variations are manifestly dipole gauge invariant. In addition, the energy current $T^\mu_\Psi$ and stress $\mathcal{T}^{\mu\nu}_\Psi$ as well as the remaining currents are manifestly dipole gauge invariant. As a consequence, under diffeomorphism transformations we obtain

$$\nabla_\nu T_\Psi{}^\nu{}_\mu + \hat{F}_{\mu\nu}J^\nu - J^{\nu\rho}\nabla_\mu\mathcal{A}_{\nu\rho} + 2\nabla_\nu(J^{\nu\rho}\mathcal{A}_{\rho\mu}) = 0, \qquad (4.10)$$

where we have imposed (4.6) and (4.7) and defined the field strength $\hat{F}_{\mu\nu} = 2\partial_{[\mu}\hat{B}_{\nu]}$. Here $T_{\Psi}{}^{\nu}{}_{\mu}$ is defined as in (2.52) using $T_{\Psi}^{\mu}$ and $\mathcal{T}_{\Psi}^{\mu\nu}$. Thus, from the point of view of (4.8), we obtain a conservation law (4.10) that is manifestly gauge invariant. Eq. (4.10) together with the dipole Ward identity (4.7) and the U(1) conservation law (2.47) obtained from (4.8) form the set of Ward identities associated to $p$-wave fracton superfluids.

## 4.2 Gradient expansion and equilibrium partition function

Given the conservation laws derived above and the dipole invariant structures (4.3)–(4.4), we may consistently construct an equilibrium partition function. However, as in every hydrodynamic theory we must first provide a gradient ordering. As in typical theories of superfluidity (see, e.g., [36]) one wishes for the effects of the superfluid velocity to be relevant at ideal order in the gradient expansion. As such, given the definition (4.3), we deduce that

$$\mathcal{A}_{\mu\nu} \sim A_{\mu\nu} \sim \mathcal{O}(1)\,, \qquad \Psi_{\mu} \sim \mathcal{O}(\partial^{-1})\,, \tag{4.11}$$

thus recovering the usual gradient order of a Goldstone field. The typical scale setting the strength of $\mathcal{O}(\partial)$ is the thermal length scale of the system. This gradient ordering implies that

$$h_{\nu}^{\mu}\hat{B}_{\mu} \sim \mathcal{O}(\partial^{-1})\,, \tag{4.12}$$

while for the time component we choose $v^{\mu}\hat{B}_{\mu} \sim \mathcal{O}(1)$ as the typical ordering of background gauge fields. Furthermore, as in typical hydrodynamic theories we choose the stresses and currents to be ideal order, that is[12]

$$T_{\Psi}^{\mu} \sim \mathcal{T}_{\Psi}^{\mu\nu} \sim J^{\mu}\tau_{\mu} \sim J^{\mu\nu} \sim O(1)\,, \tag{4.13}$$

though due to the nature of the dipole Ward identity $h^{\mu}{}_{\nu}J^{\nu} \sim \mathcal{O}(\partial)$. Given this ordering all terms in the conservation law (4.10) are of at least order $\mathcal{O}(\partial)$ except for the second term which contains the time derivative $v^{\mu}\partial_{\mu}\hat{B}_{\nu}$. The spatial component of the latter co-vector, using (4.12), is of $\mathcal{O}(1)$ leading to an inconsistent gradient scheme. To remedy this we must require that

$$v^{\mu}\partial_{\mu} \sim \mathcal{O}(\partial^2)\,. \tag{4.14}$$

The conservation law (4.10) thus implies that for ideal order fracton superfluids $\mathcal{O}(\partial^2)$ time derivatives are on the same footing as $\mathcal{O}(\partial)$ spatial derivatives. This anisotropic scaling between time and space derivatives is unusual in Aristotelian

---

[12]While this gradient expansion works for $p$-wave superfluids, it is possible to allow for a more general gradient scheme in which $p_{\nu} = \tau_{\mu}\mathcal{T}^{\mu\lambda}h_{\lambda\nu} \sim \mathcal{O}(\partial^{-1})$. This is allowed because $p_{\nu}$ enters the conservation laws and equations of motion with an appropriate number of derivatives. We will see examples of this when we work with gauge non-invariant stresses in $p$-wave fracton superfluids.

fluids, but indeed required for all the different symmetry breaking patterns of fracton hydrodynamics.[13]

Given this gradient ordering we can consider writing down the partition function (3.8), which requires classifying the possible non-vanishing ideal order scalars that can enter $S^{\mathrm{HS}}$ in equilibrium. As explained in Section 3.2, invariance of $S^{\mathrm{HS}}$ requires that there is a set of symmetry parameters $K = (k^\mu, \sigma^K, \Sigma_\mu^K)$ satisfying the conditions (3.9) in addition to

$$\delta_K \Psi_\mu = \pounds_\xi \Psi_\mu - \Sigma_\mu^K = 0 \,, \tag{4.15}$$

where, again, the symmetry parameters themselves transform as in (3.10). As for any Aristotelian fluid, we can find invariants corresponding to the temperature $T$, fluid velocity $u^\mu$ and the spatial velocity $\vec{u}^2$ (see Eq. (3.11)). The problem, as described in Section 3.2, is to find a chemical potential. It is straightforward to realise that given the presence of the Goldstone field, we can use (4.4) to construct the chemical potential $\mu_p$ according to

$$\mu_p = T\sigma^K + u^\mu \hat{B}_\mu \,, \tag{4.16}$$

which is manifestly invariant under all gauge symmetries. However, since we want this chemical potential to be of ideal order $\mu_p \sim \mathcal{O}(1)$ and given the gradient ordering (4.12), this is only possible if

$$\vec{u}^\mu = h_\nu^\mu u^\nu \sim \mathcal{O}(\partial) \,. \tag{4.17}$$

We thus recover the statement we made in Section 1, namely, a $p$-wave fracton superfluid can flow, but slowly. The gradient ordering (4.17) has the consequence that the scalar $\vec{u}^2$ introduced in (3.11) is not of ideal order but actually $\mathcal{O}(\partial^2)$ and hence does not feature, contrary to typical Aristotelian fluids, in the ideal order equilibrium partition function. In addition, there are many scalars that can be built from (4.3) and so, for simplicity, we will restrict to the only ideal order scalar that is linear in $\mathcal{A}_{\mu\nu}$, namely,[14]

$$\xi = h^{\mu\nu} \mathcal{A}_{\mu\nu} \,. \tag{4.18}$$

There are no other ideal order scalars that can be built from the symmetry parameters, gauge field $\hat{B}_\mu$ and linear in the superfluid velocity. As such that the hydrostatic effective action is of the form[15]

$$S^{\mathrm{HS}}[\tau_\mu, h_{\mu\nu}, B_\mu, A_{\mu\nu}; \Psi_\mu] = \int d^{d+1}x \, e P(T, \mu_p, \xi) + \mathcal{O}(\mathcal{A}_{\mu\nu}^2) + \mathcal{O}(\partial) \,. \tag{4.19}$$

---

[13]This particular gradient ordering agrees with the one implemented in [25].

[14]Previous works [25, 29] only consider quadratic terms in $\mathcal{A}_{\mu\nu}$ due to thermodynamic stability. For certain applications it may be that linear terms must vanish but even if such terms do vanish, as is the case for ordinary crystals [37, 67], their thermodynamic derivatives do not necessarily do so and are unaffected by stability arguments.

[15]Formally, as in [35, 36] we should also introduce an external source $K_{\mathrm{ext}}^\mu$ that couples to $\Psi_\mu$ but since for all practical purposes we set $K_{\mathrm{ext}}^\mu = 0$ we do not explicitly introduce it.

From (4.19) we can extract the equilibrium currents via (4.8) and obtain

$$
\begin{aligned}
T_\Psi^\mu &= P v^\mu - (sT + \rho\mu_p)v^\mu + (sT + \rho\mu_p)\vec{u}^\mu + \mathcal{O}(\partial)\,, \\
\mathcal{T}_\Psi^{\mu\nu} &= P h^{\mu\nu} - 2f_s \mathcal{A}_{\rho\sigma} h^{\rho(\mu} h^{\nu)\sigma} + \mathcal{O}(\partial)\,, \\
J^\mu &= \rho v^\mu - \rho \vec{u}^\mu + \mathcal{O}(\partial)\,, \\
J^{\mu\nu} &= f_s h^{\mu\nu} + \mathcal{O}(\partial)\,,
\end{aligned}
\tag{4.20}
$$

which are manifestly gauge invariant and where we have defined the entropy density $s$, charge density $\rho$, and superfluid density $f_s$ via the Gibbs–Duhem relation

$$
dP = s\,dT + \rho\,d\mu_p + f_s\,d\xi\,.
\tag{4.21}
$$

In addition, using the definition of energy density $\tau_\mu T_\Psi{}^\mu{}_\nu v^\nu = \mathcal{E}$, we find the following Euler relation

$$
\mathcal{E} + P = sT + \rho\mu_p\,.
\tag{4.22}
$$

In turn, defining the spatial momentum in the usual way, that is $p_\mu = \tau_\mu T^\mu{}_\lambda h^\lambda_\mu$, we readily see that $p_\mu = 0$. From (4.19) we can also obtain the equilibrium equation for the Goldstone field by explicitly varying with respect to $\Psi_\mu$, in particular

$$
K^\mu = J^\nu h^\mu_\nu - \nabla_\nu J^{\mu\nu} = -\rho \vec{u}^\mu - h^{\mu\nu}\nabla_\nu f_s = 0\,,
\tag{4.23}
$$

therefore explicitly setting the spatial fluid velocity to be of $\mathcal{O}(\partial)$. Note also that this equation of motion is precisely the dipole Ward identity. The currents at ideal order given in (4.20) are complete but we have noted, by adding $\mathcal{O}(\partial)$ to all expressions, that they will receive higher-order derivative corrections which can be hydrostatic or dissipative, as in [47]. However, we do not explore such corrections in this paper. Before studying the system out of equilibrium, we consider a special case of ultra-dense fracton superfluids.

## 4.3 Ultra-dense fracton superfluids

In our analysis above, we disregarded the scalar $\vec{u}^2 = h_{\mu\nu}u^\mu u^\nu$ since $\vec{u}^\mu \sim \mathcal{O}(\partial)$. However, It is possible to include it at ideal order, albeit artificially, by appropriately tuning the corresponding susceptibility, known as the kinetic mass density $m$. We thus augment the hydrostatic effective action (4.19) with this scalar, leading to

$$
S^{\mathrm{HS}}[\tau_\mu, h_{\mu\nu}, B^\mu, A_{\mu\nu}; \Psi_\mu] = \int d^{d+1}x\, e P(T, \mu_p, \xi, \vec{u}^2) + \mathcal{O}(\mathcal{A}_{\mu\nu}^2) + \mathcal{O}(\partial)\,.
\tag{4.24}
$$

To appropriately control the strength of the response to $\vec{u}^2$, we introduce a bookkeeping parameter $\ell$ of order $\ell \sim \mathcal{O}(\partial)$ such that

$$
dP = s\,dT + \rho\,d\mu_p + f_s\,d\xi + \frac{1}{2}\ell^{-2} m\,d\vec{u}^2\,.
\tag{4.25}
$$

Indeed the bookkeeping parameter implements the gradient ordering $\frac{1}{\ell^2}m \sim \mathcal{O}(\partial^{-2})$ (with $m \sim \mathcal{O}(1)$) leading to a very large kinetic mass density, hence the name *ultra-dense* fracton superfluids. This is analogous to the parameter that is introduced to control the strength of pinning of pseudo-Goldstone fields [38, 42, 65] and hence in general adding such term does not describe the true low energy behaviour of this phase, though probing the system with $\ell \gg k$ for wavenumber $k$ as we will see in Section 4.5 effectively recovers the low energy regime in which the kinetic mass is a second order transport coefficient. Nevertheless, it can be interesting to explore this construction in view of phenomenological models of fracton fluids. Controlling the strength of the kinetic mass allows to model systems whose kinetic mass can increase or decrease abruptly by dialling a thermodynamic parameter. As we will seen when looking at the modes, a very large kinetic mass can have drastic consequences in the spectrum.

When including the kinetic mass, the equilibrium currents (4.20) and Euler relation (4.22) are modified according to

$$
\begin{aligned}
T_\Psi^\mu &= Pv^\mu - (sT + \rho\mu_p + \ell^{-2}m\vec{u}^2)v^\mu + (sT + \rho\mu_p + \ell^{-2}m\vec{u}^2)\vec{u}^\mu + \mathcal{O}(\partial)\,, \\
\mathcal{T}_\Psi^{\mu\nu} &= Ph^{\mu\nu} - 2f_s\mathcal{A}_{\rho\sigma}h^{\rho(\mu}h^{\nu)\sigma} + \ell^{-2}m\vec{u}^\mu\vec{u}^\nu + 2\ell^{-2}m\vec{u}^{(\mu}v^{\nu)} + \mathcal{O}(\partial)\,, \quad (4.26) \\
\mathcal{E} + P &= sT + \mu_p\rho + \ell^{-2}m\vec{u}^2\,,
\end{aligned}
$$

while $J^\mu$ and $J^{\mu\nu}$ remain unchanged. Given these currents we can extract the momentum $p_\mu = \tau_\mu T_\Psi{}^\mu{}_\lambda h^\lambda_\mu = -\ell^{-2}m\vec{u}_\mu$. Thus for such ultra-dense fluids, the momentum is very large: it is of order $\mathcal{O}(\partial^{-1})$. As we will see below when comparing with earlier work, such scaling for the momentum appears to be typical for fracton superfluids. We will comment again on this particular case when looking at the spectrum of linear excitations. In order to do so, we first study the entropy production.

## 4.4 Entropy production

As in other theories of superfluidity, out of equilibrium the Josephson equation (or Goldstone equation of motion (4.23)) may acquire non-trivial corrections at ideal order (see, e.g., [68, 69]). To understand whether this is the case for fracton superfluids we study entropy production using the off-shell formulation of hydrodynamics [70] and postulate the existence of out of equilibrium parameters $\mathscr{B} = (\chi^\mu, \sigma^B, \Sigma_\mu^B)$ which in equilibrium revert to the values $K = (k^\mu, \sigma^K, \Sigma_\mu^K)$. These parameters act on the various fields as in the first equality in (3.9) and (4.15) with $K$ replaced by $\mathscr{B}$. The parameters $\mathscr{B}$ transform as in (3.10).

Given the parameters $\mathscr{B}$ and the action (4.8), the requirement that the second law of thermodynamics holds is embedded into the adiabaticity equation for the free energy current $N^\mu$, more precisely there must be a quadratic form $\Delta \geq 0$ such that

$$
\nabla_\mu N^\mu = -T_\Psi^\mu\delta_\mathscr{B}\tau_\mu + \frac{1}{2}\mathcal{T}_\Psi^{\mu\nu}\delta_\mathscr{B}h_{\mu\nu} - J^\mu\delta_\mathscr{B}\hat{B}_\mu + J^{\mu\nu}\delta_\mathscr{B}\mathcal{A}_{\mu\nu} + \mathcal{K}^\mu\delta_\mathscr{B}\Psi_\mu + \Delta\,. \quad (4.27)
$$

We can recast (4.27) as entropy production by defining the entropy current $S^\mu$ according to

$$N^\mu = S^\mu + \frac{1}{T}T^\mu_{\Psi\,\nu}u^\nu - \frac{\mu_p}{T}J^\mu + \frac{2}{T}u^\rho \mathcal{A}_{\rho\nu}J^{\mu\nu}\,. \tag{4.28}$$

At ideal order $N^\mu = Pu^\mu/T$ and one can explicitly check using (4.28) that $S^\mu = su^\mu$. With the definition (4.28) at hand (4.27) becomes the usual (off-shell) statement of the second law of thermodynamics

$$\begin{aligned}
\nabla_\mu S^\mu &- \frac{u^\mu}{T}(-\nabla_\nu T_\Psi{}^\nu{}_\mu - \hat{F}_{\mu\nu}J^\nu + J^{\nu\rho}\nabla_\mu \mathcal{A}_{\nu\rho} - 2\nabla_\nu(J^{\nu\rho}\mathcal{A}_{\rho\mu})) \\
&- \frac{\mu_p}{T}\nabla_\mu J^\mu - (K^\mu - (J^\nu h^\mu_\nu - \nabla_\nu J^{\mu\nu}))\delta_{\mathscr{B}}\Psi_\mu = \Delta \geq 0\,.
\end{aligned} \tag{4.29}$$

Given the gradient expansion we introduced above, the first three terms of (4.29) are of at least $\mathcal{O}(\partial)$ but the last term is not necessarily so. For (4.29) to hold we thus must require that

$$K^\mu = -a^{\mu\nu}\delta_{\mathscr{B}}\Psi_\nu + J^\nu h^\mu_\nu - \nabla_\nu J^{\mu\nu} + \mathcal{O}(\partial)\,, \tag{4.30}$$

for some positive definite $a^{\mu\nu}$. We note that the last two terms in (4.30) precisely correspond to the dipole Ward identity, which is equal to the Goldstone equation in equilibrium (4.23). Hence when the Goldstone equation of motion is satisfied $K^\mu = 0$ and when the dipole Ward identity holds we find that $\delta_{\mathscr{B}}\Psi_\mu = 0$ even out of equilibrium. At higher-orders, derivative corrections will appear in (4.30) but using the redefinition freedom $\Sigma^B_\mu \to \Sigma^B_\mu + \delta\Sigma^B_\mu$ for a gradient suppressed $\delta\Sigma^B_\mu$, we can set $\delta_{\mathscr{B}}\Psi_\mu = 0$ to all orders in the gradient expansion and hence remove $\Sigma^B_\mu$ entirely from the theory. Similar considerations hold also in the context of higher-form symmetries [35, 36]. This analysis implies that there are no new equations to take care of besides the conservation laws and Ward identities in $p$-wave fracton superfluids. We will now look at the linearised equations and the spectrum of excitations.

## 4.5 Linearised equations and modes

We now focus on linearised perturbations around particular equilibrium states. We restrict to flat spacetime backgrounds with $\tau_\mu = \delta^t_\mu$ and $h_{\mu\nu} = \delta_{ij}\delta^i_\mu \delta^j_\nu$ and gauge fields

$$B_\mu = 0\,, \qquad A_{ij} = \frac{\xi_0}{d}\delta_{ij}\,, \tag{4.31}$$

where $d$ is the number of spatial dimensions and $\xi_0$ is the equilibrium value of $\xi$. We also consider states with $u^\mu = (1,\vec{0})$ and constant $T = T_0, \mu_p = \mu_0$.[16] In turn this corresponds to the equilibrium values $\sigma^K = \mu_0/T_0$ and $\Psi_i = 0$. The subscript "0" emphasises that these are the equilibrium values of the parameters which solve the

---

[16]In Aristotelian fluids it is possible to have equilibrium configurations with non-vanishing equilibrium velocity $v^i_0$. These are difficult to realise for $p$-wave fracton superfluids but possible for $s$-wave superfluids as we will see in Section 5.5.3.

conservation law (4.10) and the Ward identities (2.47). We now consider fluctuations of the parameters around this class of equilibrium states such that

$$T = T_0 + \delta T \,, \qquad \mu_p = \mu_0 + \delta\mu_p \,, \qquad \Psi_i = \delta\Psi_i \,, \qquad u^\mu = (1, \delta v^i) \,. \qquad (4.32)$$

These fluctuations leads to fluctuations of the superfluid velocity and gauge field acording to

$$\mathcal{A}_{ij} = \frac{1}{d}\xi_0 \delta_{ij} + \partial_{(i}\delta\Psi_{j)} \,, \qquad \hat{B}_\mu = -\delta^i_\mu \delta\Psi_i \,, \qquad (4.33)$$

as well as to fluctuations of pressure $\delta P$, energy density $\delta\mathcal{E}$, charge density $\delta\rho$ and superfluid density $\delta f_s$. The conservation laws and Ward identities give $2d+2$ equations for the $2d+2$ unknowns $(\delta T, \delta\mu_p, \delta\Psi_i, \delta v^i)$ and read

$$\begin{aligned}
\partial_t \delta\mathcal{E} &= -(\mathcal{E}_0 + P_0)\partial_i \delta v^i - f_s^0 \partial_t \partial^j \delta\Psi_j \,, \\
\rho_0 \partial_t \delta\Psi_i &= -\partial_i \delta P + f_s^0 \partial_i \partial^j \delta\Psi_j \,, \\
\partial_t \delta\rho &= -\rho_0 \partial_i \delta v^i \,, \\
\ell\rho_0 \delta v^i &= -\partial^i \delta f_s \,,
\end{aligned} \qquad (4.34)$$

where in particular we note that $\xi_0$ has dropped out of the equations. In addition we note that the spatial component of the conservation law (4.10) provides dynamics for the Goldstone field $\Psi_i$. We now consider plane wave perturbations for $(\delta T, \delta\mu_p, \delta\Psi_i, \delta v^i)$ of the form $e^{i(-\omega t + k_i x^i)}$ where the wave vector has modulus $k$. Solving the resultant system of equations, noting that the last equation in (4.34) can be used to solve for $\delta v^i$, leads to two non-trivial magnon-like modes with attenuation

$$\omega = \pm\omega_p k^2 - \frac{i}{2}\Gamma_p k^2 + \mathcal{O}(k^3) \,. \qquad (4.35)$$

Here the velocity $\omega_p$ and the attenuation $\Gamma_p$ arise as the real and imaginary parts of the solution of a quadratic equation $C + iB\omega + A\omega^2 = 0$ for coefficients $A, B, C$ and whose general form is not particularly illuminating. In particular, we find that

$$\begin{aligned}
\omega_p^2 &= \frac{[(\partial\rho/\partial\mu)_0(\partial f_s/\partial\xi)_0 - (\partial\rho/\partial\xi)_0]\,(s_0^2 T_0(\partial\rho/\partial\mu)_0 + \rho_0^2 T_0(\partial s/\partial T)_0)}{T_0\rho_0^2(\partial s/\partial T)_0(\partial\rho/\partial\mu)_0^2} \\
&\quad + \mathcal{O}\left((\partial s/\partial\mu)_0\right) + \mathcal{O}\left((\partial s/\partial\xi)_0\right) \,, \\
\Gamma_p &= \frac{2(\partial\rho/\partial\xi)_0}{(\partial\rho/\partial\mu)_0} + \mathcal{O}\left((\partial s/\partial\mu)_0\right) + \mathcal{O}\left((\partial s/\partial\xi)_0\right) \,.
\end{aligned} \qquad (4.36)$$

We note that we must have $\Gamma_p \geq 0$ for stability. Magnon-like dispersion relations at ideal order are expected for $p$-wave superfluids as noted in [25, 29].[17] Typical

---

[17]The authors of [29] do not consider terms linear in the superfluid velocity $\mathcal{A}_{\mu\nu}$ but only quadratic terms. We have checked that by adding a scalar of the form $\Xi = h^{\mu\rho}h^{\nu\sigma}\mathcal{A}_{\mu\nu}\mathcal{A}_{\rho\sigma} - \frac{1}{d}h^{\mu\nu}h^{\rho\sigma}\mathcal{A}_{\mu\nu}\mathcal{A}_{\rho\sigma}$ to the hydrostatic effective action corresponding to the symmetric traceless part of the tensor $a^{ijkl}$ introduced in [29] reproduces their mode calculation in the appropriate regime of parameters and has the same form as in (4.35).

viscous corrections are expected to come with powers of $k^3$ or higher and hence of higher-order.

We can also perform the same analysis taking into account the ultra-dense corrections of Section 4.3. In this case we have to prescribe a relative ordering for the parameter $\ell$, which we do by following the procedure developed in [38, 42, 65]. In order to remain within the hydrodynamic regime we must require the wavenumber $k$ to satisfy $k \ll L_T^{-1}$ where $L_T$ is the thermal length scale. In addition, in order to stay within the large kinetic mass regime, we must require that $\ell \ll L_T^{-1}$. However, the relative scale between $\ell$ and $k$ can be different. If we focus on the regime $k \ll \ell \ll L_T^{-1}$, we expect to find small corrections to the spectrum of $p$-wave superfluids above. At ideal order, however, no such corrections appear and we again recover the modes the low energy spectrum of $p$-wave superfluids (4.35). On the other hand, in the regime $\ell \ll k \ll L_T^{-1}$, the large kinetic mass density leaves an imprint on the spectrum and instead we find a pair of sound modes of the form

$$\omega = \pm v_s^{\mathrm{UD}} k + \mathcal{O}(\ell^2) \,, \tag{4.37}$$

where $v_s^{\mathrm{UD}} = \tilde{\Omega}\ell$, with $\tilde{\Omega} \sim \mathcal{O}(1)$, is the sound speed and given in terms of a complicated function of the thermodynamic variables. As in [38, 42, 65], one may be tempted to interpret the change between these two regimes, say via an increase in temperature modelled by the variable strength of $\ell$, as a phase transition between the low energy regime of a $p$-wave fracton superfluid and a fracton fluid with very large kinetic mass. Before concluding our discussion of $p$-wave fracton superfluids, we now make an explicit comparison with [25].

## 4.6   Comparison with Głódkowski, Benítez and Surówka

We now compare the $p$-wave superfluid theory introduced above with the work of [25] using the fracton algebra (2.11) in flat spacetime and without external gauge fields. We will in particular show that the work of [25] can be interpreted as $p$-wave superfluidity.

In flat space $\tau_\mu = \delta^t_\mu$ and $h_{\mu\nu} = \delta_{ij}\delta^i_\mu\delta^j_\nu$, and with vanishing background fields $B_\mu = A_{\mu\nu} = 0$, the energy-momentum conservation equations (2.51) and the U(1) conservation equation (2.47) can be written as

$$\partial_t T^t{}_t = -\partial_i T^i{}_t \,, \qquad \partial_t T^t{}_j = -\partial_i T^i{}_j \,, \qquad \partial_t J^t = -\partial_i \partial_j J^{ij} \,, \tag{4.38}$$

where we have implemented the dipole Ward identity (2.47). In order to compare these equations with those of [25] we note that the authors of [25] have identified, to all orders in the gradient expansion, the various components of the currents according to

$$T^t{}_t = \epsilon \,, \qquad T^i{}_t = J^i_\epsilon \,, \qquad T^t{}_j = p_j \,, \qquad J^t = n \,, \tag{4.39}$$

where $\epsilon$ is the energy density $J_\epsilon^i$ the energy flux, $p_j$ the fluid momentum, and $n$ the charge density while $T^i{}_j$ and $J^{ij}$ remain as spatial stress and dipole current respectively. Introducing these identifications in (4.38) leads to Eq. (2) of [25] and therefore shows that Eq. (4.38) match those of [25]. The authors of [25] further postulate the following first law, Euler relation and Gibbs–Duhem relation

$$d\epsilon = Td\tilde{s} + \tilde{\mu}dn + U^{ij}dV_{ij}\,, \quad \epsilon + P = T\tilde{s} + \tilde{\mu}n\,, \quad dP = \tilde{s}dT + nd\tilde{\mu} - U^{ij}dV_{ij}\,, \quad (4.40)$$

where $\tilde{s}$ is the entropy density, $\tilde{\mu}$ the chemical potential, $V_{ij} = \partial_{(i}\left(n^{-1}p_{j)}\right)$ and $U^{ij}$ its thermodynamic conjugate.[18] At ideal order [25] finds the constitutive relations

$$J_\epsilon^i = (\epsilon + P)V^i - U^{ij}\partial_t\left(\frac{p_j}{n}\right)\,, \quad (4.41)$$

$$J^{ij} = -U^{ij}\,, \quad (4.42)$$

$$T^{ij} = P\delta^{ij} + V^i p^j + V^j p^i + \frac{p^k}{n}\partial_k U^{ij} + U^{ij}V^k{}_k\,, \quad (4.43)$$

where $V^i = -n^{-1}\partial_j U^{ij}$ and where we have ignored the dissipative coefficient $\alpha$ in Eq.(21) of [25] since we do not consider dissipative corrections in this paper. We now wish to show that all this can be recovered from $p$-wave superfluidity.

We begin by noting that [25] works with dipole non-invariant stresses since using the algebra (2.11), $\delta_\Sigma p_i \sim n\Sigma_i$ for a constant $\Sigma_i$. Therefore we must compute the full dipole non-invariant stresses using (4.5) which yields

$$
\begin{aligned}
T^\mu &= Pv^\mu - (sT + \rho\mu_p)v^\mu - f_s h^{\lambda\mu}v^\rho\partial_\rho\Psi_\lambda + (sT + \rho\mu_p)\vec{u}^\mu + \mathcal{O}(\partial)\,, \\
\mathcal{T}^{\mu\nu} &= Ph^{\mu\nu} + 2\rho v^{(\mu}h^{\nu)\rho}\Psi_\rho + 2h^{\rho(\mu}h^{\nu)\sigma}\Psi_\sigma\partial_\rho f_s - h^{\mu\nu}h^{\rho\sigma}\partial_\rho(f_s\Psi_\sigma) + \mathcal{O}(\partial)\,, \\
J^\mu &= \rho v^\mu - \rho\vec{u}^\mu + \mathcal{O}(\partial)\,, \\
J^{\mu\nu} &= f_s h^{\mu\nu} + \mathcal{O}(\partial)\,,
\end{aligned}
\quad (4.44)
$$

where we have specialised to flat spacetime. Comparing the dipole current in (4.44) with (4.41) we obtain, after sending $f_s \to -f_s$

$$U^{ij} = f_s\delta^{ij}\,, \qquad \rho = n\,, \qquad p_j = -n\Psi_j\,, \qquad V_{ij} = -\partial_{(i}\Psi_{j)}\,. \quad (4.45)$$

These identifications make $J^{ij}$ and $T^{ij}$ match the corresponding expressions in (4.41). In order to match the energy currents we use the dipole Ward identity (2.47) which gives

$$u^i = V^i + \mathcal{O}(\partial)\,, \quad (4.46)$$

where $u^i$ is the spatial fluid velocity. We remind the reader that the redefinition $f_s \to -f_s$ was implemented. Comparing the energy currents we deduce that

$$\epsilon = \mathcal{E}\,, \qquad sT + \rho\mu_p = \tilde{s}T + n\tilde{\mu}\,, \quad (4.47)$$

---

[18]To compare notation here with [25] we must set $\tilde{s} \to s$, $\tilde{\mu} \to \mu$ and $U_{ij} \to F_{ij}$.

where $\mathcal{E} = T^\mu{}_\nu v^\nu \tau_\mu$ for an exact match. Using the definition of the chemical potential in (4.16), we note that

$$\mu_p = \tilde{\mu} + \frac{V^i p_i}{n} \,, \tag{4.48}$$

and hence deduce that $\mu_p$ as defined in (4.16) is the effective chemical potential defined in [25] once the change $f_s \to -f_s$ is taken into account. Using now (4.47) we obtain that $\tilde{s} = (s + V^i p_i)/T$. This completes the proof that the fluid theory in [25] is precisely a type of $p$-wave fracton superfluidity at ideal order.

A few remarks are in order. The identification $p_j = -n\Psi_j$ agrees with the gradient ordering introduced in [25], namely that $p_j \sim \mathcal{O}(\partial^{-1})$. In this sense we can view the momentum $p_j$ as the Goldstone field of spontaneously broken dipole symmetry, as noted in [29]. However, since the momentum is not dipole invariant and vanishing from the point of view of the gauge invariant stresses (4.20), this point of view only holds in a specific choice of gauge. Secondly, the identification (4.46) justifies interpreting $V^i$ as the spatial fluid velocity. Thirdly, it is important to investigate the possible matching at higher orders in derivatives and when including dissipative effects. We begin by noting that in hydrodynamics out of equilibrium we have the redefinition freedom

$$T \to T + \delta T \,, \qquad u^i \to u^i + \delta u^i \,, \qquad \mu_p \to \mu_p + \delta \mu_p \,, \tag{4.49}$$

where $\delta T, \delta u^i, \delta \mu_p$ are terms that are of first or higher order in gradients, besides field redefinitions of the Goldstone field

$$\Psi_\mu \to \Psi_\mu + \delta \Psi_\mu \,. \tag{4.50}$$

We can use the field redefinitions of $T$ and $\mu_p$ in (4.49) to enforce that to all orders $T^0{}_0 = \epsilon$ and $J^0 = n$ thereby ensuring that the conservation laws for the energy density and charge density keep the same form as given by the identifications in (4.39) to all orders. We can also use the redefinition freedom associated with $\Psi_\mu$ (and hence with $p_j$ via (4.45)) given in (4.50) to keep the conservation law for the momentum the same as given by the identifications in (4.39) to all orders in derivatives. In turn, the redefinition freedom of $u^i$ can be used to remove any second or higher order terms from the dipole Ward identity (4.46). However, it cannot be used to remove first order terms that could potentially arise from dissipative or first-order corrections to $J^\mu$. We have not investigated the theory of $p$-wave superfluidity beyond ideal order and so we conclude that there are two possibilities. If there are no dissipative and first-order corrections to (4.46) then the theory in [25] is exactly the same as $p$-wave superfluidity. On the other hand, if there are dissipative and first-order corrections to (4.46), the energy currents would differ and the theory in [25] would be a restricted sector of $p$-wave superfluidity. We note that the case of ultra-dense fracton fluids discussed in Section 4.3 is an exception since it modifies the thermodynamics (4.40). This concludes our discussion of ideal fracton $p$-wave superfluids. In the next section we study $s$-wave fracton superfluids.

# 5   The s-wave fracton superfluid

In this section we consider the s-wave symmetry breaking pattern discussed in Section 3.3. In this scenario, both the U(1) symmetry and the dipole symmetry are spontaneously broken and lead to two different regimes which we refer to as U(1) fracton superfluids and pinned s-wave fracton superfluids, respectively. The analysis that follows in this case is very similar to what was discussed for p-wave fracton superfluids in Section 4 but with a few important differences.

When the dipole symmetry is spontaneously broken, as we discussed in Section 4, the Goldstone field $\tilde{\Psi}_\mu$ is part of the hydrodynamic theory and transforms as in (4.1). If, in addition, the U(1) symmetry is spontaneously broken another scalar Goldstone field $\phi$ must be introduced in the theory, which transforms under U(1) gauge transformations as

$$\delta\phi = -\sigma \,. \tag{5.1}$$

The presence of both Goldstone fields allows for the introduction of two "superfluid velocities", namely

$$\mathcal{B}_\mu = B_\mu + \partial_\mu\phi - \Psi_\mu \,, \tag{5.2a}$$

$$\mathcal{A}_{\mu\nu} = A_{\mu\nu} + h^\rho_\mu h^\sigma_\nu \nabla_{(\rho}\Psi_{\sigma)} \,, \tag{5.2b}$$

which are invariant under both U(1) and dipole transformations. The second of these is precisely the symmetric two-tensor superfluid velocity we introduced in (4.3), while the first one is a U(1) superfluid velocity from the point of view the Goldstone $\phi$, but akin to the misalignment tensor introduced in [38, 42, 65] from the point of view of $\Psi_\mu$.[19]

Given the two Goldstone fields we can parameterise a general variation of the effective action according to

$$
\begin{aligned}
\delta S &= \int d^{d+1}x\, e \left[ -T^\mu \delta\tau_\mu + \frac{1}{2}\mathcal{T}^{\mu\nu}\delta h_{\mu\nu} - J^\mu \delta B_\mu + J^{\mu\nu}\delta A_{\mu\nu} + K^\mu_\Psi \delta\tilde{\Psi}_\mu + K_\phi \delta\phi \right] \\
&= \int d^{d+1}x\, e \left[ -T^\mu_\Psi \delta\tau_\mu + \frac{1}{2}\mathcal{T}^{\mu\nu}_\Psi \delta h_{\mu\nu} - J^\mu \delta\mathcal{B}_\mu + J^{\mu\nu}\delta\mathcal{A}_{\mu\nu} + \mathcal{K}^\mu_\Psi \delta\tilde{\Psi}_\mu + \mathcal{K}_\phi \delta\phi \right] \,,
\end{aligned}
\tag{5.3}
$$

where in the second line we changed variables to (5.2) in order to work with gauge-invariant fields. We have also defined $K^\mu_\psi$ and $K_\phi$ as the responses to variations of the Goldstone fields, which when set to zero

$$K^\mu_\Psi = 0 \,, \qquad K_\phi = 0 \,, \tag{5.4}$$

---

[19]In the context of [38, 42, 65], $\mathcal{B}_\mu$ leads to massive Goldstone fields once considering the explicit dependence on $\mathcal{B}_\mu$ in the hydrostatic effective action. Here, however, as we will see, such terms do not play the same rôle.

are the dynamical equations for the Goldstone fields. These responses are related to $\mathcal{K}_\Psi^\mu$ and $\mathcal{K}_\phi$ according to

$$\mathcal{K}_\Psi^\mu = K_\Psi^\mu - \left(J^\nu h_\nu^\mu - \nabla_\nu J^{\mu\nu}\right), \qquad \mathcal{K}_\phi = K_\phi - \nabla_\mu J^\mu. \tag{5.5}$$

We note that the two variations in (5.3) differ from each other by boundary terms. The Ward identities associated with U(1) and dipole transformations give

$$\mathcal{K}_\Psi^\mu = 0, \qquad \mathcal{K}_\phi = 0, \tag{5.6}$$

while the Stueckelberg shift symmetry of $\tilde{\Psi}_\mu$ in (4.1) gives again $K_\Psi^\mu \tau_\mu = 0$. The conservation law associated to diffeomorphism transformations that arises from (5.3) reads

$$\nabla_\nu T_\Psi{}^\nu{}_\mu = -\mathcal{F}_{\mu\nu} J^\nu + J^{\nu\rho} \nabla_\mu \mathcal{A}_{\nu\rho} - 2\nabla_\nu(J^{\nu\rho} \mathcal{A}_{\rho\mu}), \tag{5.7}$$

where we have defined $\mathcal{F}_{\mu\nu} = \partial_\mu \mathcal{B}_\nu - \partial_\nu \mathcal{B}_\mu = \partial_\mu \hat{B}_\nu - \partial_\nu \hat{B}_\mu = \hat{F}_{\mu\nu}$. Given the geometry of $s$-wave fracton superfluids, we can now construct the equilibrium partition function for global thermal states.

## 5.1 Gradient expansion and equilibrium partition function

As in the case of $p$-wave fracton superfluids, we need to prescribe a gradient expansion. As we noted in Section 1 there are two possible gradient expansion schemes. In one scheme we can adopt the same ordering as for $p$-wave superfluids, namely $\mathcal{A}_{\mu\nu} \sim \mathcal{O}(1)$, and thereby deduce that $\Psi_\mu \sim \mathcal{O}(\partial^{-1})$ as in (4.11). We delegate an analysis of this scheme to Appendix C. Instead here we adopt an alternative gradient expansion proposed in [48]. This scheme sets $\mathcal{A}_{\mu\nu} \sim \mathcal{O}(\partial)$, leading to

$$\Psi_\mu \sim \mathcal{O}(1), \qquad h_\nu^\mu \mathcal{B}_\mu \sim \mathcal{O}(1), \qquad \phi \sim \mathcal{O}(\partial^{-1}). \tag{5.8}$$

We note in particular that according to this scheme, $\Psi_\mu$ scales differently than in the $p$-wave superfluid case. Due to $\mathcal{A}_{\mu\nu} \sim \mathcal{O}(\partial)$ we also have that $J^{\mu\nu} \sim \mathcal{O}(\partial^{-1})$ and $h_\nu^\mu J^\nu \sim \mathcal{O}(1)$. As a consequence, since we are demanding that $v^\mu \mathcal{B}_\mu \sim \mathcal{O}(1)$, both time and space derivatives scale in the usual way: $\partial_t \sim \mathcal{O}(\partial)$ and $\partial_i \sim \mathcal{O}(\partial)$. This gradient expansion is thus very similar to conventional Aristotelian superfluids (cf., Appendix B). We note that the scaling $\mathcal{A}_{\mu\nu} \sim \mathcal{O}(\partial)$ has the implication that $s$-wave flows must have small dipole superfluid velocity. In the context of the types of couplings to $\mathcal{A}_{\mu\nu}$ that we will consider below, this means that the dipole superfluid densiy must be treated perturbatively.

Given this gradient scheme, we may proceed with the construction of the equilibrium partition function using the invariant gauge fields in (5.2) as well as the symmetry parameters introduced in (3.9). The action of a symmetry transformation on $\Psi_\mu$ is given in (4.15), while it acts on $\phi$ according to

$$\delta_K \phi = \pounds_k \phi - \sigma^K = 0. \tag{5.9}$$

The chemical potential that can be built from the available set of fields is again given by (4.16), which we can now write as

$$\mu_p = T\sigma^K + u^\mu \hat{B}_\mu = T\sigma^K + u^\mu \mathcal{B}_\mu - u^\mu \partial_\mu \phi \,. \tag{5.10}$$

In equilibrium, it follows from (5.9) that $\mu_p = u^\mu \mathcal{B}_\mu$ and so $u^\mu \mathcal{B}_\mu$ which is an invariant in itself is not an independent scalar. We note, however, that since we require $\mu_p \sim \mathcal{O}(1)$ we deduce that $\vec{u}^\mu \sim \mathcal{O}(1)$. We see that this is rather different from the scaling (4.17) for $p$-wave superfluids as there is no longer a constraint on the spatial fluid velocity. This means that, like in ordinary Aristotelian fluids, we must include the scalar $\vec{u}^2$ and its associated response $m$: the kinetic mass density (see [46, 47] and Appendix B.2 for details).

In addition we can introduce two ideal order scalars that do not have a counterparts in $p$-wave fracton superfluids, namely

$$\nu = h^\mu_\nu u^\nu \mathcal{B}_\mu \,, \qquad \mathcal{B}^2 = h^{\mu\nu} \mathcal{B}_\mu \mathcal{B}_\nu \,. \tag{5.11}$$

Both of these scalars have counterparts in Aristotelian superfluids, though $\mathcal{B}^2$ acquires a slightly different interpretation since it also involves the vector Goldstone $\Psi_\mu$. In particular, it is analogous to the Goldstone mass/pinning terms explored in [38, 42, 65], but since it does not originate from explicit symmetry breaking it leads to different effects. There is an additional ideal order scalar, namely $v^\mu \mathcal{B}_\mu$ but it is not independent in equilibrium since $v^\mu \mathcal{B}_\mu = \nu - \mu_p$. We note that since $\mathcal{A}_{\mu\nu} \sim \mathcal{O}(\partial)$, there are no ideal order scalars that can be built from it.

With these scalars and our chosen gradient expansion we can write the hydrostatic effective action as

$$S^{\text{HS}}[\tau_\mu, h_{\mu\nu}, B_\mu, A_{\mu\nu}; \Psi_\mu, \phi] = \int d^{d+1}x \, e P(T, \mu_p, \vec{u}^2, \nu, \mathcal{B}^2) + \mathcal{O}(\partial) \,. \tag{5.12}$$

It will prove useful to introduce a new bookkeeping parameter $\ell'$, which we take to be of order $\mathcal{O}(1)$ to control the strength of the response to $\mathcal{B}^2$, as in [38, 42, 65]. Using the second variation in (5.3) we can extract the gauge invariant stresses and currents for the ideal $s$-wave fracton superfluid

$$T^\mu_\Psi = P v^\mu + (sT + \mu_p \rho + \nu N + m\vec{u}^2) u^\mu + (\mu_p - \nu) N \vec{u}^\mu + \ell'^2 (\mu_p - \nu) W h^{\sigma\mu} \mathcal{B}_\sigma + \mathcal{O}(\partial) \,,$$
$$\mathcal{T}^{\mu\nu}_\Psi = P h^{\mu\nu} + m\vec{u}^\mu \vec{u}^\nu + 2m\vec{u}^{(\mu} v^{\nu)} - 2N \mathcal{B}_\rho h^{\rho(\mu} v^{\nu)} - \ell'^2 W h^{\rho(\mu} h^{\nu)\sigma} \mathcal{B}_\rho \mathcal{B}_\sigma + \mathcal{O}(\partial)$$
$$J^\mu = \rho v^\mu - (\rho + N) \vec{u}^\mu - \ell'^2 W h^{\mu\nu} \mathcal{B}_\nu + \mathcal{O}(\partial) \,,$$
$$J^{\mu\nu} = \mathcal{O}(\partial) \,,$$
$$\tag{5.13}$$

where the entropy density $s$, charge density $\rho$, kinetic mass density $m$, U(1) superfluid density $N$, mass parameter $W$ and energy density $\mathcal{E}$ are defined via the Gibbs–Duhem and Euler relations, respectively

$$dP = s\,dT + \rho\,d\mu_p + N\,d\nu + \frac{1}{2}m\,d\vec{u}^2 + \frac{1}{2}\ell'^2 W\,d\mathcal{B}^2 \,, \quad \mathcal{E} + P = sT + \mu_p \rho + \nu N + m\vec{u}^2 \,. \tag{5.14}$$

We note that even though the stresses computed above are gauge invariant, they still give rise to a non-trivial spatial momentum $p_\mu = \tau_\nu \mathcal{T}_\Psi^{\nu\lambda} h_{\mu\lambda} = N\mathcal{B}_\nu h^\nu{}_\mu - m\vec{u}_\mu \sim \mathcal{O}(1)$ which, in particular, is of a different gradient order as for the $p$-wave case in (4.44). We note that the stresses and currents in (5.13) are precisely those of an ideal Aristotelian superfluid (B.32). Finally, we can extract both Goldstone equilibrium equations from (5.12) by explicit variation with respect to $\phi$ and $\Psi_\mu$. Specifically we find

$$
\begin{aligned}
K_{\phi,\mathrm{HS}} &= -\nabla_\mu \left( N\vec{u}^\mu \right) - \ell'^2 h^{\mu\nu} \nabla_\mu (W\mathcal{B}_\nu)\,, \\
K_{\Psi,\mathrm{HS}}^\mu &= -(\rho + N)\vec{u}^\mu - \ell'^2 W h^{\mu\nu} \mathcal{B}_\nu\,,
\end{aligned}
\tag{5.15}
$$

where we have added the subscript HS to $K_{\phi,\mathrm{HS}}$ and $K_{\Psi,\mathrm{HS}}^\mu$ to make manifest that these are the equations of motion in equilibrium. We note that $K_{\Psi,\mathrm{HS}}^\mu$ is equal to the dipole Ward identity (5.6) but $K_{\phi,\mathrm{HS}}$ is not the same as the U(1) Ward identity. We see that when the equation of motion for $\Psi_\mu$ is satisfied, the fluid velocity is given in terms of $\mathcal{B}_\nu$, which is similar to the $p$-wave case. In addition, from Eq. (5.6) we deduce that it is possible to have non-trivial spatial fluid velocities in equilibrium by appropriately tuning $h^\mu_\nu \mathcal{B}_\mu$.

Looking at the dipole Ward identity in (5.15), obtained by setting $K_{\Psi,\mathrm{HS}}^\mu = 0$, we can distinguish between two different regimes. If we allow $\ell'$ to be very large, $\ell' \sim \mathcal{O}(\partial^{-1})$, we may use this Ward identity to set $h^\mu_\nu \mathcal{B}_\mu = 0$ to all orders in the derivative expansion and thereby eliminate $\Psi_\mu$ from the theory as described in [41]. In particular, from $h^\mu_\nu \mathcal{B}_\mu = 0$ we deduce that $\Psi_\mu = h^\nu_\mu(B_\nu + \partial_\nu \phi)$[20], and we end up with a theory with just a singly scalar Goldstone field $\phi$. This theory can be described from the very beginning by just introducing one Goldstone field that spontaneously breaks the U(1) and the dipole symmetry, which we describe in detail in Section 5.5. On the other hand, if we take $\ell' \sim \mathcal{O}(1)$, the effects of $h^\mu_\nu \mathcal{B}_\mu$ become significant and define a pinned $s$-wave fracton superfluid. In fact in this regime the action (5.12) is very similar to the Aristotelian superfluid of Appendix B.2 but with a constraint on the fluid velocity given by $K_{\Psi,\mathrm{HS}}^\mu = 0$. In turn, fractonic effects are not visible at ideal order since $J^{\mu\nu} = 0$. We will investigate below possible signatures of a conserved dipole moment.

## 5.2 Higher-derivative hydrostatic corrections

For the $s$-wave fracton superfluid, the symmetric tensor gauge field only enters at first order in derivatives. There are several gauge invariant first-order scalars that can be build from the available structures, in particular.

$$
\xi = h^{\mu\nu} \mathcal{A}_{\mu\nu}\,, \qquad \Omega = h^{\mu\nu} \nabla_\mu \mathcal{B}_\nu\,, \qquad u^\mu u^\nu \mathcal{A}_{\mu\nu}\,, \qquad u^\mu \mathcal{B}^\nu \mathcal{A}_{\mu\nu}\,, \qquad \mathcal{B}^\mu \mathcal{B}^\nu \mathcal{A}_{\mu\nu}\,,
\tag{5.16}
$$

---

[20]It would be interesting to study this identity from the point of view of inverse Higgs constraints as in [71].

where $\mathcal{B}^\mu = h^{\mu\nu}\mathcal{B}_\nu$. There is one additional first-order scalar that may be constructed using $A_{\mu\nu}$ but is not independent, since $h^{\mu\nu}(A_{\mu\nu} + \nabla_\mu(B_\mu + \partial_\mu\phi)) = \xi + \Omega$. Furthermore, there are scalars associated with derivatives of the ideal order thermodynamic potentials such as $v^\mu\partial_\mu T$ that feature in Aristotelian fluids [46, 47]. Our purpose in what follows is not to make an exhaustive study but to highlight the possible fractonic effects due to potential higher-order derivative corrections.

We note that in $p$-wave superfluids $\mathcal{A}_{\mu\nu} \sim \mathcal{O}(1)$, making it a phenomenological assumption which scalars built from $\mathcal{A}_{\mu\nu}$ should be included at ideal order in the pressure. In contrast, $\mathcal{A}_{\mu\nu} \sim \mathcal{O}(\partial)$ within the gradient scheme for $s$-wave superfluids that we introduced above, and so only specific powers of $\mathcal{A}_{\mu\nu}$ and its derivatives can enter at a given order in the gradient expansion. As $\mathcal{A}_{\mu\nu}$ includes a background scale we must control its strength by introducing a bookkeeping parameter $\varepsilon \sim \mathcal{O}(\partial)$ analogous to $\ell'$ and treat the inclusion of the dipole superfluid velocity $\mathcal{A}_{\mu\nu}$ perturbatively. To understand the effect of higher-order corrections we consider a general set of corrections to the ideal order hydrostatic effective action involving $\xi$ which we parameterise as

$$\mathcal{P}(T, \mu_p, \vec{u}^2, \nu, \mathcal{B}^2, \xi) = P(T, \mu_p, \vec{u}^2, \nu, \mathcal{B}^2) + \sum_{I=1} \varepsilon^I \kappa_I(T, \mu_p, \vec{u}^2, \nu, \mathcal{B}^2)\xi^I, \qquad (5.17)$$

where $\kappa_I(T, \mu_p, \vec{u}^2, \nu, \mathcal{B}^2)$ are a set of higher-order transport coefficients that multiply the $I$'th power of $\xi$. Working with such a pressure we can expand our results to any given order in $\varepsilon$ if necessary. It is convenient to write a Gibbs-Duhem–type relation for $\mathcal{P}$, in particular

$$d\mathcal{P} = \tilde{s}dT + \tilde{\rho}d\mu_p + \tilde{N}d\nu + \frac{1}{2}\tilde{m}d\vec{u}^2 + \frac{1}{2}\ell'^2\tilde{W}d\mathcal{B}^2 + \tilde{f}_s d\xi, \qquad (5.18)$$

where $\tilde{f}_s$ is the superfluid density and the quantities $\tilde{s}, \tilde{\rho}, \tilde{N}, \tilde{m}, \tilde{W}$ are the conjugate variables to $T, \mu_p, \vec{u}^2, \mathcal{B}^2$ which can be written in terms of $P, \kappa_I, \xi$ using (5.17); for example, we have that

$$\tilde{s} = \frac{\partial P}{\partial T} + \sum_{I=1} \varepsilon^I \frac{\partial \kappa_I}{\partial T}\xi^I, \qquad \tilde{f}_s = \sum_{I=1} I\varepsilon^I \kappa_I \xi^{I-1}. \qquad (5.19)$$

Now using the second variation of (5.3) we can obtain the gauge invariant stresses that correct those of (5.13) by including the effects of $\xi$

$$T^\mu_\Psi = \mathcal{P}v^\mu + (\tilde{s}T + \mu_p\tilde{\rho} + \nu\tilde{N} + \tilde{m}\vec{u}^2)u^\mu + (\mu_p - \nu)\tilde{N}\vec{u}^\mu + \ell'^2(\mu_p - \nu)\tilde{W}h^{\sigma\mu}\mathcal{B}_\sigma + \mathcal{O}(\partial),$$

$$\mathcal{T}^{\mu\nu}_\Psi = \mathcal{P}h^{\mu\nu} + \tilde{m}\vec{u}^\mu\vec{u}^\nu + 2\tilde{m}\vec{u}^{(\mu}v^{\nu)} - 2\tilde{N}\mathcal{B}_\rho h^{\rho(\mu}v^{\nu)} - \ell'^2\tilde{W}h^{\rho(\mu}h^{\nu)\sigma}\mathcal{B}_\rho\mathcal{B}_\sigma$$
$$\qquad - 2\tilde{f}_s\mathcal{A}_{\rho\sigma}h^{\rho\mu}h^{\nu\sigma} + \mathcal{O}(\partial)$$

$$J^\mu = \tilde{\rho}v^\mu - (\tilde{\rho} + \tilde{N})\vec{u}^\mu - \ell'^2\tilde{W}h^{\mu\nu}\mathcal{B}_\nu + \mathcal{O}(\partial),$$

$$J^{\mu\nu} = \tilde{f}_s h^{\mu\nu} + \mathcal{O}(\partial),$$

$$(5.20)$$

where we have kept "$\mathcal{O}(\partial)$" in the expressions as a reminder that there are many other possible first-order corrections. We can also define an Euler-type relation involving the pressure $\mathcal{P}$ that takes the form

$$\tilde{\mathcal{E}} + \mathcal{P} = \tilde{s}T + \mu_p \tilde{\rho} + \nu \tilde{N} + \tilde{m}\vec{u}^2 \,, \tag{5.21}$$

where $\tilde{\mathcal{E}} = \tau_\mu T_\Psi^\mu$. Due to these modifications, the equations for the Goldstone fields (5.15) are modified to

$$\begin{aligned}
K_{\phi,\mathrm{HS}} &= \frac{\delta}{\delta\phi} \int d^{d+1}e\, \mathcal{P} = -\nabla_\mu \left( \tilde{N}\vec{u}^\mu \right) - \ell'^2 h^{\mu\nu} \nabla_\mu (\tilde{W}\mathcal{B}_\nu) \,, \\
K_{\Psi,\mathrm{HS}}^\mu &= \frac{\delta}{\delta\Psi_\mu} \int d^{d+1}e\, \mathcal{P} = -(\tilde{\rho} + \tilde{N})\vec{u}^\mu - h^{\mu\nu}\partial_\nu \tilde{f}_s - \ell'^2 \tilde{W} h^{\mu\nu}\mathcal{B}_\nu \,.
\end{aligned} \tag{5.22}$$

We see the appearance of the term $h^{\mu\nu}\partial_\nu \tilde{f}_s$ in $K_{\Psi,\mathrm{HS}}^\mu$ as the one for the $p$-wave superfluid. If the mass term were absent ($\ell' = 0$), this equation would imply that $\vec{u}^\mu = 0$ in equilibrium at ideal order since $\partial_\nu \tilde{f}_s \sim \mathcal{O}(\partial)$ and we would thus recover the scaling $\vec{u}^\mu \sim \mathcal{O}(\partial)$. However, in general this is not the case.

Lastly, using the first variation in (5.3) we can extract the gauge non-invariant stresses, yielding

$$\begin{aligned}
T^\mu &= \mathcal{P}v^\mu - (\tilde{s}T + \mu_p\tilde{\rho} + \nu\tilde{N} + \tilde{m}\vec{u}^2)u^\mu - \tilde{N}\vec{u}^\mu v^\rho B_\rho - f_s h^{\mu\nu} v^\rho \nabla_\rho \Psi_\nu \\
&\quad - \ell'^2 \tilde{W} v^\rho h^{\mu\sigma} \mathcal{B}_\rho \mathcal{B}_\sigma + \mathcal{O}(\partial) \,, \\
\mathcal{T}^{\mu\nu} &= \mathcal{P}h^{\mu\nu} - 2\tilde{f}_s \mathcal{A}_{\rho\sigma} h^{\rho\mu} h^{\nu\sigma} - 2\tilde{N}\mathcal{B}_\rho h^{\rho(\mu} v^{\nu)} - \ell'^2 W h^{\rho(\mu} h^{\nu)\sigma} \mathcal{B}_\rho \mathcal{B}_\sigma \\
&\quad + 2\tilde{\rho} v^{(\mu} h^{\nu)\rho} \Psi_\rho + (2h^{\rho(\mu} h^{\nu)\sigma} - h^{\mu\nu} h^{\rho\sigma})\nabla_\sigma(\tilde{f}_s \Psi_\rho) + \tilde{m}\vec{u}^\mu \vec{u}^\nu + 2\tilde{m}\vec{u}^{(\mu} v^{\nu)} + \mathcal{O}(\partial) \,.
\end{aligned} \tag{5.23}$$

The important quantity to be extracted from here is the momentum $p_\mu = \tau_\nu \mathcal{T}_\Psi^{\lambda\nu} h_{\mu\lambda} = \tilde{N}\mathcal{B}_\nu h_\mu^\nu - \tilde{\rho}\Psi_\mu - \tilde{m}\vec{u}_\mu \sim \mathcal{O}(1)$ and thus it is a combination of the invariant part, extracted above, and the $p$-wave contribution that we deduced in Section 4.6. We now turn our attention to a derivation of the Goldstone equations of motion from entropy production.

## 5.3 Entropy production

Our goal now is to understand the dynamics of the Goldstone fields $\phi$ and $\Psi_\mu$ out of equilibrium so that later we can derive the spectrum of linearised perturbations. To this end, we proceed as in Section 4.4. In particular, using the second variation of the action in (5.3) we can derive the off-shell adiabaticity equation

$$\nabla_\mu N^\mu = -T_\Psi^\mu \delta_{\mathscr{B}}\tau_\mu + \frac{1}{2}\mathcal{T}_\Psi^{\mu\nu}\delta_{\mathscr{B}}h_{\mu\nu} - J^\mu \delta_{\mathscr{B}}\mathcal{B}_\mu + J^{\mu\nu}\delta_{\mathscr{B}}\mathcal{A}_{\mu\nu} + \mathcal{K}_\phi \delta_{\mathscr{B}}\phi + \mathcal{K}_\Psi^\mu \delta_{\mathscr{B}}\Psi_\mu + \Delta \,, \tag{5.24}$$

where $\Delta \geq 0$ is a quadratic form that needs to be determined. This equation can be rewritten as the off-shell second law of thermodynamics by defining the entropy current via

$$N^\mu = S^\mu + \frac{1}{T}T_\Psi{}^\mu{}_\nu u^\nu - \frac{\mu_p}{T}J^\mu + \frac{2}{T}u^\rho \mathcal{A}_{\rho\nu}J^{\mu\nu}\,. \qquad (5.25)$$

At ideal order, $N^\mu = Pu^\mu/T$ and $S^\mu = su^\mu$. Including the derivative corrections we discussed above we have $N^\mu = \mathcal{P}u^\mu/T$ and $S^\mu = \tilde{s}u^\mu$. Using this definition of the entropy current we can recast the adiabaticity equation as a linear combination of the Ward identities

$$\nabla_\mu S^\mu - \frac{u^\mu}{T}(-\nabla_\nu T^\nu{}_\mu - \mathcal{F}_{\mu\nu}J^\nu + J^{\nu\rho}\nabla_\mu \mathcal{A}_{\nu\rho} - 2\nabla_\nu(J^{\nu\rho}\mathcal{A}_{\rho\mu}))$$
$$- \frac{u^\nu \mathcal{B}_\nu}{T}\nabla_\mu J^\mu - \mathcal{K}_\phi \delta_\mathscr{B}\phi - \mathcal{K}_\Psi^\mu \delta_\mathscr{B}\Psi_\mu = \Delta \geq 0\,. \qquad (5.26)$$

Given the gradient expansion introduced above for $s$-wave fracton superfluids, and the ideal order currents obtained in (5.13), we must have that

$$K_\phi = -\alpha \delta_\mathscr{B}\phi - \gamma^\mu \delta_\mathscr{B}\Psi_\mu + \nabla_\mu J^\mu - K_{\phi,\mathrm{HS}} + \mathcal{O}(\partial)\,,$$
$$K_\Psi^\mu = -\alpha^{\mu\nu}\delta_\mathscr{B}\Psi_\nu + \gamma^\mu \delta_\mathscr{B}\phi + (J^\nu h^\mu_\nu - \nabla_\nu J^{\mu\nu}) + \mathcal{O}(\partial)\,, \qquad (5.27)$$

for dissipative transport coefficients $\alpha \geq 0$ and $\alpha^{\mu\nu} \geq 0$ while $\gamma^\mu$ is unconstrained. Note that we added the hydrostatic correction to $K_\phi$ which appears when considering the equilibrium currents. When the equations of motion for the Goldstone fields are satisfied $K_\phi = K_\Psi^\mu = 0$ and the dipole and Ward identities imposed we deduce that

$$\delta_\mathscr{B}\Psi_\nu = \mathcal{O}(\partial)\,, \qquad (5.28)$$

where the terms of $\mathcal{O}(\partial)$ may arise due to dissipative corrections. Because $\Psi_\nu \sim \mathcal{O}(1)$ for $s$-wave fracton superfluids, we can only use the redefinition freedom associated with $\Sigma^B_\mu$ to remove potential contributions at order $\mathcal{O}(\partial^2)$ or higher. Nevertheless, Eq. (5.28) can still be used determine $\Sigma^B_\mu$ in terms of gradients of the remaining hydrodynamic variables and hence remove it from the spectrum. In addition, we find the Josephson equation for $\phi$, namely

$$v^\mu \mathcal{B}_\mu = \nu - \mu_p - \frac{T}{\alpha}K_{\phi,\mathrm{HS}} + \mathcal{O}(\partial)\,, \qquad (5.29)$$

where $K_{\phi,\mathrm{HS}}$ is given in (5.22). We have ignored corrections arising from other potential first order derivative terms. We will now make use of this equation to compute the spectrum of linear perturbations of $s$-wave fracton superfluids.

## 5.4 Linearised equations and modes

As in the case of $p$-wave fracton superfluids, we wish to find clear signatures of $s$-wave fracton superfluidity. We will accomplish this by computing linear perturbations in

flat spacetime $\tau_\mu = \delta^t_\mu$ and $h_{\mu\nu} = \delta_{ij}\delta^i_\mu\delta^j_\nu$ with background fields $B_\mu = 0$ and constant nonzero $A_{ij}$. In these backgrounds we consider equilibrium states with[21] $u^\mu = (1, \vec{0})$ and constant thermodynamic potentials $T_0, \mu_0, \xi_0, \tilde{\rho}_0, \tilde{s}_0, f_s^0, \tilde{N}_0, \tilde{W}_0$ together with $\nu = \mathcal{B}^2 = \Psi_\mu = 0$ and $\phi = \mu_0 t$ in equilibrium. We then perform general perturbations around this equilibrium state leading to

$$
\begin{aligned}
u^\mu &= (1, \delta v^i)\,, & T &= T_0 + \delta T\,, & \mu_p &= \mu_0 + \delta\mu_p\,, \\
\nu &= 0\,, & \xi &= \xi_0 + \partial_i\delta\Psi_i\,, & \mathcal{B}^2 &= 0\,, \\
\phi &= \mu_0 t + \delta\phi\,, & \Psi_i &= \delta\Psi_i\,, & \mathcal{B}_t &= \mu_0 + \partial_t\delta\phi\,, \\
\mathcal{B}_i &= \partial_i\delta\phi - \delta\Psi_i\,, & \mathcal{A}_{ij} &= \frac{\xi_0}{d} + \partial_{(i}\delta\Psi_{j)}\,,
\end{aligned} \tag{5.30}
$$

and we note that $\nu$ and $\mathcal{B}^2$ remain zero to linear order in perturbations. The equations of motions that provide dynamics to these perturbations are the conservation equations (5.7), the dipole and U(1) Ward identities (5.6) and the Josephson equation (5.29). We will from the very beginning consider the stresses and currents which include higher-derivative corrections (5.20). In this context, these equations become

$$
\begin{aligned}
\partial_t\delta\tilde{\mathcal{E}} &= -\ell'^2\tilde{W}_0\mu_0(\partial^i\partial_i\delta\phi - \partial^i\delta\Psi_i) - \partial_i\delta v^i(\tilde{s}_0 T_0 + \mu_0(\tilde{N}_0 + \tilde{\rho}_0)) - f_s^0\partial_t\partial^i\delta\Psi_i\,, \\
\partial_t\delta\Psi_i &= \frac{1}{(\tilde{N}_0 + \tilde{\rho}_0)}\left[\partial_i\delta\mathcal{P} + \tilde{N}_0\partial_t\partial_i\delta\phi - f_s^0\partial_i\partial^j\delta\Psi_j - \tilde{m}_0\partial_t\delta v_i\right]\,, \\
\partial_t\delta\tilde{\rho} &= -(\tilde{N}_0 + \tilde{\rho}_0)\partial_i\delta v^i - \ell'^2\tilde{W}_0(\partial^i\partial_i\delta\phi - \partial^i\delta\Psi_i)\,, \\
\partial^i\delta\tilde{f}_s &= -(\tilde{N}_0 + \tilde{\rho}_0)\delta v^i - \ell'^2\tilde{W}_0(\partial^i\delta\phi - \delta\Psi^i)\,, \\
\partial_t\delta\phi &= \delta\mu_p + \frac{T_0}{\alpha_0}\left[\tilde{N}_0\partial_i\delta v^i + \ell'^2\tilde{W}_0(\partial^i\partial_i\delta\phi - \partial^i\delta\Psi_i)\right]\,.
\end{aligned} \tag{5.31}
$$

We note that the second to last equation above can be used to eliminate $\delta v^i$ from the equations when $\ell' \sim \mathcal{O}(1)$. We also note that $\mathcal{B}^2$ appearing in (5.14) with "mass" $\tilde{W}_0$ in equilibrium does not feature in the second linearised equation above giving dynamics to $\Psi_i$. For this reason, the interpretation of this term is different than what was considered in [38, 42, 65] in which context such terms in the hydrostatic effective action would give rise to a mass term in the Josephson equation.

Before we analyse solutions to (5.31) in momentum space, we note that these equations depend on the control parameter $\ell'$, which determines the regime of the $s$-wave superfluid, and they depend implicitly on $\varepsilon$ via the energy density $\tilde{\mathcal{E}}$ and pressure $\mathcal{P}$ as well as through $f_s^0$. The relative scaling between $\ell'$, $\varepsilon$ and momentum $k$ can be chosen arbitrarily leading to many possible regimes. The requirement that the superfluid velocity $\mathcal{A}_{\mu\nu} \sim \mathcal{O}(\partial)$ is treated perturbatively implies that all results

---

[21]It is possible to realise equilibrium states with non-zero spatial velocity but we do not consider these here. We briefly study these effects in the U(1) fracton superfluid regime in Section 5.5.3.

should be expanded in powers of $\varepsilon$. With this mind below we study in detail the special case in which the mass term is absent ($\ell' = 0$), the "pinned" regime $\ell' \ll k$ and the U(1) regime $\ell' \gg k$. We will comment on the relative ordering between $\varepsilon$ and $k$ at the end of this section.

### Modes with vanishing "mass term" $\ell' = 0$

We begin by considering plane wave perturbations for the case when the "mass term" is absent, so that $\ell' = 0$. In the regime $\varepsilon \gg k$, we find a pair of modes of the form

$$\omega = \pm \omega_{\mathcal{B}^2} k^2 - \frac{i}{2} \Gamma_{\mathcal{B}^2} k^4 + \mathcal{O}(k^5) , \tag{5.32}$$

where

$$
\begin{aligned}
\omega_{\mathcal{B}^2}^2 = {} & \frac{1}{(\tilde{N}_0 + \tilde{\rho}_0)^2 \left[ (\partial \tilde{s}/\partial \mu_p)_0^2 - (\partial \tilde{s}/\partial T)_0 (\partial \tilde{\rho}/\partial \mu_p)_0 \right]} \\
& \times \left[ 2\tilde{s}_0 (\tilde{N}_0 + \tilde{\rho}_0) \left[ (\partial \tilde{s}/\partial \xi)_0 (\partial \tilde{\rho}/\partial \xi)_0 - (\partial \tilde{s}/\partial \mu_p)_0 (\partial \tilde{f}_s/\partial \xi)_0 \right] \right. \\
& \quad + (\tilde{N}_0 + \tilde{\rho}_0)^2 \left[ (\partial \tilde{s}/\partial T)_0 (\partial \tilde{f}_s/\partial \xi)_0 - (\partial \tilde{s}/\partial \xi)_0^2 \right] \\
& \quad \left. + \tilde{s}_0^2 \left[ (\partial \tilde{\rho}/\partial \mu_p)_0 (\partial \tilde{f}_s/\partial \xi)_0 - (\partial \tilde{\rho}/\partial \xi)_0^2 \right] \right] ,
\end{aligned}
\tag{5.33}
$$

and $\Gamma_{\mathcal{B}^2}$ is another complicated expression of the thermodynamic variables. In principle, this pair of modes can be magnon-like or diffusive depending on the sign of $\omega_{\mathcal{B}^2}^2$. However, for stability we require $\omega_{\mathcal{B}^2}^2 \geq 0$ and $\Gamma_{\mathcal{B}^2} \geq 0$ and hence the modes are magnon-like but with sub-diffusive behaviour.

The result presented in (5.32) and (5.33) should be understood as being implicitly expanded in small $\varepsilon$, which gives

$$\omega = \pm \varepsilon D_{\mathcal{B}^2} k^2 + \mathcal{O}(k^3) , \tag{5.34}$$

with

$$
\begin{aligned}
D_{\mathcal{B}^2}^2 = {} & \frac{1}{(\partial s/\partial \mu_p)_0^2 - (\partial \rho/\partial \mu_p)_0 (\partial s/\partial T)_0} \left[ 2\kappa_2^0 (\partial s/\partial T)_0 - (\partial \kappa_1/\partial T)_0^2 \right) \\
& + 2s_0 (N_0 + \rho_0)^{-1} ((\partial \kappa_1/\partial T)_0 (\partial \kappa_1/\partial \mu_p)_0 - 2\kappa_2^0 (\partial s/\partial \mu_p)_0) \\
& + s_0^2 (N_0 + \rho_0)^{-2} (2\kappa_2^0 (\partial \rho/\partial \mu_p)_0 - (\partial \kappa_1/\partial \mu_p)_0^2) \right] .
\end{aligned}
\tag{5.35}
$$

In the equation above the quantities without a tilde are calculated using $P$ and not $\mathcal{P}$ (cf. (5.17)). We furthermore note that the modes (5.33) do not appear at ideal order since at ideal order $\partial \tilde{s}/\partial \xi = \partial \tilde{f}_s/\partial \xi = 0$. However they do appear at subleading orders in derivatives when including $\kappa_i$ in (5.17). At higher-orders we will also find corrections of the form $\sim \varepsilon^2 k^2$.

**Modes in the "pinned" regime $\ell' \ll k$**

We now turn on the "mass term" by assuming $\ell' \neq 0$. As in the case of ultra-dense fracton superfluids of Section 4.5 we can consider two different regimes depending on the relative strength of $\ell'$ with respect to the wavenumber $k$. Here we consider the "pinned" (see footnote 10 for an explanation of this terminology) $s$-wave superfluid regime with $\ell' \ll k$ and later the U(1) regime with $\ell' \gg k$. In particular, in the "pinned" regime we consider the scalings $\ell' \ll k \ll \varepsilon \ll L_T^{-1}$, which leads to four modes of the form

$$\omega = \pm \hat{v} \ell' k + \mathcal{O}(\ell'^2), \tag{5.36a}$$

$$\omega = \pm \omega_{\mathcal{B}^2} k^2 + \mathcal{O}(\ell'^2), \tag{5.36b}$$

where $\omega_{\mathcal{B}^2}$ was defined in (5.33), while the square of the velocity of the linear mode is

$$\begin{aligned}
\hat{v}^2 = \tilde{s}_0^2 \tilde{W}_0 (\partial \tilde{f}_s/\partial \xi)_0 \Big/ \Big[ & (\tilde{N}_0 + \tilde{\rho}_0)^2 \big( (\partial \tilde{f}_s/\partial T)_0^2 - (\partial \tilde{s}/\partial T)_0 (\partial \tilde{f}_s/\partial \xi)_0 \big) \\
& - \tilde{s}_0 (\tilde{N}_0 + \tilde{\rho}_0) \big( (\partial \tilde{f}_s/\partial \mu_p)_0 (\partial \tilde{f}_s/\partial T)_0 - (\partial \tilde{s}/\partial \mu_p)_0 (\partial \tilde{f}_s/\partial \xi)_0 \big) \\
& + \tilde{s}_0^2 \big( (\partial \tilde{f}_s/\partial \mu_p)_0^2 - (\partial \tilde{\rho}/\partial \mu_p)_0 (\partial \tilde{f}_s/\partial \xi)_0 \big) \Big].
\end{aligned} \tag{5.37}$$

We note that the expression for $\hat{v}$ is affected by the higher order corrections provided by $\kappa_i$ in (5.17). In addition, depending on the sign $\omega_{\mathcal{B}^2}^2$, we find a pair of diffusive/"magnon" modes with attenuation/"magnon velocity" $\omega_{\mathcal{B}^2}$ at second order in the gradient expansion. At ideal order the mode structure resembles that of an Aristotelian fluid and is in stark contrast to $p$-wave fracton superfluids. Again, these modes should be expanded in $\mathcal{O}(\varepsilon)$, which gives (5.33) for the diffusive/"magnon" mode, while the expansion of the linear mode in (5.36a) gives

$$\omega = \pm \hat{v}_0 \ell' k + \mathcal{O}(\ell'^2, \varepsilon), \tag{5.38}$$

where

$$\begin{aligned}
\hat{v}_0^2 = 2 \kappa_2^0 s_0^2 W_0 \Big/ \Big[ & (N_0 + \rho_0)^2 \big( (\partial f_s/\partial T)_0^2 - 2 \kappa_2^0 (\partial s/\partial T)_0 \big) \\
& - s_0 (N_0 + \rho_0) \big( (\partial \kappa_1/\partial \mu_p)_0 (\partial \kappa_1/\partial T)_0 - 2 \kappa_2^0 (\partial \tilde{s}/\partial \mu_p)_0 \big) \\
& + s_0^2 \big( (\partial \kappa_1/\partial \mu_p)_0^2 - 2 \kappa_2^0 (\partial \rho/\partial \mu_p)_0 \big) \Big],
\end{aligned} \tag{5.39}$$

is the $\mathcal{O}(1)$ piece in the $\varepsilon$-expansion of $\hat{v}$ as defined in (5.37).

**Modes in the U(1) regime $\ell' \gg k$**

On the other hand, in the U(1) fracton superfluid regime in which $\ell'$ is large, i.e., $k \ll \ell' \sim \varepsilon \ll L_T^{-1}$, we find a pair of sound modes

$$\omega = \pm v_s k - \frac{i}{2} \Gamma_s k^2 + \mathcal{O}(k^3), \tag{5.40}$$

where the speed of sound $v_s$ and attenuation $\Gamma_s$ are given by

$$v_s^2 = \frac{\ell'^2 \tilde{W}_0 \tilde{s}_0^2 (\partial\tilde{\rho}/\partial\mu_p)_0}{((\tilde{N}_0 + \tilde{\rho}_0)^2 + \tilde{m}_0 \tilde{W}_0 \ell'^2)((\partial\tilde{s}/\partial\mu_p)_0^2 - (\partial\tilde{s}/\partial T)_0 (\partial\tilde{\rho}/\partial\mu_p)_0)} \,,$$

$$\Gamma_s = \ell'^2 \frac{\tilde{\rho}_0^2 T_0 \tilde{W}_0}{\alpha_0 ((\tilde{N}_0 + \tilde{\rho}_0)^2 + \tilde{m}_0 \tilde{W}_0 \ell'^2)} \,. \tag{5.41}$$

This pair of sound modes is present at ideal order in derivatives and hence constitutes the low energy spectrum. Expanding the sound mode in the small parameter $1/\ell' \ll 1$, we find that $v_s = \bar{v}_s + \mathcal{O}(1/\ell'^2)$ and $\Gamma_s = \bar{\Gamma}_s + \mathcal{O}(1/\ell'^2)$, where

$$\bar{v}_s^2 = \frac{\tilde{s}_0^2 (\partial\tilde{\rho}/\partial\mu_p)_0}{\tilde{m}_0 ((\partial\tilde{s}/\partial\mu_p)_0^2 - (\partial\tilde{s}/\partial T)_0 (\partial\tilde{\rho}/\partial\mu_p)_0)} \,, \qquad \bar{\Gamma}_s = \frac{T_0 \tilde{\rho}_0^2}{\alpha_0 \tilde{m}_0} \,. \tag{5.42}$$

As we alluded to above, we can integrate out $\Psi_\mu$ in the regime where $\ell' \gg k$, which only leaves the scalar Goldstone $\phi$. Indeed, we will demonstrate in Section 5.5 that we can recover the low energy spectrum (5.40) by just considering a theory with a single Goldstone field from the very beginning using the gauge-fixed formalism of Section 2.6. In the addition to the sound modes (5.40) we also find a pair of "magnon" modes

$$\omega = \pm\omega_M k^2 - \frac{i}{2}\Gamma_M k^4 + \mathcal{O}(1/\ell'^2) \,, \tag{5.43}$$

where we have expanded the non-trivial thermodynamic expression for the subdiffusive attenuation coefficient $\Gamma_M$ for large $\ell'$ to simplify the expressions and defined the "magnon velocity"[22]

$$\omega_M^2 = -\frac{(\partial\tilde{f}_s/\partial\xi)_0}{(\partial\tilde{\rho}/\partial\mu_p)_0} \,. \tag{5.44}$$

We have assumed that $\omega_M^2 > 0$ and $\Gamma_M \geq 0$ for stability. Looking at this expression, we see that $\omega_M$ is only non-trivial if we consider second order derivative corrections in (5.17). We also see that adding higher-order derivative corrections modifies the "magnon velocity" at this given order of $k$.

These modes should be expanded in powers of $\varepsilon$, which is equivalent to considering the regime where $\varepsilon \sim k$. In this regime we find the following pair of modes

$$\omega = \pm\bar{v}_s\big|_{\varepsilon=0} k \pm \varepsilon \tilde{v} k + \frac{i\bar{\Gamma}_s\big|_{\varepsilon=0}}{2} k^2 + \mathcal{O}(L^3, 1/\ell') \,,$$

$$\bar{v}_s\big|_{\varepsilon=0} = \frac{s_0 \sqrt{(\partial\rho/\partial\mu_p)_0}}{\sqrt{m_0[(\partial s/\partial\mu_p)_0^2 - (\partial\rho/\partial\mu_p)_0 (\partial s/\partial T)_0]}} \,, \tag{5.45}$$

where $\bar{v}_s$ and $\bar{\Gamma}_s$ are given in (5.42), while $\tilde{v} \sim \kappa_1$ is the $\mathcal{O}(\varepsilon)$ term in the expansion of $\bar{v}_s$, though we refrain from writing it explicitly. There is also a pair of

---

[22]The spectrum for $\ell' = 1$ is given by the sounds modes in (5.40) with $v_s^2$ and $\Gamma_s$ in (5.41) with $\ell' = 1$ and the magnon modes in (5.43).

magnon/diffusive modes of the form

$$\omega = \pm\omega_M\big|_{\varepsilon=0}\varepsilon k^2 + \mathcal{O}(\varepsilon^4, k^4, \dots), \qquad \left(\omega_M\big|_{\varepsilon=0}\right)^2 = -\frac{2\kappa_2^0}{(\partial\rho/\partial\mu_p)_0}, \qquad (5.46)$$

where $\omega_M$ is given in (5.44).

Finally, we note that we can interpret a large increase in $\ell'$, mediated for instance by an increase or decrease in temperature, as a phase transition between a U(1) fracton superfluid regime and the "pinned" $s$-wave superfluid regime (see Fig. 1). We will comment on this further in Section 6.

### Summary of the mode structure

Here we summarise the results we obtained on the modes of $s$-wave superfluids. The gradient expansion scheme we used required that the dipole superfluid velocity $\mathcal{A}_{\mu\nu}$ is gradient suppressed $\mathcal{A}_{\mu\nu} \sim \mathcal{O}(\partial)$. This means that $s$-wave flows are required to have small dipole superfluid velocity. To enforce it we introduced a small parameter $\varepsilon$ in front of the transport coefficients $\kappa_i$, whose relative scaling with respect to $k$ and $\ell'$ can differ. We noted that assuming $\varepsilon \sim k$ or $\varepsilon \gg k$ gives rise to the same results for the modes in the U(1) regime, where for $\ell' \gg k$ with $\varepsilon \sim k$ we found a mode structure of the form

$$\begin{aligned}
\omega &= \pm\left(V_{(0,1)} + \varepsilon V_{(1,2)} + \varepsilon^2 V_{(2,3)} + \dots\right)k + \left(R_{(0,2)} + \varepsilon R_{(1,3)} + \varepsilon^2 R_{(2,4)} + \dots\right)k^2 \\
&\quad + \mathcal{O}(k^3, \varepsilon^3, 1/\ell'), \\
\omega &= \pm\left(T_{(2,2)}\varepsilon + T_{(3,4)}\varepsilon^2 + \dots\right)k^2 + \mathcal{O}(k^3, \varepsilon^3, 1/\ell'),
\end{aligned} \qquad (5.47)$$

where $V, R, T$ are coefficients that depend on the thermodynamic variables and the subscript $(X, Y)$ indicates at which derivative order $X$ they appear and at which order $Y$ in a momentum expansion $k$ they appear. For instance $V_{(0,1)}$ appears at ideal order in derivatives and at linear order in momentum while $V_{(1,2)}$ appears at first order in derivatives and at quadratic order in momentum. Truncating the modes (5.47) to quadratic order in momentum requires only taking into account the coefficients $V_{(0,1)}, V_{(1,2)}, R_{(0,2)}$. We see from (5.47) that at each coefficient for a given power of $k$ receives corrections due to $\varepsilon$. This is not unexpected since if we were dealing with a typical U(1) superfluid and treated the superfluid density perturbatively, the same type of corrections would appear.[23] The modes in the pinned regime $\ell' \ll k$ follow a similar structure to (5.47), though the corresponding coefficients $V, R, T$ come with an additional index that keeps track of the powers of $\ell'$. Finally, it is also possible to consider the regime in which $\varepsilon \ll k$. In this case, depending on the exact scaling

---

[23]This would amount to expanding the modes in (B.24) and (B.25) in a small $f$ expansion. We also note that the same type of corrections would appear when turning on a weak background source such as a background electric or magnetic field and in the latter case giving rise to cyclotron motion [72].

between $\varepsilon$ and $k$ various coefficients could be pushed to higher-orders in $k$. For instance, if we scale $\varepsilon \sim k^2$ then the coefficient $V_{(1,2)}$ would change its order in the momentum expansion according to $V_{(1,2)} \rightarrow V_{(1,3)}$.

As we mentioned in Section 1 we discuss a different gradient expansion for $s$-wave superfluids in Appendix C. It is instructive to compare the structure of (5.47) with the structure we get using this alternative scheme for $\ell' \gg k$. The major difference is that the linear term in $k$ in the first mode in (5.47) only appears at second order in derivatives so that $\omega = \pm V_{(2,1)} k + R_{(0,2)} k^2 + ...$ due to a second order gradient correction to the hydrostatic partition function proportional to $m\vec{u}^2$ where $m$ is the mass density while the term quadratic in $k^2$ already appears at ideal order in gradients. This suggests that higher-order corrections (second order gradient corrections) are introducing novel effects in the infra-red (low $k$ regime, in particlar at linear order in $k$) and therefore we adopted the other gradient scheme.

This concludes our discussion of the modes in $s$-wave superfluids. Below we discuss the U(1) regime of the $s$-wave fracton superfluid from a different point of view.

## 5.5 The U(1) regime of the $s$-wave fracton superfluid

In this section we discuss in more detail the U(1) fracton superfluid regime of the $s$-wave fracton superfluid introduced in the previous section. This is the regime in which $\ell' \sim \mathcal{O}(\partial^{-1})$ and the vector Goldstone $\Psi_\mu$ can be integrated out leading to a theory that only contains $\phi$ as the low energy degree of freedom. Alternatively we can also view this regime as the regime that is probed when looking at $k \ll \ell' \ll L_T^{-1}$ from the point of view of the $s$-wave superfluid theory introduced above. As mentioned in Section 2.6, this case is better addressed using the gauge fixed fields since this leads to manifestly gauge invariant stresses.

Instead of introducing two Goldstone fields and integrating one out, we consider the outcome of just U(1) spontaneous symmetry breaking and introduce the Goldstone field $\phi$ transforming as in (5.1). Using the fields $\Phi$ and $\tilde{A}_{\mu\nu}$ and their transformations under U(1) as in (2.67) we can construct a gauge invariant scalar potential $\tilde{\Phi}$ and a gauge invariant superfluid velocity $\tilde{\mathcal{A}}_{\mu\nu}$ according to

$$\tilde{\Phi} = \Phi - v^\mu \partial_\mu \phi \,, \qquad \tilde{\mathcal{A}}_{\mu\nu} = \tilde{A}_{\mu\nu} + h_\mu^\rho h_\nu^\sigma \nabla_\rho \partial_\sigma \phi \,. \tag{5.48}$$

Given these invariants we can recast the variation of a general action according to

$$\delta S = \int d^{d+1}x \, e \left( -\bar{T}^\mu \delta\tau_\mu + \frac{1}{2}\bar{\mathcal{T}}^{\mu\nu} \delta h_{\mu\nu} - \bar{J}^0 \delta\Phi + J^{\mu\nu}\delta\tilde{A}_{\mu\nu} + K_\phi \delta\phi \right)$$
$$= \int d^{d+1}x \, e \left( -\bar{T}_\phi^\mu \delta\tau_\mu + \frac{1}{2}\bar{\mathcal{T}}_\phi^{\mu\nu} \delta h_{\mu\nu} - \bar{J}^0 \delta\tilde{\Phi} + J^{\mu\nu}\delta\tilde{\mathcal{A}}_{\mu\nu} + \mathcal{K}_\phi \delta\phi \right) \,, \tag{5.49}$$

where we have defined the gauge invariant stresses $\bar{T}_\phi^\mu$ and $\bar{\mathcal{T}}_\phi^{\mu\nu}$ as well as

$$\mathcal{K}_\phi = K_\phi + v^\mu \partial_\mu \bar{J}^0 - \nabla_\mu \nabla_\nu J^{\mu\nu} \,. \tag{5.50}$$

The Ward identity associated with gauge transformations and the equation of motion for the Goldstone fields gives, respectively

$$\mathcal{K}_\phi = 0\,, \qquad K_\phi = 0\,. \tag{5.51}$$

As for the diffeomorphism Ward identity we can use the second variation in (5.49) to write it as

$$\nabla_\nu \bar{T}_\phi{}^\nu{}_\mu = -\bar{J}^0 \partial_\mu \tilde{\Phi} + J^{\nu\rho}\nabla_\mu \tilde{\mathcal{A}}_{\nu\rho} - 2\nabla_\nu(\tilde{\mathcal{A}}_{\mu\rho}J^{\nu\rho})\,, \tag{5.52}$$

where we have defined the gauge invariant energy-momentum tensor $\bar{T}_\phi{}^\nu{}_\mu = -\bar{T}_\phi^\mu \tau_\nu + \bar{\mathcal{T}}_\phi{}^{\mu\rho}h_{\rho\nu}$. We will now use these considerations to construct the equilibrium partition function.

### 5.5.1 Gradient expansion and equilibrium partition function

As in the previous $s$-wave fracton superfluid case, we require $\tilde{\Phi} \sim \mathcal{O}(1)$ and $\tilde{\mathcal{A}}_{\mu\nu} \sim \mathcal{O}(\partial)$ leading to the same scaling for the Goldstone field as in (5.8). The latter condition implies that flows in the U(1) regime must have a small dipole superfluid velocity. To construct appropriate scalar invariants, we consider the transformations of the gauge fixed fields under the symmetry parameter

$$\begin{aligned}
\delta_K \Phi &= \pounds_k \Phi - v^\mu \partial_\mu \sigma^K = 0\,, \\
\delta_K \tilde{A}_{\mu\nu} &= \pounds_k \tilde{A}_{\mu\nu} + h_\mu^\rho h_\nu^\sigma \nabla_\rho \nabla_\sigma \sigma^K = 0\,,
\end{aligned} \tag{5.53}$$

while the Aristotelian sources transform as (3.9) and $\phi$ as (5.9). Besides the invariant scalars $T, \vec{u}^2$ we can define the chemical potential

$$\mu_U = T\sigma^K + \Phi - \vec{u}^\mu \partial_\mu \phi\,. \tag{5.54}$$

Requiring $\mu_U \sim \mathcal{O}(1)$ we deduce that $\vec{u}^\mu \sim \mathcal{O}(1)$. The scalar potential $\tilde{\Phi}$ is an invariant in itself but the equilibrium condition for $\phi$ in (5.9) sets $\tilde{\Phi} = \mu_U$ in equilibrium and hence $\tilde{\Phi}$ is not an independent scalar. Since no ideal order scalars can be built from $\tilde{\mathcal{A}}_{\mu\nu}$ we have the ideal order hydrostatic effective action for U(1) fracton superfluids

$$S[\tau_\mu, h_{\mu\nu}, \Phi, \tilde{A}_{\mu\nu}; \phi] = \int d^{d+1}x\, e P(T, \vec{u}^2, \mu_U) + \mathcal{O}(\partial)\,. \tag{5.55}$$

Using (5.49) we can easily extract the stresses and currents and obtain

$$\begin{aligned}
\bar{T}_\phi^\mu &= v^\mu P - (sT + m\vec{u}^2)v^\mu + (sT + m\vec{u}^2)\vec{u}^\mu + \mathcal{O}(\partial)\,, \\
\bar{\mathcal{T}}_\phi^{\mu\nu} &= Ph^{\mu\nu} + m\vec{u}^\mu\vec{u}^\nu + 2m\vec{u}^{(\mu}v^{\nu)} + \mathcal{O}(\partial)\,, \\
\bar{J}^0 &= -\rho + \mathcal{O}(\partial)\,, \quad J^{\mu\nu} = \mathcal{O}(\partial)\,,
\end{aligned} \tag{5.56}$$

where we have defined the entropy density, charge density and superfluid density via the Gibbs–Duhem relation $dP = sdT + \rho d\mu_U + md\vec{u}^2/2$. Explicitly computing the

energy density using (5.56) we obtain the Euler relation $\mathcal{E} + P = sT + m\vec{u}^2$, while momentum $p^\mu = \tau_\nu \bar{\mathcal{T}}_\phi^{\ \mu\nu} = -m\vec{u}^\mu \sim \mathcal{O}(1)$. Finally, we can obtain the equilibrium equation for the Goldstone $\phi$ by varying (5.55) with respect to it yielding

$$K_\phi = \nabla_\mu(\rho\vec{u}^\mu)\,, \tag{5.57}$$

where we have used the Ward identity (5.51). It is interesting to note that in this formulation of $s$-wave superfluids, there is no constraint on the fluid velocity, which is what is expected when considering $\ell' \sim \mathcal{O}(\partial^{-1})$. This also makes it clear that equilibrium configurations with non-zero fluid velocity can be realised.

As for the earlier analysis of $s$-wave superfluids, we would like to explore fractonic effects by turning on higher-derivative corrections. We consider the same class of corrections as in (5.17) and define

$$\mathcal{P}(T, \mu_p, \vec{u}^2, \xi) = P(T, \mu_p, \vec{u}^2) + \sum_{I=1} \varepsilon^I \tilde{\kappa}_I(T, \mu_p, \vec{u}^2)\xi^I\,, \tag{5.58}$$

for some transport coefficients $\tilde{\kappa}_I$ and with a slight abuse of notation we defined $\xi = h^{\mu\nu}\tilde{\mathcal{A}}_{\mu\nu}$. As in the previous section, because $\tilde{\mathcal{A}}_{\mu\nu} \sim \mathcal{O}(\partial)$ contains a background scale, we have introduced the bookkeeping coefficient $\varepsilon \sim \mathcal{O}(\partial)$ in (5.58). Given this modified hydrostatic effective action we compute the corrected stresses and currents leading to

$$\begin{aligned}
\bar{T}_\phi^\mu &= v^\mu\mathcal{P} - (\tilde{s}T + \tilde{m}\vec{u}^2)v^\mu + (\tilde{s}T + \tilde{m}\vec{u}^2)\vec{u}^\mu + \mathcal{O}(\partial)\,,\\
\bar{\mathcal{T}}_\phi^{\ \mu\nu} &= \mathcal{P}h^{\mu\nu} + \tilde{m}\vec{u}^\mu\vec{u}^\nu + 2\tilde{m}\vec{u}^{(\mu}v^{\nu)} - 2\tilde{f}_s\tilde{\mathcal{A}}_{\rho\sigma}h^{\rho(\mu}h^{\nu)\sigma} + \mathcal{O}(\partial)\,,\\
\bar{J}^0 &= -\tilde{\rho} + \mathcal{O}(\partial)\,, \quad J^{\mu\nu} = \tilde{f}_s h^{\mu\nu} + \mathcal{O}(\partial)\,,
\end{aligned} \tag{5.59}$$

where $\tilde{s}, \tilde{\rho}, \tilde{m}, \tilde{f}_s$ are defined as in (5.18). Furthermore, the Goldstone equation (5.57) becomes

$$K_\phi = \nabla_\mu(\tilde{\rho}\vec{u}^\mu) + h^{\mu\nu}\nabla_\mu\nabla_\nu\tilde{f}_s\,, \tag{5.60}$$

We can also consider working with gauge non-invariant stresses. Using the first equality in (5.49) we can extract such stresses from (5.55) and obtain

$$\begin{aligned}
\bar{T}^\mu &= \mathcal{P}v^\mu + (\tilde{s}T + \tilde{m}\vec{u}^2)u^\mu + \tilde{\rho}v^\mu v^\nu\partial_\nu\phi - 2\tilde{f}_s v^\rho h^{\sigma\mu}\nabla_\rho\partial_\sigma\phi + h^{\mu\nu}v^\rho\nabla_\nu(\tilde{f}_s\partial_\rho\phi)\,,\\
\bar{\mathcal{T}}^{\mu\nu} &= \mathcal{P}h^{\mu\nu} + \tilde{m}\vec{u}^\mu\vec{u}^\nu + 2\tilde{m}\vec{u}^{(\mu}v^{\nu)} + 2\tilde{\rho}v^{(\mu}h^{\nu)\lambda}\partial_\lambda\phi - 2\tilde{f}_s h^{\rho\mu}h^{\sigma\nu}\tilde{A}_{\rho\sigma} - 2\tilde{f}_s h^{\rho\mu}h^{\nu\sigma}\nabla_\rho\partial_\sigma\phi\\
&\quad + (2h^{\rho(\mu}h^{\nu)\sigma} - h^{\mu\nu}h^{\rho\sigma})\nabla_\sigma(\tilde{f}_s\partial_\rho\phi)\,.
\end{aligned} \tag{5.61}$$

From the point of view of these stresses we can see that momentum is now $p_\mu = -\tilde{m}\vec{u}_\mu - \tilde{\rho}h_\mu^\nu\partial_\nu\phi \sim \mathcal{O}(1)$, thus taking the same gradient ordering as in the general $s$-wave case. In this case spatial momentum is aligned with a "superfluid velocity" given by the gradient of $\phi$ and with $\vec{u}_\mu$. We now consider entropy production for U(1) fracton superfluids.

### 5.5.2 Entropy production

Before linearising the equations and studying linearised perturbations we need to derive the Goldstone equation of motion $K_\phi$ out of equilibrium. As in previous sections we consider the adiabaticity equation, now using (5.49), which takes the form

$$\nabla_\mu N^\mu = -\bar{T}_\phi^\mu \delta_{\mathscr{B}}\tau_\mu + \frac{1}{2}\bar{\mathcal{T}}_\phi^{\mu\nu}\delta_{\mathscr{B}}h_{\mu\nu} - \bar{J}^0\delta_{\mathscr{B}}\tilde{\Phi} + J^{\mu\nu}\delta_{\mathscr{B}}\tilde{\mathcal{A}}_{\mu\nu} + \mathcal{K}_\phi\delta_{\mathscr{B}}\phi + \Delta\,, \quad (5.62)$$

where $\Delta \geq 0$ is a quadratic form. It is again useful to rewrite the adiabaticity equation in terms of entropy production by defining

$$N^\mu = S^\mu + \frac{1}{T}T_\phi{}^\mu{}_\nu u^\nu + \frac{2}{T}J^{\mu\nu}\tilde{\mathcal{A}}_{\nu\rho}u^\rho\,, \quad (5.63)$$

where $N^\mu = Pu^\mu/T$ and $S^\mu = su^\mu$ at ideal order and $N^\mu = \mathcal{P}u^\mu/T$ and $S^\mu = \tilde{s}u^\mu$ when including the higher-order derivative terms we considered. This leads to the off-shell second law of thermodynamics

$$\nabla_\mu S^\mu - \frac{u^\mu}{T}\left[-\nabla_\nu\bar{T}_\phi{}^\nu{}_\mu - \bar{J}^0\partial_\mu\tilde{\Phi} + J^{\nu\rho}\nabla_\mu\tilde{\mathcal{A}}_{\nu\rho} - 2\nabla_\nu(\tilde{\mathcal{A}}_{\mu\rho}J^{\nu\rho})\right] - \mathcal{K}_\phi\delta_{\mathscr{B}}\phi = \Delta \geq 0\,. \quad (5.64)$$

Given the gradient ordering introduced above, the first two terms in the equation above are at least of order $\mathcal{O}(\partial)$. Therefore we must have that

$$K_\phi = -\tilde{\alpha}\delta_{\mathscr{B}}\phi - v^\mu\partial_\mu\bar{J}^0 + \nabla_\mu\nabla_\nu J^{\mu\nu} - \nabla_\mu(\tilde{\rho}\vec{u}^\mu) - h^{\mu\nu}\nabla_\mu\nabla_\nu\tilde{f}_s + \mathcal{O}(\partial^2)\,, \quad (5.65)$$

where $\tilde{\alpha}$ is a dissipative transport coefficient satisfying $\tilde{\alpha} \geq 0$. Once the Goldstone equation of motion is satisfied, $K_\phi = 0$, and the U(1) Ward identity imposed, we find

$$\tilde{\Phi} = \mu_U - \frac{T}{\tilde{\alpha}}\left(\nabla_\mu(\tilde{\rho}\vec{u}^\mu) + h^{\mu\nu}\nabla_\mu\nabla_\nu\tilde{f}_s\right) + \mathcal{O}(\partial)\,, \quad (5.66)$$

where the two last terms in this equation are precisely the equation of motion for $\phi$ in equilibrium (5.60). We note that we have ignored other potential higher-order derivative corrections. We will now use this to look at perturbations around equilibrium states.

### 5.5.3 Linearised equations and modes

As in previous sections we now wish to find signatures of this superfluid phase by finding the spectrum of linear excitations. We consider flat spacetime backgrounds with $\tau_\mu = \delta_\mu^t$ and $h_{\mu\nu} = \delta_{ij}\delta_\mu^i\delta_\nu^j$ and gauge fields

$$\Phi = 0\,, \qquad \tilde{A}_{ij} = \frac{\xi_0}{d}\delta_{ij}\,, \quad (5.67)$$

where $\xi_0$ is the equilibrium value of $\xi$. We also consider states with $u^\mu = (1, \vec{0})$ and constant $T = T_0$, $\mu_U = \mu_0$. In turn this corresponds to the equilibrium values $\sigma^K = \mu_0/T_0$ and $\phi = \mu_0 t$. Considering fluctuations of the various fields we have

$$
\begin{aligned}
T = T_0 + \delta T\,, \quad \mu_U = \mu_0 + \delta\mu_U\,, \qquad \phi = \mu_0 t + \delta\phi\,, \\
u^\mu = (1, \delta v^i)\,, \qquad \tilde{\Phi} = \mu_0 + \partial_t \delta\phi\,, \quad \tilde{\mathcal{A}}_{ij} = \frac{\xi_0}{d}\delta_{ij} + \partial_i\partial_j\delta\phi\,.
\end{aligned}
\tag{5.68}
$$

Using the conservation law (5.52), the Ward identity (5.51), the Goldstone equation (5.66) and taking the stresses and currents with high-derivative corrections (5.59) we obtain the linearised equations

$$
\begin{aligned}
\partial_t\delta\mathcal{P} &= \tilde{s}_0\partial_t\delta T + T_0\partial_t\delta\tilde{s} + \tilde{s}_0 T_0 \partial_i\delta v^i + \tilde{\rho}_0\partial_t\partial_t\delta\phi + \tilde{f}_s^0\partial_t\partial^i\partial_i\delta\phi\,, \\
\partial_i\delta\mathcal{P} &= \tilde{\rho}_0\partial_i\partial_t\delta\phi + \tilde{f}_s^0\partial_i\partial^j\partial_j\delta\phi + \tilde{m}_0\partial_t\delta v_i\,, \\
\partial_t\delta\tilde{\rho} &= \partial^i\partial_i\delta\tilde{f}_s\,, \\
\partial_t\delta\phi &= \delta\mu_U - \frac{T_0}{\tilde{\alpha}_0}\left[\partial_i(\tilde{\rho}_0\delta v^i) + \partial^i\partial_i\delta\tilde{f}_s\right]\,.
\end{aligned}
\tag{5.69}
$$

Using plane wave perturbations as in the previous sections, and choosing the scaling $\varepsilon \gg k$, these equations lead to 4 non-vanishing gapless modes of the form

$$
\begin{aligned}
\omega &= \pm\bar{v}_s k - \frac{i}{2}\bar{\Gamma}_s k^2 + \mathcal{O}(k^3)\,, \\
\omega &= \pm\omega_M k^2 - \frac{i}{2}\Gamma_M k^4\,,
\end{aligned}
\tag{5.70}
$$

where $\omega_M$ and $\Gamma_M$ are as in (5.43). If $\omega_M^2$ is positive then the first two modes in (5.70) are magnon-like modes with $\omega_M^2$ the "magnon velocity" and $\Gamma_M$ is a subdiffusive coefficient. Otherwise, if $\omega_M^2$ is negative then the first two modes are diffusive modes but since the ansatz is $e^{i(-\omega t + k_i x^i)}$, allowing for this possibility would lead to an instability and as such we discard it as unphysical. We thus take $\omega_M^2 \geq 0$ and $\Gamma_U \geq 0$ for stability at arbitrary $k$. For the sound mode, we find

$$
\begin{aligned}
\bar{v}_s^2 &= \frac{\tilde{s}_0^2(\partial\tilde{\rho}/\partial\mu_p)_0}{\tilde{m}_0((\partial\tilde{s}/\partial\mu_p)_0^2 - (\partial\tilde{s}/\partial T)_0(\partial\tilde{\rho}/\partial\mu_p)_0)}\,, \\
\bar{\Gamma}_s &= \frac{T_0\tilde{\rho}_0^2}{\tilde{\alpha}_0\tilde{m}_0} - \frac{\tilde{s}_0\tilde{\rho}_0^2(\partial\tilde{\rho}/\partial\mu_p)_0}{\tilde{\alpha}_0\tilde{m}_0((\partial\tilde{s}/\partial\mu_p)_0^2 - (\partial\tilde{s}/\partial T)_0(\partial\tilde{\rho}/\partial\mu_p)_0)}\,.
\end{aligned}
\tag{5.71}
$$

As we mentioned in 5.4, we should recover the low energy spectrum of the $s$-wave superfluids when $\ell' \gg k$. Indeed, the magnon-like modes in (5.70) are precisely the magnon-like modes of the $s$-wave (5.43), but the attenuation of the sound mode is *different* from (5.42). To remedy this, we write $\tilde{\alpha}_0$ as

$$
\tilde{\alpha}_0 \to \alpha_0 + \Delta(\alpha_0)\,,
\tag{5.72}
$$

which we can solve for $\Delta(\alpha_0)$ to make the attenuations match. However, $\Gamma_M$ *also* involves $\tilde{\alpha}_0$, and so redefining it means that the subdiffusive attenuations do not match. Again, we can fix this by redefining, for instance, $(\partial\tilde{\rho}/\partial\xi)_0 \to (\partial\tilde{\rho}/\partial\xi)_0 + F$, where $F$ is a complicated function of the thermodynamic variables. This is similar to the various redefinitions that need to be implemented to obtain the liquid regime after melting [65]. As before, these modes should be expanded in powers of $\varepsilon$, which is equivalent to the regime $\varepsilon \sim k$ and which reproduces the modes in (5.45) and (5.46), though the attenuations only match after performing a field redefinition à la (5.72). This expansion clarifies that the sound mode appears at ideal order in derivatives while the magnon mode appears at first order in derivatives. The regime $\varepsilon \ll k$ leads to similar conclusions as discussed at the end of Section 5.4.

In addition, contrary to the $p$-wave fracton superfluids, the U(1) fracton superfluid allows for equilibrium states with non-zero background velocity $v_0^i$ such that $k_i v_0^i = v_0 k \cos\theta$ for some angle $\theta$. If we perturb around such states with $u^\mu = (1, v_0^i)$, the linearised (modified) Gibbs–Duhem relation becomes

$$\delta\mathcal{P} = \tilde{s}_0 \delta T + \tilde{\rho}_0 \delta\mu_p + \tilde{m}_0 v_0^i \delta v^i - i\tilde{f}_s^0 k^2 \delta\phi\,. \tag{5.73}$$

For simplicity, we assume that $v_0^i$ is aligned with $k_i$ (i.e., $\theta = 0$), which leads to modes of the form

$$\begin{aligned}
\omega &= (\pm\bar{v}_s + bv_0 + \mathcal{O}(v_0^2))k - \frac{i}{2}(\bar{\Gamma}_s + cv_0 + \mathcal{O}(v_0^2))k^2\,, \\
\omega &= (\pm\omega_M + (\pm A + iB)v_0^2 + \mathcal{O}(v_0^3))k^2 + \mathcal{O}(v_0 k^3)\,,
\end{aligned} \tag{5.74}$$

where $b, c, A, B$ are complicated functions of the thermodynamic variables and where we have expanded in small $v_0$ to simplify the expressions. Thus, including a non-zero background velocity strongly affects the mode structure in the U(1) regime, which is similar to what happens for an ordinary Aristotelian fluid [47]. In particular we can see that the sound speeds have different velocities due to a non-vanishing equilibrium velocity. This is a natural consequence of the absence of boost symmetry. This concludes our analysis of fracton superfluids.

# 6 Discussion

In this section, we provide an overview of our results and list important future directions inspired by our work with the goal of understanding many-body quantum systems with emergent fracton symmetries.

We began by gauging the fracton algebra in order to extract both the background geometry as well as the low-energy degrees of freedom, which are those relevant for hydrodynamics, that describe field theories with a conserved dipole moment. This procedure allowed us to recover the gauge fields present in symmetric tensor gauge

theories that have been widely studied in the literature, but also more general gauge fields that reduce to those of the symmetric tensor gauge theory when an appropriate curvature constraint is imposed. In particular, the gauge field $\tilde{A}^a_\mu$ that we found in Section 2.4 reduces to the symmetric tensor gauge field $A_{\mu\nu}$ once the U(1) part of the Cartan curvature is set to zero.

Our main motivation was to understand the thermodynamics of equilibrium global states in such theories and their near-equilibrium dynamics. As such, we began by showing that, at finite temperature, no such global thermal states with non-zero flow velocity can exist unless the fracton symmetries are spontaneously broken.[24]

Given this crucial observation, we introduced two different classes of fracton superfluids distinguished by their particular symmetry breaking pattern; i.e., whether one, or both, of the U(1) and the dipole symmetries are spontaneously broken, which leads to two distinct superfluid phases, one of which has two different regimes. We noted several novel features of fracton superfluids compared to conventional Aristotelian superfluids which we studied in Appendix B, and which had not yet been coupled to non-trivial Aristotelian backgrounds. For $p$-wave superfluids these features included, for instance, the observation that the spatial fluid velocity satisfies $\vec{u}^\mu \sim \mathcal{O}(\partial)$ and that the gradient expansion imposes a different scaling for space and time derivatives. For $s$-wave superfluids we followed the gradient scheme proposed in [48] which leads to the point of view that at ideal order $s$-wave fracton superfluids are conventional Aristotelian superfluids with a constraint on the spatial fluid velocity imposed by the dipole Ward identity.

We have also demonstrated that these different classes of superfluids have distinctive signatures, highlighted by the different ideal order mode structure depicted in Fig. 1. The possibility of measuring such dispersion relations, say via specific correlators, would allow to determine which phase of fracton superfluidity a given system would be in. We have also shown that fracton hydrodynamic theories considered earlier in the literature [23–25] fall into the class of $p$-wave fracton superfluids introduced in Section 4, which is also the class of theories discussed in [29].[25]

It is important to clarify a few points regarding the $s$-wave fracton superfluid phase that we described in Section 3.3. From an effective field theory point of view, the hydrostatic effective action (5.12) should contain all scalars allowed by symmetry.

---

[24]At zero temperature it is possible to describe other fracton phases such as what is referred to as the "normal phase" in [50] which does not require the introduction of Goldstone fields.

[25]The recent reference [50] introduces two classes of fracton superfluids, the dipole condensate, which is equivalent to the $p$-wave fracton superfluid studied here, and the charge condensate which is equivalent to the $s$-wave fracton superfluid studied here. Also, Ref. [50] focuses on zero temperature fluids without momentum conservation and with an emphasis on dissipative effects. The $p$-wave and $s$-wave phases introduced here appear to have the same field content but a proper comparison is not possible given that in this paper we work at finite temperature with both energy and momentum conservation, and we did not investigate dissipative effects.

We note that as soon as we add the scalar $\mathcal{B}^2$ (without controlling the coupling constant $\ell'$) we can use the dipole Ward identity (5.15) to eliminate $\Psi_\mu$ from the hydrodynamic theory [41]. This procedure is equivalent to considering a large $\ell' \sim \mathcal{O}(\partial^{-1})$. In such circumstances, the field content of the theory only contains a single Goldstone field $\phi$ but both U(1) and dipole symmetries are spontaneously broken. This situation we described in further detail in Section 5.5. Thus at low energies, there are only two fracton superfluid phases: the $p$-wave phase and the U(1) regime of the $s$-wave phase.[26] However, taking $\ell' \sim \mathcal{O}(1)$ we can describe a "pinned" $s$-wave fracton superfluid phase, that shares similar features to [38, 42, 65], in which a "mass term" for the vector Goldstone $\Psi_\mu$ is included in the hydrostatic effective action.[27] The introduction of the parameter $\ell'$ allows us to model a phase transition from a pinned $s$-wave fracton superfluid regime ($\ell' \ll k$) to a U(1) fracton superfluid regime ($\ell' \gg k$). In [41], a microscopic model for $s$-wave fracton superfluids was proposed in which the "mass term" becomes large at low temperatures and the vector Goldstone $\Psi_\mu$ can be integrated out. In the context of this model, $\ell'$ increases with decreasing temperature. It would be interesting to better understand this transition from the point of view of this microscopic model.

A few comments about the possible gradient expansions for $s$-wave superfluids are in order. Originally we introduced a gradient expansion similar to $p$-wave superfluids and applied it to $s$-wave superfluids, leading to the scalings $\Psi_\mu \sim \mathcal{O}(\partial^{-1})$ and $\phi \sim \mathcal{O}(\partial^{-2})$ as well as $\vec{u}^\mu \sim \mathcal{O}(\partial^{-1})$. One of the issues with this scaling is that, contrary to $p$-wave superfluids, in $s$-wave superfluids the dipole Ward identity does not constrain the spatial fluid velocity to be gradient-suppressed and we are not aware of a way to impose such a constraint by hand as $\vec{u}^\mu$ evolves in time. The second issue with such a scaling is that including the supposedly second order term $\int d^{d+1}x\, em\vec{u}^2$ for a coefficient $m(T, \mu_U, \xi)$ in the hydrostatic effective action (5.55) leads to the appearance of a linear mode. Hence the addition of this particular higher derivative term is affecting the infra-red regime.[28] We have discussed in detail in Appendix C this gradient expansion and its physical consequences. In the main text we instead followed the alternative gradient scheme proposed in [48]. Within this scheme the dipole superfluid velocity $\mathcal{A}_{\mu\nu}$ must be treated perturbatively since $\mathcal{A}_{\mu\nu} \sim \mathcal{O}(\partial)$ and which we accounted for by treating the corresponding dipole superfluid density $f_s$ perturbatively. We summarised the mode structure in Section 5.4 and showed that this gradient scheme leads to a consistent expansion in wavenumber $k$ in which the dipole superfluid velocity appears as small corrections to the various coefficients. As we noted in Section 5, we only studied a particular subset of higher order corrections

---

[26]We thank Akash Jain and Piotr Surówka for pointing this out to us.

[27]We note that the origin of the "mass term" in [38, 42, 65] is different from the one employed here since none of the dipole symmetries are explicitly broken.

[28]One might also speculate whether this is related to UV/IR mixing, which is known to occur in fracton theories [5].

to the hydrostatic effective action and a more thorough investigation of the higher-derivative structure of $s$-wave superfluids would be useful to clarify the nature of the gradient expansion and which gradient expansion is the most suitable one to organise the hydrodynamic theory. We also note that to the level of effects and higher-derivative corrections that we studied here, both these schemes give rise to the same spectra at second order in the gradient expansion.

A theory of fracton superfluids at ideal order constitutes the first step towards understanding fracton theories in thermal equilibrium and identifying potential experimental signatures. Nevertheless, many interesting properties of many-body quantum systems generically rely on the structure of dissipation. While we have not considered dissipative effects in detail, we expect that many interesting types of transport will appear at higher orders in the gradient expansion, including sub-diffusive behaviour and more reported in [23–25, 29, 50]. Moreover, while we have identified the thermodynamics and linear excitations of fracton superfluids, we have not computed hydrodynamic correlation functions which are important for potential measurements; we leave this important generalisation for future work.

Another interesting direction involves working with the gauge field $\tilde{A}^a_\mu$ obtained by gauging the fracton algebra in Section 2.4 without imposing the curvature constraint (2.33) that reduces $\tilde{A}^a_\mu$ to the symmetric tensor gauge field $A_{\mu\nu}$. This generalisation would allow for further effects even at ideal order, including the presence of additional chemical potentials.[29] Finally, it is also possible to consider fracton hydrodynamic theories with conserved multipole moments. The geometry and gauge fields for such theories were discussed in [18] and the methodology developed in this work could be applied to these cases as well. More generally, one could aim at studying hydrodynamic theories with subsystem symmetries [73, 74]. This should be possible by gauging subsystem symmetries as in Section 2.2 and obtain the curved background fields.[30]

## Acknowledgments

We are grateful to José Figueroa-O'Farrill, Jelle Hartong, Akash Jain and Piotr Surówka for useful discussions. We also thank Akash Jain, Kristan Jensen, Ruochuan Liu and Eric Mefford for sharing their draft [48]. JA is partly supported by the Dutch Institute for Emergent Phenomena (DIEP) cluster at the University of Amsterdam via the programme Foundations and Applications of Emergence (FAEME). The work of EH is supported by Jelle Hartong's Royal Society University Research Fellowship

---

[29]The authors of Ref. [29] introduce a gauge field similar to $\tilde{A}^a_\mu$. We thus expect that our approach will completely match that of [29] once such an extension is carried out. We have nevertheless matched our results with [29] to the extent that the chemical potentials we construct using $A_{\mu\nu}$ match those introduced in that reference.

[30]Gauging of subsystem symmetries has been considered in certain contexts, e.g. [75].

(renewal) "Non-Lorentzian String Theory" (URF\R\221038) via an enhancement award.

# A  Variational calculus for Aristotelian geometry

In this appendix, we provide useful and explicit variational relations for Aristotelian geometry as well as for the fluid variables that are defined using the background geometry, complementing Section 2.3. By varying the completeness relation

$$\delta^\mu_\nu = -v^\mu \tau_\nu + h^{\mu\rho} h_{\rho\nu}\,, \tag{A.1}$$

and using the following properties of the fields that make up the Aristotelian geometry

$$v^\mu \tau_\mu = -1\,, \qquad v^\mu h_{\mu\nu} = \tau_\mu h^{\mu\nu} = 0\,, \tag{A.2}$$

we obtain the variations

$$\delta e = e\left(-v^\mu \delta\tau_\mu + \frac{1}{2} h^{\mu\nu}\delta h_{\mu\nu}\right)\,, \quad \delta h^{\mu\nu} = 2v^{(\mu} h^{\nu)\rho}\delta\tau_\rho - h^{\mu\rho} h^{\nu\sigma}\delta h_{\rho\sigma}\,,$$
$$\delta h^\mu_\nu = v^\mu h^\rho_\nu \delta\tau_\rho - \tau_\nu h^{\mu\rho} v^\sigma \delta h_{\rho\sigma}\,, \qquad \delta v^\mu = v^\mu v^\nu \delta\tau_\nu - h^{\mu\nu} v^\rho \delta h_{\rho\nu}\,. \tag{A.3}$$

As explained in Section 3.2, hydrodynamic theories based on Aristotelian geometry in equilibrium configurations involve a Killing vector $k^\mu$ that defines the local temperature $T$ and the fluid velocity $u^\mu$ defined via (3.11). Varying these quantities leads to

$$\delta T = -T u^\mu \delta\tau_\mu\,, \qquad \delta u^\mu = -u^\mu u^\nu \delta\tau_\nu\,, \qquad \delta\vec{u}^2 = u^\mu u^\nu \delta h_{\mu\nu} - 2u^2 u^\mu \delta\tau_\mu\,. \tag{A.4}$$

We recall that the affine connection that we use throughout the paper is given by (2.21), and that we assume that this is torsion-free, corresponding to the vanishing of the intrinsic torsion

$$K_{\mu\nu} = \tau_{\mu\nu} = 0\,. \tag{A.5}$$

The Riemann tensor associated with the affine connection (2.21) is defined via the Ricci identity

$$[\nabla_\mu, \nabla_\nu] X_\rho = R_{\mu\nu\rho}{}^\sigma X_\sigma\,,$$
$$[\nabla_\mu, \nabla_\nu] Y^\rho = -R_{\mu\nu\sigma}{}^\rho Y^\sigma\,, \tag{A.6}$$

where $X_\mu$ is an arbitrary 1-form and $Y^\mu$ an arbitrary vector. The components of the Riemann tensor are

$$R_{\mu\nu\rho}{}^\sigma = -\partial_\mu \Gamma^\sigma_{\nu\rho} + \partial_\nu \Gamma^\sigma_{\mu\rho} - \Gamma^\sigma_{\mu\lambda}\Gamma^\lambda_{\nu\rho} + \Gamma^\sigma_{\nu\lambda}\Gamma^\lambda_{\mu\rho}\,. \tag{A.7}$$

The Ricci scalar is defined as the contraction $R_{\mu\nu} = R_{\mu\rho\nu}{}^\rho$. Finally, varying the connection (2.21) and setting $K_{\mu\nu} = \tau_{\mu\nu} = 0$ produces the result

$$
\begin{aligned}
\delta\Gamma^\rho_{\mu\nu} = {}&-v^\rho \nabla_\mu \delta\tau_\nu + \frac{1}{2}h^{\rho\sigma}(\nabla_\mu \delta h_{\nu\sigma} + \nabla_\nu \delta h_{\mu\sigma} - \nabla_\sigma \delta h_{\mu\nu}) \\
&- h^{\rho\lambda}\tau_\nu v^\sigma \nabla_{(\mu}\delta h_{\lambda)\sigma} + \frac{1}{2}h^{\rho\lambda}\tau_\nu \pounds_v \delta h_{\mu\lambda}\,,
\end{aligned}
\tag{A.8}
$$

where we remark that the two terms on the second line will play no rôle in the main text since they always appear contracted with a spatial tensor. The restriction to $\tau_{\mu\nu} = 0$ means that we do not have access to the full energy current but only its divergence (see, e.g., [76]), while the restriction to $K_{\mu\nu} = 0$ means that we also do not have access to the most general stress-momentum tensor $\mathcal{T}^{\mu\nu}$. However, such terms play no rôle when restricting to torsion-free spacetimes.

# B  Ideal relativistic and Aristotelian superfluids

In this appendix we briefly review ideal relativistic superfluids and later we provide a formulation of ideal Aristotelian superfluids coupled to curved geometries and nontrivial gauge fields. The purpose of this appendix is for the reader to familiarise themselves with conventional theories of superfluid dynamics, which can be contrasted with the fracton superfluid theories introduced in the main text.

## B.1  Ideal relativistic superfluids

The ideal relativistic superfluid was described using equilibrium partition function methods, for example, in [30] (see also [36, 68]). The dynamics of relativistic superfluids was discussed in [77, 78]. We begin by introducing the gradient expansion and equilibrium partition function. Then we move on to entropy production and linearised fluctuations.

### B.1.1  Gradient expansion and equilibrium partition function

Relativistic superfluids are coupled to a background metric $g_{\mu\nu}$ and a gauge field $A_\mu$ that transforms under gauge transformations as $\delta A_\mu = \partial_\mu \sigma$ where $\sigma$ is a gauge parameter. Since we are dealing with a superfluid, the description includes a Goldstone field $\phi$ that transforms as[31]

$$
\delta_\sigma \phi = \sigma\,,
\tag{B.1}
$$

under U(1) gauge transformations with parameter $\sigma$. This allows us to define the superfluid velocity according to

$$
\xi_\mu := A_\mu - \partial_\mu \phi\,,
\tag{B.2}
$$

---

[31]Note that the U(1) Goldstone transforms with the opposite sign in this appendix as compared to the U(1) Goldstone we introduced for the $s$-wave fracton superfluid in Section 5.

which is manifestly gauge invariant. Requiring the superfluid velocity to be ideal order $\xi_\mu \sim \mathcal{O}(1)$ implies that

$$A_\mu \sim \mathcal{O}(1)\,, \qquad \phi \sim \mathcal{O}(\partial^{-1})\,. \tag{B.3}$$

In equilibrium, we introduce the set of parameters $K$ consisting of an isometry captured by a Killing vector field $k^\mu$ and a background U(1) parameter $\sigma^K$ such that

$$\delta_K g_{\mu\nu} = \pounds_k g_{\mu\nu} = 0\,, \quad \delta_K A_\mu = \pounds_k A_\mu + \partial_\mu \sigma^K = 0\,, \quad \delta_K \phi = \pounds_k \phi + \sigma^K = 0\,. \tag{B.4}$$

Note that these imply that $\delta_K \xi_\mu = 0$ in equilibrium. The U(1) parameter transforms as

$$\delta \sigma^K = \pounds_\zeta \sigma^K - \pounds_k \sigma\,, \tag{B.5}$$

under diffeomorphisms parameterised by $\zeta^\mu$ and U(1) gauge transformations with parameter $\sigma$. This result may be obtained from requiring that $\delta(\delta_K \phi) = 0$ and using the last equation in (B.4). From the gauge-invariant object $\xi_\mu$ and the Killing vector $k^\mu$, we can construct the following gauge invariant hydrodynamical variables

$$T = \frac{T_0}{\sqrt{-k^2}}\,, \qquad u^\mu = \frac{k^\mu}{\sqrt{-k^2}}\,, \qquad \mu = u^\mu A_\mu + T\sigma^K\,, \qquad \xi^2 = g^{\mu\nu}\xi_\mu \xi_\nu\,, \tag{B.6}$$

where $k^2 = g_{\mu\nu}k^\mu k^\nu$, and where the first three are the usual scalars introduced for a relativistic charged fluid with unbroken U(1) global symmetry. Note that we also have an *extra* scalar $u^\mu \xi_\mu$ but this is the same as $\mu$ in equilibrium, since

$$u^\mu \xi_\mu = u^\mu A_\mu - u^\mu \partial_\mu \phi = u^\mu A_\mu + T\sigma^K = \mu\,, \tag{B.7}$$

where we used that $\delta_K \phi = 0$, so that $u^\mu \partial_\mu \phi = T\pounds_k \phi = -T\sigma^K$. In terms of these variables, the ideal hydrostatic effective action takes the form

$$S^{\text{HS}}[g_{\mu\nu}, A_\mu; \phi] = \int d^{d+1}x\, \sqrt{-g} P(T, \mu, \xi^2) + \mathcal{O}(\partial)\,. \tag{B.8}$$

The variation of $P$ gives rise to the following Gibbs–Duhem relation

$$\begin{aligned} dP &= \left(\frac{\partial P}{\partial T}\right)_{\mu,\xi^2} dT + \left(\frac{\partial P}{\partial \mu}\right)_{T,\xi^2} d\mu + \left(\frac{\partial P}{\partial \xi^2}\right)_{T,\mu} d\xi^2 \\ &:= s\, dT + \rho\, d\mu - \frac{1}{2}f\, d\xi^2\,, \end{aligned} \tag{B.9}$$

where we defined

$$s = \left(\frac{\partial P}{\partial T}\right)_{\mu,\xi^2}\,, \qquad \rho = \left(\frac{\partial P}{\partial \mu}\right)_{T,\xi^2}\,, \qquad f = -2\left(\frac{\partial P}{\partial \xi^2}\right)_{T,\mu}\,. \tag{B.10}$$

Variations of $S^{\text{HS}}$ with respect to the background fields and Goldstone field can be parameterised as

$$\delta S^{\text{HS}} = \int d^{d+1}x \sqrt{-g} \left( \frac{1}{2} T^{\mu\nu} \delta g_{\mu\nu} + J^\mu \delta A_\mu - K_{\phi,\text{HS}} \delta\phi \right), \qquad \text{(B.11)}$$

where $K_{\phi,\text{HS}}$ is the hydrostatic contribution to the Goldstone equation of motion. In equilibrium $K_{\phi,\text{HS}} = 0$. In order to extract the equilibrium currents, we note that the variations of the gauge invariant scalars in (B.6) are

$$\delta T = \frac{T}{2} u^\mu u^\nu \delta g_{\mu\nu}, \qquad \delta u^\mu = \frac{1}{2} u^\mu u^\nu u^\rho \delta g_{\nu\rho}, \qquad \delta\mu = \frac{\mu}{2} u^\mu u^\nu \delta g_{\mu\nu} + u^\mu \delta A_\mu,$$
$$\delta\xi^2 = -g^{\mu\rho} g^{\nu\sigma} \delta g_{\rho\sigma} \xi_\mu \xi_\nu + 2g^{\mu\nu} \xi_\mu \delta A_\nu - 2\xi^\mu \partial_\mu \delta\phi. \qquad \text{(B.12)}$$

Using these, and the relation $\delta\sqrt{-g} = \frac{1}{2}\sqrt{-g} g^{\mu\nu} \delta g_{\mu\nu}$, the energy-momentum tensor and U(1) current, as well as the hydrostatic equation of motion for $\phi$, become

$$T^{\mu\nu} = \mathcal{E} u^\mu u^\nu + P\Delta^{\mu\nu} + f\xi^\mu \xi^\nu,$$
$$J^\mu = \rho u^\mu - f\xi^\mu, \qquad \text{(B.13)}$$
$$K_{\phi,\text{HS}} = \nabla_\mu(f\xi^\mu),$$

where $\xi^\mu = g^{\mu\nu}\xi_\nu$ and $\Delta^{\mu\nu} = g^{\mu\nu} + u^\mu u^\nu$, and where we used the Euler relation and first law of thermodynamics

$$\mathcal{E} + P = sT + \mu\rho, \qquad d\mathcal{E} = Tds + \mu d\rho + \frac{1}{2} f d\xi^2. \qquad \text{(B.14)}$$

In turn, the U(1) and diffeomorphism Ward identities are given by, respectively

$$\nabla_\mu J^\mu + K^\phi_{\text{HS}} = 0,$$
$$\nabla_\nu T^\nu{}_\mu + J^\nu F_{\nu\mu} - K^\phi_{\text{HS}} \xi_\mu = 0, \qquad \text{(B.15)}$$

where we used the U(1) Ward identity to simplify the diffeomorphism Ward identity.

### B.1.2 Entropy production and the Josephson equation

Out of equilibrium the equation of motion for the Goldstone typically differs from its hydrostatic counterpart. To derive it, we consider the adiabaticity equation derivable from (B.11), namely

$$\nabla_\mu N^\mu = \frac{1}{2} T^{\mu\nu} \delta_{\mathscr{B}} g_{\mu\nu} + J^\mu \delta_{\mathscr{B}} A_\mu - K_\phi \delta_{\mathscr{B}} \phi + \Delta, \qquad \text{(B.16)}$$

for a quadratic form $\Delta \geq 0$. Here $\mathscr{B} = (\chi^\mu, \sigma^B)$ is the set of symmetry parameters that in equilibrium yield $K = (k^\mu, \sigma^K)$. The action of $\mathscr{B}$ on the various background sources and dynamical fields is the same as for $K$. We note that at ideal order we have $N^\mu = Pu^\mu/T$. We have also defined $K_\phi$ as the full equation for the Goldstone

field $\phi$ which on-shell satisfies $K_\phi = 0$. Instead of finding solutions to the adiabaticity it is convenient to define the entropy current via

$$N^\mu = S^\mu + \frac{u^\nu}{T} T^\mu{}_\nu + \frac{\mu}{T} J^\mu \,, \tag{B.17}$$

where at ideal order $S^\mu = s u^\mu$. In terms of this we can recast the adiabaticity equation as the off-shell second law of thermodynamics

$$\nabla_\mu S^\mu + \frac{u^\nu}{T}(\nabla_\mu T^\mu{}_\nu - J^\mu F_{\nu\mu}) + \frac{\mu}{T}\nabla_\mu J^\mu + K_\phi \delta_\mathscr{B} \phi = \Delta \,. \tag{B.18}$$

To solve this equation we note that given the gradient expansion introduced earlier, the first 3 terms in the above equation are of $\mathcal{O}(\partial)$. This allows us to extract the Josephson equation in the form

$$K_\phi = \frac{\alpha}{T}(\mu - u^\mu \xi_\mu) + \nabla_\mu(f \xi^\mu) + \mathcal{O}(\partial) \,, \tag{B.19}$$

for a dissipative transport coefficient $\alpha \geq 0$ and where we have used the ideal order currents to recover the hydrostatic correction $\nabla_\mu(f \xi^\mu)$. We precisely see that when $K_\phi = 0$, $u^\mu \xi_\mu$ is determined in terms of the other hydrodynamic variables.

### B.1.3 Linearised equations and modes

We now consider the linearised equations of motion and linear perturbations in flat spacetime. We can choose Cartesian coordinates $x^\mu = (t, x^i)$ in terms of which the metric takes the form

$$ds^2 = \eta_{\mu\nu} dx^\mu dx^\nu = -dt^2 + \delta_{ij} dx^i dx^j \,. \tag{B.20}$$

In addition we focus on vanishing gauge fields $A_\mu = 0$ and perturb around an equilibrium state with constant temperature and chemical potential $T_0, \mu_0$, and with $u^\mu = (1, \vec{0})$ and $\phi = -\mu_0 t$. The relevant perturbations can be parameterised as

$$T = T_0 + \delta T \,, \qquad u^\mu = \delta_t^\mu + \delta v^i \delta_i^\mu \,, \qquad \phi = -\mu_0 t + \delta\phi \,, \qquad \mu = \mu_0 + \delta\mu \,, \tag{B.21}$$

where $\sigma_0^K = \mu_0 / T_0$. This parameterisation implies

$$\xi_\mu = \mu_0 \delta_\mu^t - \partial_\mu \delta\phi \,, \quad \xi^0 = \partial_t \delta\phi - \mu_0 \,, \quad \xi^i = -\partial_i \delta\phi \,, \quad \delta\xi^2 = 2\mu_0 \partial_t \delta\phi \,. \tag{B.22}$$

The dynamics of the perturbations are governed by the conservation law and the U(1) Ward identity (B.15) as well as the Josephson equation (B.19). Ignoring the dissipative coefficient $\alpha$, the linearised equations become

$$\begin{aligned}
\partial_t \delta\mathcal{E} &= -\mu_0^2 \partial_t \delta f + 2 f_0 \mu_0 \partial_t \partial_t \delta\phi - w_0 \partial_i \delta v^i - f_0 \mu_0 \partial^i \partial_i \delta\phi \,, \\
\partial^i \delta P &= -w_0 \partial_t \delta v^i - f_0 \mu_0 \partial_t \partial^i \delta\phi \,, \\
\partial_t \delta\rho &= -\mu_0 \partial_t \delta f + f_0 \partial_t \partial_t \delta\phi - f_0 \partial^i \partial_i \delta\phi - \rho_0 \partial_i \delta v^i \,, \\
\partial_t \delta\phi &= -\delta\mu \,,
\end{aligned} \tag{B.23}$$

where we have defined $w_0 = \mathcal{E}_0 + P_0$. Using a plane wave ansatz for the perturbations we find two pairs of sound modes

$$\omega = \pm v_1 k \,, \qquad \omega = \pm v_2 k \,. \tag{B.24}$$

These frequencies are solutions of an equation of the form $A_0 - B_0 \omega_1^2 + C_0 \omega_1^4 = 0$ for some coefficients $A_0, B_0, C_0$ which are complicated functions of the thermodynamic variables. In terms of these, the sound speeds are given by

$$v_1^2 = \frac{1}{2C_0}\left( B_0 - \sqrt{B_0^2 - 4A_0 C_0} \right) \,, \qquad v_2^2 = \frac{1}{2C_0}\left( B_0 + \sqrt{B_0^2 - 4A_0 C_0} \right) \,. \tag{B.25}$$

The second pair of sound modes is the well-known "second sound" of superfluids. This concludes our discussion of ideal relativistic superfluids.[32]

## B.2 Ideal Aristotelian superfluids

In this appendix we discuss ideal Aristotelian superfluids. This class of superfluids has been explored in [42] and also in [79]. However, a full analysis of their ideal order structure and the coupling to general curved Aristotelian geometries and non-trivial gauge fields has not previously appeared in the literature and we provide such extension.[33] Nevertheless, our analysis, even at ideal order, is not exhaustive as for instance we do not compute the linear spectrum of excitations. Such an analysis is interesting and important, and we leave it for future work.

### B.2.1 Gradient expansion and equilibrium partition function

Aristotelian superfluids couple to the Aristotelian structure $(\tau_\mu, h_{\mu\nu})$ introduced in Section 2.3 and a gauge field $A_\mu$. In addition, as for the relativistic case we introduce a Goldstone field $\phi$. These fields transform according to

$$\delta\tau_\mu = \pounds_\xi \tau_\mu \,, \qquad \delta h_{\mu\nu} = \pounds_\xi h_{\mu\nu} \,, \qquad \delta A_\mu = \pounds_\xi A_\mu + \partial_\mu \sigma \,, \qquad \delta\phi = \pounds_\xi \phi + \sigma \,, \tag{B.26}$$

under infinitesimal diffeomorphisms $\xi^\mu$ and gauge transformations $\sigma$. These fields allow us to define the gauge invariant superfluid velocity as in (B.2). Requiring it to be $\mathcal{O}(1)$ implies again the gradient scheme (B.3). Given this we can now focus on additional equilibrium scalars. Under the existence of the isometry parameters $K = (k^\mu, \sigma^K)$, where $\sigma^K$ transforms as in (B.5), satisfying the equilibrium conditions

$$\delta_K \tau_\mu = \pounds_k \tau_\mu = 0 \,, \quad \delta_K h_{\mu\nu} = \pounds_k h_{\mu\nu} = 0 \,, \quad \delta_K A_\mu = \pounds_k A_\mu + \partial_\mu \sigma^K = 0 \,, $$
$$\delta_K \phi = \pounds_k \phi + \sigma^K = 0 \,, \tag{B.27}$$

---

[32]First order dissipative corrections to relativistic superfluids were considered at length in [66].
[33]On the other hand, Galilean superfluids have been comprehensively investigated [80].

it is possible to construct various additional scalars. In particular, we can construct the temperature $T$ and square of fluid velocity $\vec{u}^2$ introduced in (3.11). In addition we can form the following gauge invariant scalars

$$\bar{\xi}^2 = h^{\mu\nu}\xi_\mu\xi_\nu\,, \qquad \mu = T\sigma^K + u^\mu A_\mu\,, \qquad \nu = h_\nu^\mu u^\nu \xi_\mu\,. \qquad (B.28)$$

We note that all these scalars are order $\mathcal{O}(1)$ and that $\vec{u}^\mu \sim \mathcal{O}(1)$. The superfluid velocity also allows us to define two more scalars, namely, $v^\mu\xi_\mu$ and $u^\mu\xi_\mu$. However these are not independent since $\nu + v^\mu\xi_\mu = u^\mu\xi_\mu$, and in equilibrium $\mu = u^\mu\xi_\mu$. As there is no restriction on $\vec{u}^\mu$, a conventional Aristotelian fluid can have additional ideal order scalars compared with the $s$-wave fracton superfluids of Section 5. In particular, for $s$-wave fracton superfluids the scalar $\nu$ is the same as for the Aristotelian superfluids described here. However, a version of $\bar{\xi}^2$ instead plays the rôle of a "mass term" and requires a controlling parameter $\ell'$, while $\mu$ appearing in (B.28) cannot be defined and is instead replaced by $\mu_p$.

Given these considerations, the ideal order hydrostatic effective action takes the form

$$S^{\text{HS}}[\tau_\mu, h_{\mu\nu}, A_\mu; \phi] = \int d^{d+1}x\, e P(T, \vec{u}^2, \mu, \nu, \bar{\xi}^2) + \mathcal{O}(\partial)\,. \qquad (B.29)$$

A generic variation of this effective action can be parameterised according to

$$\delta S^{\text{HS}} = \int d^{d+1}x\, e\left[-T^\mu\delta\tau_\mu + \frac{1}{2}\mathcal{T}^{\mu\nu}\delta h_{\mu\nu} + J^\mu\delta A_\mu + K_{\phi,\text{HS}}\delta\phi\right]\,, \qquad (B.30)$$

where $T^\mu$ is the energy current, $\mathcal{T}^{\mu\nu}$ the stress-momentum tensor, $J^\mu$ the charge current and $K_{\phi,\text{HS}} = 0$ the Goldstone equation in equilibrium. The U(1) and diffeomorphism Ward identities are, respectively,

$$\nabla_\mu J^\mu = K_{\phi,\text{HS}}\,, \qquad \nabla_\mu T^\mu{}_\nu = -J^\mu F_{\mu\nu} - K_{\phi,\text{HS}}\xi_\nu\,. \qquad (B.31)$$

We remind the reader that we are working under the assumption of the absence of torsion. Using the action variation (B.30) and the variational formulae in Appendix A we can extract the ideal order currents

$$\begin{aligned}
T^\mu &= Pv^\mu + sTu^\mu + m\vec{u}^2 u^\mu + \mu\rho u^\mu - fv^\rho h^{\sigma\mu}\xi_\rho\xi_\sigma + N(\nu u^\mu - v^\nu\xi_\nu\vec{u}^\mu)\,, \\
\mathcal{T}^{\mu\nu} &= Ph^{\mu\nu} + mu^\mu u^\nu - f\xi_\rho\xi_\sigma h^{\rho\mu}h^{\sigma\nu} - 2N\xi_\rho h^{\rho(\mu}v^{\nu)}\,, \\
J^\mu &= -\rho v^\mu + (\rho + N)\vec{u}^\mu + fh^{\mu\nu}\xi_\nu\,,
\end{aligned} \qquad (B.32)$$

where we introduced the entropy density $s$, kinetic mass $m$, charge density $\rho$, spatial superfluid density $N$ and superfluid density $f$ via the Gibbs–Duhem relation[34]

$$dP = sdT + \frac{1}{2}md\vec{u}^2 + \rho d\mu + Nd\nu + \frac{1}{2}fd\bar{\xi}^2\,. \qquad (B.33)$$

---

[34]We note that our Gibbs–Duhem relation in (B.33) is more general than the one presented in [79]. In particular, by generalising the responses in [79] to $g^i = mv_n^i + Nv_s^i$ and $h^i = -Nv_n^i - f_sv_s^i$, the authors of [79] could have included the independent response $N$ to the scalar $\nu$ that we introduced in (B.28). The ideal order thermodynamic properties of Aristotelian superfluids that we consider here are thus more general than what has so far been studied in the literature.

We note that the currents presented in (B.32) are precisely the same as those for $s$-wave superfluids at ideal order (5.13) with $\xi_\mu$ replaced by $\mathcal{B}_\mu$. The Euler relation is determined by the relation $\mathcal{E} = \tau_\mu T^\mu$, yielding $\mathcal{E} + P = sT + m\vec{u}^2 + \mu\rho + N\nu$. Finally, the equilibrium equation for $\phi$ is given by

$$K_{\phi,\text{HS}} = h^{\mu\nu}\nabla_\mu(f\xi_\nu) + \nabla_\mu(N\vec{u}^\mu)\,. \tag{B.34}$$

We will now determine the non-equilibrium corrections to the Goldstone equation of motion.

### B.2.2 Entropy production and the Josephson equation

In this section we briefly look at entropy production in order to derive the Josephson equation. Following the same footsteps as in the relativistic case in Section B.1.2, we can write the off-shell second law of thermodynamics as

$$\nabla_\mu S^\mu - \frac{u^\nu}{T}\left(-\nabla_\mu T^\mu{}_\nu - J^\mu F_{\mu\nu}\right) + \frac{\mu}{T}\nabla_\mu J^\mu - K_\phi \delta_{\mathscr{B}}\phi = \Delta \geq 0\,, \tag{B.35}$$

where $K_\phi = 0$ is the Josephson equation out of equilibrium that we wish to determine. Written in this way, the second law of thermodynamics takes the same form as its relativistic counterpart in (B.18). Analogously, given the gradient scheme employed here, we derive the same Josephson equation (B.19) but with the hydrostatic correction in (B.19) replaced by $K_{\phi,\text{HS}}$ given in (B.34). This concludes our analysis of ideal Aristotelian superfluids.

## C  Alternative gradient expansion for $s$-wave superfluids

In this appendix we give details of the gradient scheme that we first considered applying to $s$-wave superfluids. This consists of requiring, as for $p$-wave superfluids, that $\mathcal{A}_{\mu\nu} \sim \mathcal{O}(1)$ leading to an ideal order superfluid velocity and $\Psi_\mu \sim \mathcal{O}(\partial^{-1})$, as in (4.11). Such scaling for the vector Goldstone implies a similar version of (4.12) and an unusual scaling for the U(1) Goldstone field, namely

$$h^\mu_\nu \mathcal{B}_\mu \sim \mathcal{O}(\partial^{-1})\,, \qquad \phi \sim \mathcal{O}(\partial^{-2})\,. \tag{C.1}$$

In turn the requirement that $v^\mu \mathcal{B}_\mu \sim \mathcal{O}(1)$ leads directly to the anisotropic scaling between time and space derivatives as in (4.14) and the scalings (4.13). As such, the gradient expansion is the same as for $p$-wave fracton superfluids but with the addition of the scaling of $\phi$ as in (C.1). We can now begin to build invariant ideal order scalars and construct the chemical potential (5.10). We note, however, that since we require $\mu_p \sim \mathcal{O}(1)$ we deduce again the gradient suppression of $\vec{u}^\mu \sim \mathcal{O}(\partial)$ as in (4.17). Again, in equilibrium, $\mu_p = u^\mu \mathcal{B}_\mu$ and so $u^\mu \mathcal{B}_\mu$ is not independent.

In addition we can introduce two ideal order scalars that do not have a counterpart in $p$-wave fracton superfluids, namely

$$\nu = h_\nu^\mu u^\nu \mathcal{B}_\mu \,, \qquad \Omega = h^{\mu\nu} \nabla_\mu \mathcal{B}_\nu \,. \tag{C.2}$$

The $\nu$ thermodynamic potential is also present at ideal order in the scheme employed in Section 5 and also features in the set of ideal order scalars in conventional Aristotelian superfluids (see Appendix B.2 above). The other scalar $\Omega$ is of ideal order in this gradient expansion scheme but first order in the other scheme and for Aristotelian superfluids. We begin to see that from the point of view of the gradient expansion in this appendix, ideal order $s$-wave superfluids share some similarities with Aristotelian superfluids but also several differences, in constrast with the scheme we adopted throughout the main text.

Additional ideal order scalars can be built but they are also not independent in equilibrium, in particular

$$v^\mu \mathcal{B}_\mu = \nu - \mu_p \,, \qquad h^{\mu\nu}(A_{\mu\nu} + \nabla_\mu(B_\mu + \partial_\mu \phi)) = \xi + \Omega \,, \tag{C.3}$$

where $\xi = h^{\mu\nu} \mathcal{A}_{\mu\nu}$ was introduced in (4.18). The scalar $\vec{u}^2$ is now pushed to $\mathcal{O}(\partial^2)$. This phase of $s$-wave fracton superfluids also allows to introduce a different class of scalars, in particular

$$\mathcal{B}^2 = h^{\mu\nu} \mathcal{B}_\mu \mathcal{B}_\nu \,, \tag{C.4}$$

which is $\mathcal{O}(\partial^{-2})$. This is analogous to the Goldstone mass/pinning terms explored in [38, 42, 65], but since it does not originate from explicit symmetry breaking it leads to different effects. In order to do so we need to introduce a new bookkeeping parameter $\tilde{\ell}$, which we take to be of order $\mathcal{O}(\partial)$ to control the strength of the response to $\mathcal{B}^2$.

With these scalars and gradient expansion we can write the hydrostatic effective action as

$$S^{\mathrm{HS}}[\tau_\mu, h_{\mu\nu}, B_\mu, A_{\mu\nu}; \Psi_\mu, \phi] = \int d^{d+1}x \, e P(T, \mu_p, \nu, \mathcal{B}^2, \xi) + \mathcal{O}(\nabla_\mu \mathcal{B}_\nu) + \mathcal{O}(\mathcal{A}_{\mu\nu}^2) + \mathcal{O}(\partial) \,, \tag{C.5}$$

where, for simplicity, we have ignored the scalar $\Omega$ and other possible scalars of the abstract form $\mathcal{A}\nabla\mathcal{B}$. Using the second variation in (5.3) we can extract the gauge invariant stresses and currents in the form

$$
\begin{aligned}
T_\Psi^\mu &= P v^\mu + (sT + \mu_p \rho + \nu N) u^\mu + (\mu_p - \nu) N \vec{u}^\mu + \tilde{\ell}^2 (\mu_p - \nu) W h^{\sigma\mu} \mathcal{B}_\sigma + \mathcal{O}(\partial) \,, \\
\mathcal{T}_\Psi^{\mu\nu} &= P h^{\mu\nu} - 2 f_s \mathcal{A}_{\rho\sigma} h^{\rho\mu} h^{\nu\sigma} - 2 N \mathcal{B}_\rho h^{\rho(\mu} v^{\nu)} - \tilde{\ell}^2 W h^{\rho(\mu} h^{\nu)\sigma} \mathcal{B}_\rho \mathcal{B}_\sigma + \mathcal{O}(\partial) \\
J^\mu &= \rho v^\mu - (\rho + N) \vec{u}^\mu - \tilde{\ell}^2 W h^{\mu\nu} \mathcal{B}_\nu + \mathcal{O}(\partial) \,, \\
J^{\mu\nu} &= f_s h^{\mu\nu} + \mathcal{O}(\partial) \,,
\end{aligned}
\tag{C.6}
$$

where the entropy density $s$, charge density $\rho$, U(1) superfluid density $N$, mass parameter $W$ and dipole superfluid density $f_s$ and energy density $\mathcal{E}$ are defined via the Gibbs–Duhem and Euler relations, respectively

$$dP = s\,dT + \rho\,d\mu_p + N\,d\nu + \frac{1}{2}\tilde{\ell}^2 W\,d\mathcal{B}^2 + f_s\,d\xi\,, \quad \mathcal{E} + P = sT + \mu_p\rho + \nu N\,. \quad \text{(C.7)}$$

We note that even though the stresses computed are gauge invariant, they still give rise to a non-trivial spatial momentum $p_\mu = \tau_\nu \mathcal{T}_\Psi^{\nu\lambda} h_{\mu\lambda} = N\mathcal{B}_\nu h^\nu{}_\mu \sim \mathcal{O}(\partial^{-1})$ which is of the same gradient order as for the $p$-wave case in (4.44).[35] Finally, we can extract both Goldstone equilibrium equations from (5.12) by explicit variation with respect to $\phi$ and $\Psi_\mu$. Specifically we find

$$\begin{aligned}
K_{\phi,\text{HS}} &= -\nabla_\mu\left(N\vec{u}^\mu\right) - \tilde{\ell}^2 h^{\mu\nu}\nabla_\mu(W\mathcal{B}_\nu)\,, \\
K_{\Psi,\text{HS}}^\mu &= -(\rho + N)\vec{u}^\mu - h^{\mu\nu}\partial_\nu f_s - \tilde{\ell}^2 W h^{\mu\nu}\mathcal{B}_\nu\,.
\end{aligned} \quad \text{(C.8)}$$

We see that when the equation of motion for $\Psi_\mu$ is satisfied, the fluid velocity is given in terms of derivatives of $f_s$ and $\mathcal{B}_\nu$, which is similar to the $p$-wave case. In addition, it is possible to have non-trivial spatial fluid velocities in equilibrium by appropriately tuning $h_\nu^\mu \mathcal{B}_\mu$.

As with the gradient scheme of Section 5 we can distinguish two different regimes. If we would allow $\tilde{\ell}$ to be very large $\tilde{\ell} \sim \mathcal{O}(1)$ then we can use this Ward identity to set $h_\nu^\mu \mathcal{B}_\mu = 0$ to all orders in the derivative expansion and eliminate $\Psi_\mu$ from the theory leading to a U(1) fracton superfluid. On the other hand, if we take $\tilde{\ell} \sim \mathcal{O}(\partial)$, the effects of $h_\nu^\mu \mathcal{B}_\mu$ become significant and define a pinned $s$-wave fracton superfluid. The results of entropy production in Section 5.3 remain unchanged with this different gradient expansion, however given the gradient ordering of $\Psi_\mu$ employed here we can set $\Sigma_\mu^B = 0$ to all orders in the gradient expansion as for $p$-wave fracton superfluids.

## C.1 Linearised equations and modes

We now wish to compute the linearised spectrum with this gradient expansion scheme. We use the same equilibrium states as described in Section 5.4 and obtain the following linearised equations

$$\begin{aligned}
\partial_t \delta\mathcal{E} &= -\tilde{\ell}^2 W_0 \mu_0 (\partial^i \partial_i \delta\phi - \partial^i \delta\Psi_i) - \partial_i \delta v^i (s_0 T_0 + \mu_0(N_0 + \rho_0)) - f_s^0 \partial_t \partial^i \delta\Psi_i\,, \\
\partial_t \delta\Psi_i &= \frac{1}{(N_0 + \rho_0)}\left[\partial_i \delta P + N_0 \partial_t \partial_i \delta\phi - f_s^0 \partial_i \partial^j \delta\Psi_j\right]\,, \\
\partial_t \delta\rho &= -(N_0 + \rho_0)\partial_i \delta v^i - \tilde{\ell}^2 W_0(\partial^i \partial_i \delta\phi - \partial^i \delta\Psi_i)\,, \\
\partial^i \delta f_s &= -(N_0 + \rho_0)\delta v^i - \tilde{\ell}^2 W_0(\partial^i \delta\phi - \delta\Psi^i)\,, \\
\partial_t \delta\phi &= \delta\mu_p + \frac{T_0}{\alpha}\left[N_0 \partial_i \delta v^i + \tilde{\ell}^2 W_0(\partial^i \partial_i \delta\phi - \partial^i \delta\Psi_i)\right]\,.
\end{aligned} \quad \text{(C.9)}$$

---

[35]We see that for $s$-wave superfluids, the gauge invariant stresses can have terms of the order $\mathcal{O}(\partial^{-1})$ as discussed in Section 4.2 for $p$-wave superfluids.

We can prooceed and solve this system of equations as in Section 5.4. If we turn off the mass term ($\tilde{\ell} = 0$) we find again the magnon modes (5.32) with velocity given by (5.33) but where the "tildes" are removed from the expression. Contrary to the other gradient scheme, these magnon modes appear at ideal order.

Turning on the mass parameter by considering $\tilde{\ell} \neq 0$ we find in the pinned $s$-wave regime $\tilde{\ell} \ll k \ll L_T^{-1}$ a pair of magnon modes and a pair of sound modes with the form (5.36) but where $\hat{v}$ is given in terms of thermodynamic quantities without "tildes" In contrast to the other gradient scheme, both these pairs of modes are part of the low energy ideal order spectrum.

In the opposite U(1) fracton superfluid regime in which $k \ll \tilde{\ell} \ll L_T^{-1}$, we find a pair of magnon-like modes as in (5.43) with "magnon velocity" given by (5.44) but without "tildes". In addition we also find a pair of sound modes of the form (5.40) but with $v_s$ and $\Gamma_s$ given in (5.41) with $\tilde{m}_0 = 0$ and removing the "tilde" from all other quantities as well as with $\ell'$ replaced by $\tilde{\ell}$. To note is that in the absence of $\tilde{m}_0$ from (5.41), both the velocity and attenuation scale with $\tilde{\ell}$. This means that in the strict U(1) regime of $s$-wave superfluids in which $\tilde{\ell} \gg k$ we do not find the sound mode with velocity and attenuation in (5.42). This is the reason why in the main text we adopted the other gradient scheme since when adding a second order term (according to this gradient scheme) proportional to $\vec{u}^2$ to the partition function (C.5), the sound mode with velocity (5.42) suddenly appears. This signals a mismatch between the gradient expansion and an expansion in wavenumber, appearing to be a case of UV/IR mixing. Thus in the low energy regime ($k \ll \tilde{\ell} \ll L_T^{-1}$), according to this scheme, we only find a pair of magnon modes at ideal order. The phase transition depicted in the right hand side of Fig. 1 is still accurate according to this gradient scheme as long we replace $\ell'$ by $\tilde{\ell}$ and we remove the sound mode in the U(1) regime as it does not feature the low energy spectrum, though it appears at next order when including $\vec{u}^2$ in the partition function. It should be noted that at second order in derivatives both schemes give the same spectrum of modes, at least to what concerns the effects and terms we considered in this paper. We give a few more details about the U(1) regime below.

## C.2   U(1) regime

For the U(1) regime studied in Section 5.5 we can also employ the gradient scheme discussed in this appendix. This is done by requiring $\tilde{\mathcal{A}}_{\mu\nu} \sim \mathcal{O}(1)$ and following the same footsteps as above leading to (C.1) and $\vec{u}^\mu \sim \mathcal{O}(\partial)$. In this context we can introduce the ideal order hydrostatic effective action in the form

$$S[\tau_\mu, h_{\mu\nu}, \Phi, \tilde{A}_{\mu\nu}; \phi] = \int d^{d+1}x \, e P(T, \mu_U, \xi) + \mathcal{O}(\tilde{\mathcal{A}}_{\mu\nu}^2). \qquad \text{(C.10)}$$

Using (5.49) we can easily extract the stresses and currents and obtain

$$\bar{T}_\phi^\mu = v^\mu P - sTv^\mu + sT\vec{u}^\mu + \mathcal{O}(\partial)\,,$$
$$\bar{\mathcal{T}}_\phi^{\mu\nu} = Ph^{\mu\nu} - 2f_s\tilde{\mathcal{A}}_{\rho\sigma}h^{\rho(\mu}h^{\nu)\sigma} + \mathcal{O}(\partial)\,, \qquad \text{(C.11)}$$
$$\bar{J}^0 = -\rho + \mathcal{O}(\partial)\,, \quad J^{\mu\nu} = f_sh^{\mu\nu} + \mathcal{O}(\partial)\,,$$

where we have defined the entropy density, charge density and superfluid density via the Gibbs–Duhem relation $dP = sdT + \rho d\mu_U + f_s d\xi$. Explicitly computing the energy density using (5.56) we obtain the Euler relation $\mathcal{E}+P = sT$, while momentum $p^\mu = \tau_\nu \bar{\mathcal{T}}_\phi^{\mu\nu} = 0$. Finally, we can obtain the equilibrium equation for the Goldstone $\phi$ by varying (C.10) with respect to it yielding

$$K_\phi = \nabla_\mu(\rho\vec{u}^\mu) + h^{\mu\nu}\nabla_\mu\nabla_\nu f_s\,, \qquad \text{(C.12)}$$

where we have used the Ward identity (5.51). We note that this Ward identity does not impose a restriction on $\vec{u}^\mu$ and hence we can have non-vanishing equilibrium velocities. The entropy production analysis remains the same as in Section 5.5.2. We now look at linearised fluctuations around the equilibrium states of Section 5.5.3 giving rise to the equations

$$\partial_t\delta P = s_0\partial_t\delta T + T_0\partial_t\delta s + s_0T_0\partial_i\delta v^i + \rho_0\partial_t\partial_t\delta\phi + f_s^0\partial_t\partial^i\partial_i\delta\phi\,,$$
$$\partial_i\delta P = \rho_0\partial_i\partial_t\delta\phi + f_s^0\partial_i\partial^j\partial_j\delta\phi\,,$$
$$\partial_t\delta\rho = \partial^i\partial_i\delta f_s\,, \qquad \text{(C.13)}$$
$$\partial_t\delta\phi = \delta\mu_U - \frac{T_0}{\tilde{\alpha}_0}\left[\partial_i(\rho_0\delta v^i) + \partial^i\partial_i\delta f_s\right]\,.$$

Using plane wave perturbations as in the previous sections, these equations lead to three non-vanishing modes of the form

$$\omega = \pm\omega_M k^2 - \frac{i}{2}\Gamma_M k^4\,, \qquad \omega = -i\gamma - \frac{i}{2}\Gamma_U k^4\,, \qquad \text{(C.14)}$$

where $\omega_M^2$ is given in (5.44) but without the "tildes" and where $\Gamma_U$ is a complicated function of the thermodynamic parameters. We thus see that at ideal order we find a pair of magnon modes but no sound modes. In addition we also find a gapped mode with damping $\gamma$ given by

$$\gamma = \frac{\alpha_0}{\rho_0^2}\frac{s_0^2(\partial\rho/\partial\mu_U)_0}{s_0(\partial\rho/\partial\mu_U)_0 - T_0(\partial s/\partial\mu_U)_0^2 + T_0(\partial s/\partial T)_0(\partial\rho/\partial\mu_U)_0}\,. \qquad \text{(C.15)}$$

We see that the attenuation of the gapped mode is controlled by the dissipative coefficient $\alpha$. For stability at arbitrary $k$ we require $\gamma, \Gamma_U \geq 0$. We note, however, that when taking into account first order gradient corrections we expect additional contributions to (C.14). As we mentioned in Section 5.4, we should recover the low

energy spectrum of the $s$-wave superfluids (ignoring gapped modes or other higher energy modes) when $\tilde{\ell} \gg k$. Indeed, the magnon-like modes in (C.14) are precisely the magnon-like modes of the $s$-wave (5.43) with "magnon velocity" given by (5.44) but without "tildes". We thus also find perfect agreement of the two low energy spectra using this gradient scheme. At second order in derivatives, including the scalar $\vec{u}^2$ in the partition function leads to the sound mode in (5.70).

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
