# Peer review of "Ideal fracton superfluids"

_SciPost Physics_

## Round 1 · Referee Report · Anonymous (Referee 1) · 2023-6-4

Strengths

1-Clear, unambiguous, and important conclusion: there is no such thing as a fracton fluid, only superfluids (i.e., at least part of the fractonic symmetry must be spontaneously broken).
2-Encompasses previous work. Rather, encapsulates most previous work in the s-wave superfluid state.
3-Describes in detail a different phase of fracton superfluid (p-wave) based on a different pattern of symmetry breaking.
4-Clear presentation.

Weaknesses

1-Lacks a general treatment which includes intrinsic dipoles.
2-Significant overlap with contemporaneous work.
3-Minor literature gaps.

Report

The SciPost general acceptance criteria are met: the presentation is clear and understandable, the abstract is detailed and the introduction properly explains the problem and, along with the conclusion, summarizes the achievements of the paper, there are sufficient details to reconstruct the main findings, the central references are present with only minor gaps missing, and there are clear paths toward future work.

As regards the Expectations (at least one required) for publication in SciPost, I think that the paper mostly fits in point 3: "Open a new pathway in an existing or a new research direction, with clear potential for multipronged follow-up work", but also point 2: "Present a breakthrough on a previously-identified and long-standing research stumbling block" in the sense that it addresses the question of the existence (or, in this case, non-existence) of fractonic fluids (versus superfluids). It details two types of spontaneous symmetry breaking: breaking of both the charge U(1) and dipole (s-wave) and breaking of just the dipole (p-wave). The paper performs the hydrodynamic mode analysis of both types.

I merely have some small requests for improvement. With these small changes, I recommend the article be published in SciPost.

Requested changes

1-I would request that the authors dedicate more space and effort into hashing out the differences between their paper and their Ref. [46]. I think that the phrase "significant overlap" is insufficient here. For example, reference could be made to the derivative counting used in the main text, which is the same as (or taken from?) [46], versus that in Appendix C. Also, obviously, since [46] has come out already, the bibliographic reference should be updated.
2-Cite “Efficiently preparing Schrodinger’s cat, fractons and non-Abelian topological order in quantum devices” by Ruben Verresen, Nathanan Tantivasadakarn, and Ashvin Vishwanath https://arxiv.org/abs/ 2112.03061 in addition to [10] for examples of testing fractons using ultracold atoms. In this case, we’re talking about Rydberg atom arrays.
3-Cite “Hyperbolic Fracton Model, Subsystem Symmetry, and Holography” by Han Yan https://arxiv. org/abs/1807.05942 in addition to [11] for examples of fractons in holography.
4-I think these are typos (math typos here; I'm not going to correct grammar or spelling): right before eq. (2.28), $F$ should be $\mathscr{F}$ and in eq. (2.30), I think the left hand side should be $\tilde{A}{}_{\mu}^{a} e_{\nu a} - \tilde{A}{}_{\nu}^{a} e_{\mu a}$.

  • validity: high
  • significance: high
  • originality: high
  • clarity: high
  • formatting: excellent
  • grammar: excellent

Author:  Emil Have  on 2024-01-04  [id 4219]

(in reply to Report 1 on 2023-06-04)
Category:
answer to question

We are grateful to the referee for the report. In the attached file, the points raised in the report are addressed in detail.

Attachment:

Report_1_response.pdf

---

## Round 1 · Referee Report · Daniele Musso (Referee 2) · 2023-6-7

Report

The paper adopts a hydrodynamic approach in order to characterize the behavior of many-body quantum systems with emergent fracton symmetries. The topic is timely and interesting for the high-energy community (at least) but most probably beyond it. Indeed, the paper is well connected with the literature, both recent and less so.

The paper is extensive, detailed and both systematic and pedagogic in aim. However, the current version falls short in these latter aspects. A revision in the presentation of some aspects seems to be in order to the purpose of improving clearness and to ease the understanding. Besides, it appears there are some issues in relation to some of the reported results. The remarks in the attached file go in the direction of improving on these aspects and I regard it necessary to address them before recommending the manuscript for publication on SciPost.

Let me strongly encourage the authors to pursue on, especially in the direction of a deeper understanding of the relations among the modes that they study with the Nambu-Goldstone structure of fracton theories, for instance their counting and dispersion properties.

My congratulations for the work done.

Attachment

  • validity: -
  • significance: -
  • originality: -
  • clarity: -
  • formatting: -
  • grammar: -

Author:  Emil Have  on 2024-01-04  [id 4220]

(in reply to Report 2 by Daniele Musso on 2023-06-07)
Category:
answer to question

We are grateful to the referee for the report. In the attached file, the points raised in the report are addressed in detail.

Attachment:

Report_2_response.pdf

---

## Editorial Decision

resubmitted